# Transdiagnostic compulsivity is associated with reduced reminder setting, only partially attributable to overconfidence

**Annika Boldt[1]\*, Celine Ann Fox[2], Claire M Gillan[2], Sam Gilbert[1]**

[1]Institute of Cognitive Neuroscience, University College London, London, United Kingdom; [2]School of Psychology, Trinity College Dublin, Dublin, Ireland

## eLife Assessment

This **important** work addresses the relationship between the transdiagnostic compulsivity dimension and confidence as well as confidence-related behaviours like reminder setting. The relationship between confidence and compulsive disorders has recently received a lot of attention and has been considered to be a key cognitive change. The authors paired an elegant experimental design and pre-registration to give **convincing** evidence of the relationship between compulsivity, reminder setting, and confidence. In the revised version they thoroughly addressed the reviewer's comments, in particular adding new analyses clarifying how their findings relate to prediction error based learning as well as presenting additional recovery analyses and psychometric curves further strengthening the manuscript.

**\*For correspondence:**
a.boldt@ucl.ac.uk

**Competing interest:** The authors declare that no competing interests exist.

**Abstract** In the current study, we explored the behavioural and cognitive correlates of the transdiagnostic trait 'compulsive behaviour and intrusive thought' (CIT) in humans. CIT is associated with impaired metacognition, which in turn has been associated with cognitive offloading behaviours such as external reminder setting that play a key role in fulfilling cognitive goals. In an online study (*N*=600), we investigated individual differences in compulsivity, metacognition, and external reminder usage. Compulsive individuals had reduced preference for external reminders. This was partially, but not fully, attributable to their relative overconfidence. In contrast to previous studies, we found no evidence for an impaired confidence-action link: compulsive individuals used their metacognition to guide offloading just as much as their non-compulsive counterparts. Given the compensatory nature of cognitive offloading, our findings imply that compulsive individuals are at increased risk of inadequate external memory support. Along with transdiagnostic variation in the general population, this finding could also have implications for clinical conditions, such as obsessive-compulsive disorder (OCD).

## Introduction

In recent studies of clinically relevant individual differences, there has been a paradigm shift towards the study of transdiagnostic traits, challenging the traditional, diagnostic approach. Using factor analysis, temporally stable (see *Fox et al., 2023*; *Sookud et al., 2024*), transdiagnostic phenotypes can be extracted from extensive symptom datasets (*Wise et al., 2023*). These traits are not confined to a single clinical diagnosis but instead can span a range of conditions, at the same time addressing the diagnostic heterogeneity within conditions, such as obsessive-compulsive disorder (OCD; e.g. *Gillan et al., 2016*; *Wise and Dolan, 2020*). There are obvious practical benefits of these methodologies, such as their potential to reduce the clinical burden by making the treatment of comorbid

**eLife digest** You have just been prescribed a new course of antibiotics; will you schedule alarms to make sure you take your treatment as you should – three times a day, every day, for the next week? Or will you trust yourself to remember to do so unprompted?

You may find it easy to make this choice, yet it is in fact a rather complex task. Research has shown that the use of reminders (a process known as cognitive offloading) is guided, in part, by how confident we are about our ability to remember. Accurately assessing our own cognitive skills, however, can be shaped by a range of psychological factors. People with high levels of compulsivity, for example, tend to struggle with judging their own abilities. This trait, commonly present in a range of mental health conditions such as obsessive-compulsive disorder, is characterized by repetitive behaviors and intrusive thoughts. Here, Boldt et al. investigate whether differences in compulsivity can impact how and when people choose to set reminders.

To do so, an online study was conducted on 600 adults from the general population. Before completing a highly demanding memory task, participants first answered questionnaires assessing traits including compulsivity and anxiety. They were also asked to predict how well they would perform on the test.

When going through the memory task, participants could choose to use reminders to help themselves at the start of each trial. By doing so, however, they knew they would earn fewer points for each accurate answer given.

The results showed that individuals who scored higher on compulsivity tended to set fewer reminders. This was partly because they were more confident in their memory than other participants, but also because compulsivity itself seemed to directly reduce reminder use.

Taken together, these findings suggest that people who are highly compulsive may not adequately use memory aids even when they might need them. Although none of the participants had a clinical diagnosis, the results could inform future studies of conditions such as obsessive-compulsive disorder, as well as guide the design of interventions to support memory and daily functioning.

conditions more efficient and effective (*Harvey, 2025*). At the same time, they contribute valuable insights into mental health conditions by increasing statistical power and opening new avenues of inquiry (*Dalgleish et al., 2020*).

In the present study, our focus lies on the latter with the goal to investigate the downstream cognitive and behavioural correlates associated with transdiagnostic compulsivity. This symptom dimension represents a clinical concept characterised by an inability to regulate repetitive behaviours that are harmful to oneself, commonly observed in a variety of conditions, particularly OCD, schizophrenia, addiction, and eating disorders. Previous research links transdiagnostic compulsivity to impairments in metacognition, defined as thinking about one's own thoughts, encompassing a broad spectrum of self-reflective signals, such as feelings of confidence (e.g. *Rouault et al., 2018*; *Seow and Gillan, 2020*; *Benwell et al., 2022*; *Fox et al., 2023*; *Fox et al., 2024*; *Hoven et al., 2023a*). Other studies have shown that metacognitive signals such as feelings of confidence guide cognitive offloading strategies like setting external reminders as memory aids (e.g. *Gilbert, 2015*; *Boldt and Gilbert, 2019*). Here, we aim to bridge these two literatures by investigating compulsivity, metacognition, and cognitive offloading within a single experimental paradigm. While compulsivity and cognitive offloading have both separately been linked to metacognition, the relationship between the two – with metacognition as a potential mediating factor – has not previously been systematically examined. This matters because cognitive offloading plays an integral role in our daily lives and is a key contributor to our effectiveness as cognitive agents (*Gilbert et al., 2023*).

## Metacognition guides reminder setting

Reminders constitute an example of cognitive offloading, defined as the use of physical action to reduce the cognitive demands of a task. By offloading memory demands this way, we not only increase the likelihood of successfully completing tasks (*Boldt and Gilbert, 2019*), but we may also free up cognitive resources for other activities (*Dupont et al., 2023*). Choosing between setting a reminder and relying on memory is not a trivial matter. Prior research has emphasised the role of metacognition

in determining when individuals resort to cognitive offloading (*Gilbert et al., 2023*; *Gilbert, 2015*; *Boldt and Gilbert, 2019*; *Sachdeva and Gilbert, 2020*; *Risko and Gilbert, 2016*): People tend to set more reminders when they feel less confident. In other words, people tend to set reminders when they *think* that they will forget, and this effect holds even after taking into account actual memory ability (e.g. *Boldt and Gilbert, 2019*). The link between confidence and offloading is observed both for situational fluctuations in confidence due to varying task difficulties (*state* variable; *Boldt and Gilbert, 2022*) and for a general predisposition towards over- or underconfidence (*trait* variable; *Boldt and Gilbert, 2019*).

## Metacognition, compulsivity, and checking behaviours

Given the known metacognitive impairments associated with compulsivity, changes in reminder-setting behaviour are plausible. More specifically, individuals characterised by transdiagnostic compulsivity have been consistently found to exhibit overconfidence (*Rouault et al., 2018*; *Seow and Gillan, 2020*; *Benwell et al., 2022*; *Fox et al., 2023*; *Fox et al., 2024*; *Hoven et al., 2023a*). If we consider the link between reminder setting and confidence, this implies a reduced likelihood of utilising external aids, such as reminders. However, while transdiagnostic compulsivity is liked to overconfidence, the opposite pattern of underconfidence is more common in patients with OCD, a compulsive disorder (as reviewed in *Hoven et al., 2019*). Recent research suggests that metacognitive impairments in transdiagnostic compulsivity and OCD may originate from different mechanisms (*Hoven et al., 2023b*; *Hoven et al., 2023c*), advising caution against broad generalisations between these groups. It should also be noted that the composite measure of transdiagnostic compulsivity includes questionnaire items linked not only with OCD but also other clinical conditions such as eating disorders (*Tasca et al., 2011*; *Gillan et al., 2016*). This results in an overlap between transdiagnostic compulsivity and other traits such as rigid perfectionism.

Despite opposite trends in metacognitive monitoring performance (under- versus overconfidence), individuals high in transdiagnostic compulsivity and those with a diagnosis of OCD show similar impairments in metacognitive control, characterised by a disrupted connection between confidence and future actions (*Seow and Gillan, 2020*; *Vaghi et al., 2017*). Metacognitive impairments are also central to explanations of compulsive behaviours, notably in OCD patients. In such patients, compulsivity can manifest in the form of checking behaviours, e.g., checking that doors are locked or that appliances are switched off (*Den Ouden et al., 2022*). Whilst checking behaviours are also present in other compulsive disorders (e.g. 'body checking' in eating disorders; *Mountford et al., 2006*), in OCD, these checks are often repetitive and ritualised and are typically associated with obsessive thoughts. However, the exact function that checking compulsions serve is unclear; patients commonly report that they have the aim of reducing anxiety generally, preventing a feared consequence from taking place or that they are performed automatically and without thinking (*Starcevic et al., 2011*). Understanding these motivators has been challenging as studies rely on self-report of often highly individual real-world behaviours.

Some research argues that OCD patients' checking arises from low memory confidence despite intact memory (*Tolin et al., 2001*). Our study has the potential to shed some light on the link between confidence and checking: While checking behaviours can be seen as a way of ensuring that a necessary action *was* performed in the past, reminder setting is a way of ensuring that a necessary action *will be* performed in the future. In other words, a reminder can serve as a future checkpoint that allows us to revisit a task at an appropriate time to complete it, perhaps by setting an alarm on our phone, jotting down a note, or strategically placing a related object somewhere visible. Given these insights, one might expect an increased reliance on reminders among OCD patients as they strive to establish more checkpoints. By contrast, seeing as transdiagnostic compulsivity is associated with increased confidence, this could be associated with the opposite pattern: a decreased reliance on reminders.

## Three possible mechanisms for changes in reminder setting

If, as hypothesised, compulsivity is linked with altered reminder setting, this could be attributed to at least three underlying mechanisms. First is the *Metacognitive Control Mechanism*: Previous research has found that more compulsive individuals tend to have impaired metacognitive control (*Seow and Gillan, 2020*), meaning they use metacognitive signals to a lesser extent to guide future behaviour. Compulsivity is a hallmark symptom of OCD, and similar deficits in metacognitive control have been

observed in a case-control studies comparing OCD patients with healthy controls examining how confidence and action are correlated (*Vaghi et al., 2017*; though see also *Hoven et al., 2023b*; *Marzuki et al., 2022*). In the context of our study, a Metacognitive Control Mechanism would be reflected in a disrupted relationship between confidence levels and their tendency to set reminders (i.e. the interaction between the bias to be over- or underconfident and transdiagnostic 'compulsive behaviour and intrusive thought' (CIT) in a regression model predicting a bias to set reminders).

Second, more compulsive individuals might conceivably differ in their reminder-setting strategies due to an altered level of confidence. We call this the *Metacognitive Monitoring Mechanism*, which suggests that the issue arises when forming the confidence signal, rather than in its behavioural application (for clarification on metacognitive monitoring vs. control in cognitive offloading, see *Boldt and Gilbert, 2022*). Prior evidence exists for overconfidence in compulsivity (*Rouault et al., 2018*; *Seow and Gillan, 2020*; *Benwell et al., 2022*; *Fox et al., 2023*; *Fox et al., 2024*; *Hoven et al., 2023a*), which would therefore result in fewer reminders.

Lastly, there could be a direct link between compulsivity and reminder usage, independent of any metacognitive influence. We refer to this as the *Direct Mechanism* and it constitutes any possible influences that affect reminder setting in highly compulsive CIT participants outside of metacognitive mechanisms, such as perfectionism and the wish to control the task without external aids. Our study aims to differentiate between these three mechanisms. Back when we preregistered our hypotheses, only a limited number of studies about confidence and transdiagnostic CIT were available. This resulted in us hypothesising to find support for the *Metacognitive Control Mechanism* and that highly compulsive individuals would offload more due to an increased need for checkpoints. Both of these hypotheses turned out to be incorrect.

## Anxious-depressed transdiagnostic phenotype

As well as investigating individual differences in compulsivity, we also measured an anxious-depression (AD) factor. Based on the previous findings, we predicted opposite influence of these two factors on confidence. Whereas compulsivity has been linked to increased confidence, AD individuals typically display relative underconfidence (*Rouault et al., 2018*; *Seow and Gillan, 2020*; *Benwell et al., 2022*). By taking a transdiagnostic approach, we were able to jointly investigate the influence of these two factors of confidence which could potentially cancel out if they were investigated separately.

## Online reminder-setting task

In the present preregistered study, we asked 600 participants drawn from the general population to complete several individual differences questionnaires. These responses were then weighted to produce both a 'CIT' factor and an 'AD' factor (*Gillan et al., 2016*; *Wise and Dolan, 2020*). Participants' scores on these factors were then correlated with their behaviour in a reminder-setting task, which was a modified, 20 min version of the online reminder setting task developed by *Gilbert et al., 2020*; Figure 6.

Participants performed a highly demanding, short-term memory task. On some trials, they relied on internal memory alone (which typically resulted in poor accuracy); on other trials, they could set external reminders (which dramatically improved accuracy). The key manipulation was the number of points associated with the two strategies. Correct responses always earned 10 points if participants used internal memory, but a lower number of points between 2 and 9 if they used external reminders. The latter number of points varied from trial to trial, and participants were required each time to decide which strategy they preferred (e.g. 10 points for each correct response with internal memory or 6 points for each correct response with external reminders). The 'optimal indifference point' (OIP) was that point value at which an unbiased individual would be indifferent between the two strategies based on their objective accuracy in the two conditions. The 'actual indifference point' (AIP) was the point at which they were actually indifferent, based on all of their decisions. By comparing these two values, we obtained a 'reminder bias': the extent to which an individual had a pro- or anti-reminder bias relative to their individually calculated optimal strategy. Note that this is different from the absolute rate of reminder usage, because the same absolute rate might reflect inadequate use of reminders in a person with poor memory and excessive reminder usage in a person with good memory ability. Along with the reminder bias, we also calculated a metacognitive bias, which represents participants' over- or underconfidence in memory ability, relative to objective performance. Our study controlled

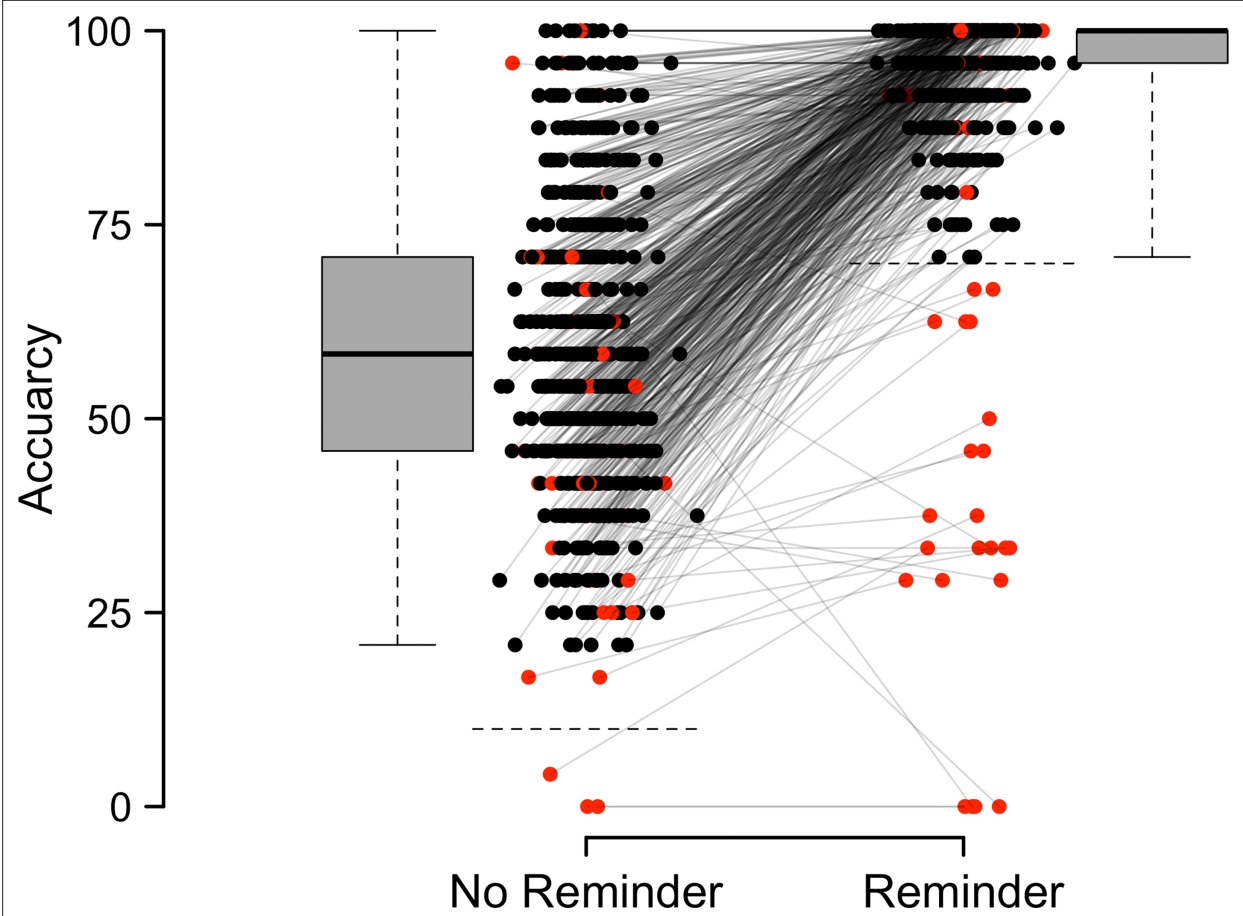

**Figure 1.** Average accuracy as a function of whether a reminder was used. 'No Reminder': forced internal condition; 'Reminder': forced external condition. Each pair of dots linked by a line indicates one participant. The red data points are excluded participants. The box plots indicate the median surrounded by the interquartile range (25th and 75th percentile). The whiskers show the minimum and maximum. The preregistered exclusion criteria for the accuracies with or without reminder are indicated as horizontal dotted lines (10% and 70%, respectively).

for age, gender, educational attainment, as well as cognitive ability (ICAR5; *Kirkegaard and Bjerrekær, 2016*), and working memory.

Previewing our results, in line with previous evidence, we found that confidence varied positively with the CIT factor and negatively with the AD factor. However, contrary to our initial expectations, more compulsive individuals offloaded less rather than more, and there was no evidence for disruption in the link between metacognition and offloading. Instead, we discovered an incomplete mediation effect: while a significant proportion of the reduced reminder setting could be attributed to overconfidence, not all the variance was accounted for by this variable. Even after controlling for it, compulsivity still predicted reduced reminder setting. This constitutes a combination of the *Metacognitive Monitoring Mechanisms* and the *Direct Mechanism*.

## Results

Here, we present the results of a preregistered online study on the relationship between reminder setting, metacognition, and transdiagnostic compulsivity. We excluded 69 out of a total of 669 participants based on our six preregistered criteria described in the Materials and methods section, leaving us with a final sample of 600 participants. All participants completed a previously validated reminder setting task in combination with 49 items from six mental health questionnaires. Three hundred and seventy-five participants identified as male, 218 as female and 7 as other. Participants were on average 32.9 years of age (min = 18; max = 76). *Figure 1* shows the included (black) and excluded (red)

data, with higher average performance for included participants when reminders were used (96.1%) compared to when people had to do the task unaided (59.2%).

We calculated six key measures for each participant:

1. The first relevant measure is the OIP. The OIP describes the reward value (2–9 points) at which an unbiased, reward-maximising participant should be indifferent between the two strategies: using reminders or relying on their own memory. The OIP is calculated from their accuracy with and without reminders. Imagine a participant who achieves 60% accuracy when using their own memory or 100% accuracy when using reminders. In this case the OIP would be 6, because scoring 6 points per item with reminders (100% accuracy) would earn the same number of points as scoring 10 points with internal memory (60% accuracy). For any reward above 6, it would be optimal to choose external reminders; for any reward below 6, it would be optimal to choose internal memory.

2. In contrast, the second relevant measure is the AIP, which is the number of points at which participants showed indifference between the two strategies. This measure is calculated by fitting a psychometric function to participants' choices at different levels of reward for targets when reminders were used. Please note that all choices were used to calculate the AIP, as participants only found out whether or not they would use a reminder after the decision was made.

3. Together, these variables can be used to calculate the third measure, the *reminder bias*, which is the difference between the OIP and the AIP and therefore reflects participants' tendency to over- or underuse reminders, relative to the optimal strategy. Note that the optimal strategy is calculated individually for each participant and will depend on their own level of performance when using internal memory and external reminders.

4. Fourth, we calculated a *metacognitive bias*, reflecting participants' over- or underconfidence. This is calculated by subtracting objective accuracy (percentage of targets remembered when

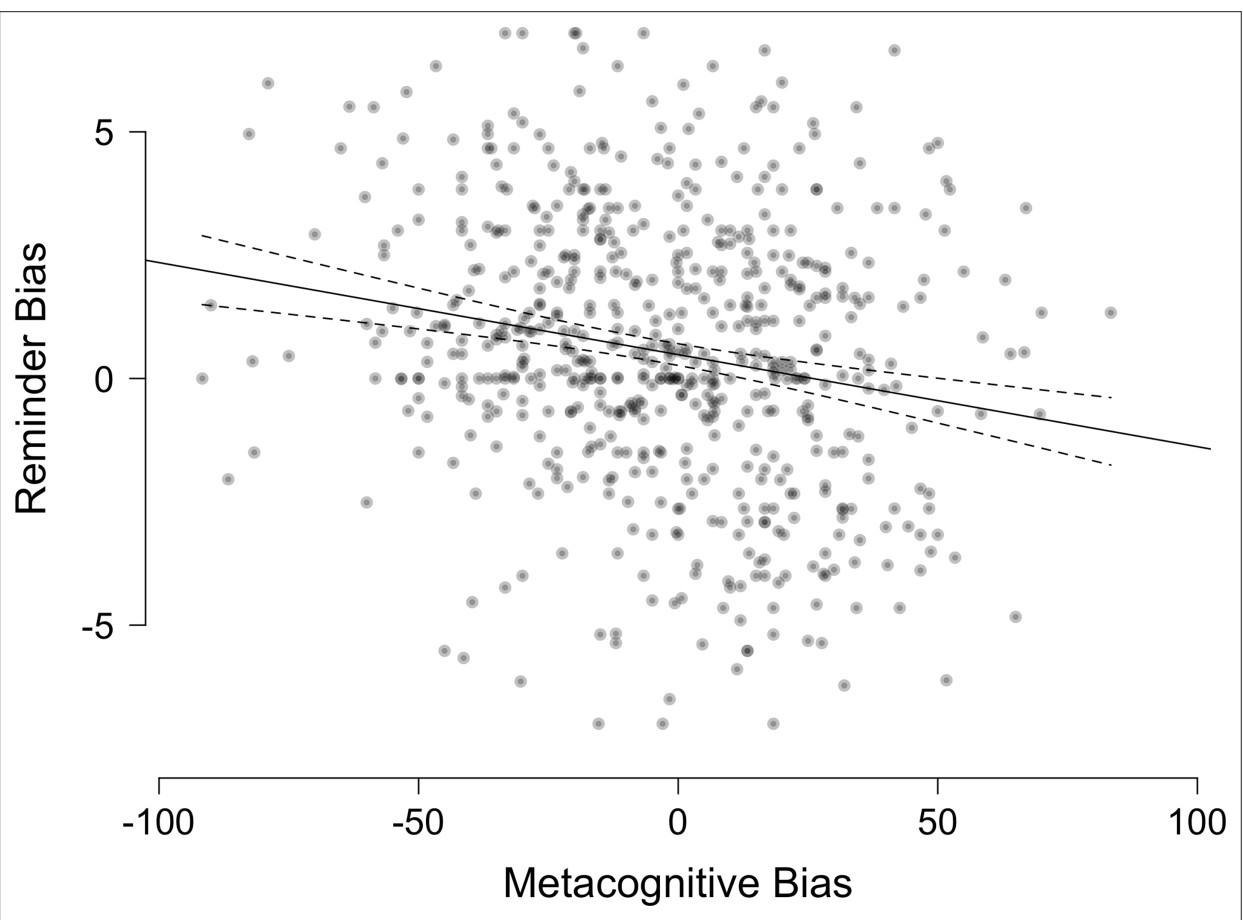

**Figure 2.** People's tendency to set reminders above or below the optimal offloading strategy (reminder bias) plotted against people's tendency towards over- or underconfidence (metacognitive bias). The solid line indicates the fitted relationship between both variables. The dashed lines represent the 95% confidence interval around it. Each circle represents a single participant.

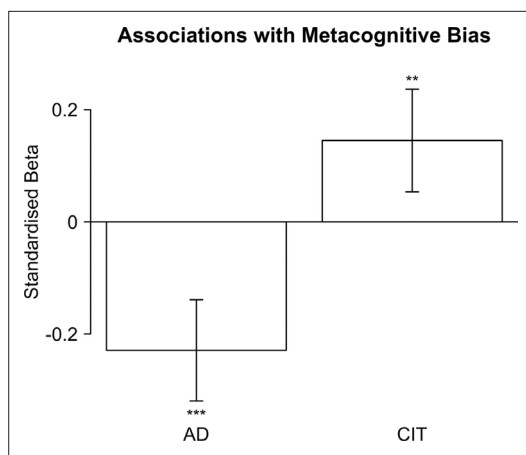

**Figure 3.** Standardised regression weights for the 'anxious-depression' (AD) factor and the 'compulsive behaviour and intrusive thought' (CIT) factor predicting metacognitive bias. Error bars indicate 95% confidence intervals. Asterisks indicate significance: '***': <0.001; '**': <0.01; '*': <0.05.

using internal memory) from the percentage that they predicted that they would be able to remember.

5. Fifth and sixth, based on the questionnaire ratings, we calculated how much someone scored on the transdiagnostic CIT and AD factors. Our analyses focus on the relationship between these key measures.

## Replication and sanity checks

In the following section, we aim, where the design allows it, to replicate four previous effects for this task. First, with Hypothesis 1, we predicted that the reminder bias and metacognitive bias are negatively correlated, replicating previous findings (as reviewed in *Gilbert et al., 2023*). This effect tests the above-mentioned link between metacognition and cognitive offloading: the less confident someone feels, the more they use reminders. There was indeed a significant negative correlation, $r=-0.2$, p<0.001 (*Figure 2*). Second, in replication of previous findings (e.g. *Gilbert et al., 2020*; *Sachdeva and Gilbert, 2020*; *Kirk et al., 2021*; *Engeler and Gilbert, 2020*), Hypothesis 2 expressed our expectation to find an excessive use of reminders reflected in significantly higher OIPs compared to AIPs. In other words, we expected the reminder bias to be greater than zero, which was indeed the case, $m=0.52$, $t(599) = 5.1$, p<0.001, $d=0.21$. Third, with Hypothesis 3, we expected to replicate that participants would be underconfident in their own memory (e.g. *Engeler and Gilbert, 2020*), expressed in an average, negative metacognitive bias. Our data supported this hypothesis, $m=-3.64$, $t(599) = -3.1$, p=0.001, $d=-0.13$. Fourth, Hypothesis 4 predicts that as in previous studies, we would find evidence for compensatory reminder use. Keeping in mind that the OIP reflects the cut-off at which participants should be indifferent between offloading and not offloading and the AIP the cut-off they actually displayed, then looking at these two measures together should show that participants with poorer memory and greater benefit from reminders (lower OIP) tend to use them more (lower AIP). Indeed, the OIP and AIP were positively correlated, suggesting participants who benefited most from reminders were more likely to use them, $r=0.36$, p<0.001. Taken together, we found that participants showed the usual hallmarks of this offloading task, using their confidence to strategically decide when to offload, general tendencies for setting reminders and for underconfidence, and compensatory reminder use.

## Testing our key hypotheses
### Elevated confidence in CIT and reduced confidence in AD

We predicted that the metacognitive bias would correlate negatively with AD (Hypothesis 8a; more AD individuals tend to be underconfident). For CIT, we preregistered a non-directional, significant link with metacognitive bias (Hypothesis H6a). We found support for both hypotheses, both for AD, $\beta=-0.23$, SE = 0.05, $t=-4.99$, p<0.001, and CIT, $\beta=0.15$, SE = 0.05, $t=3.11$, p=0.002, controlling for age, gender, and educational attainment (*Figure 3*; see also *Appendix 1—table 1*). Note that for CIT, this effect was positive, and more compulsive individuals tend to be overconfident.

We furthermore preregistered to also test this for raw confidence (percentage of circles participants predicted they will remember, rather than the accuracy-corrected metacognitive bias score; Hypotheses H8b and H6b). Indeed, the same patterns were found for both AD, $\beta=-0.29$, SE = 0.04, $t=-6.43$, p<0.001, and CIT, $\beta=0.12$, SE = 0.05, $t=2.76$, p=0.006 (see *Appendix 1—table 2*). Including scores from the cognitive ability test as an additional covariate (Hypotheses H8c and H6c, respectively) furthermore did not change the results, AD, $\beta=-0.20$, SE = 0.05, $t=-4.46$, p<0.001; CIT, $\beta=0.12$, SE = 0.05, $t=2.57$, p=0.011 (see *Appendix 1—table 3*). Taken together, these results suggest that

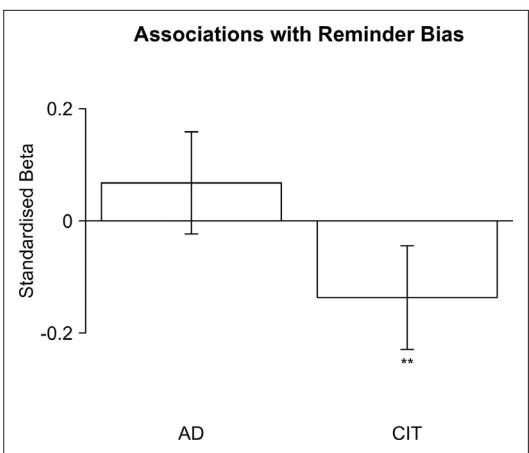

**Figure 4.** Standardised regression weights for the 'anxious-depression' (AD) factor and the 'compulsive behaviour and intrusive thought' (CIT) factor predicting reminder bias. Error bars indicate 95% confidence intervals. Asterisks indicate significance: '***': <0.001; '**': <0.01; '*': <0.05.

concordant with our hypotheses, compulsivity was linked to inflated confidence and anxiety to deflated confidence.

## Contrary to expectations, compulsivity reduced pro-offloading bias

We expected to find a positive link between CIT factor scores and reminder bias. In other words, we predicted that more compulsive individuals would show a greater pro-offloading bias, relative to the optimal strategy (Hypothesis H5a). However, our results showed the exact opposite effect with a significantly reduced reminder bias in compulsive individuals, $\beta$=–0.14, SE = 0.05, $t$=–2.91, p=0.004, controlling for age, gender, and educational attainment (*Figure 4*; see also *Appendix 1—table 4*). This trend persisted when, instead, we predicted the absolute number of reminders chosen by the participant (Hypothesis H5b), $\beta$=–0.09, SE = 0.05, $t$=–1.94, p=0.053 (see *Appendix 1—table 5*), as well as when predicting the AIP (Hypothesis H5c), $\beta$=0.10, SE = 0.05, $t$=2.25, p=0.025 (see *Appendix 1—table 6*).

Previous studies have found reduced working memory in OCD (*Harkin and Kessler, 2011*), which could potentially lead to increased reminder use in compulsivity. However, the reduced reminder bias persisted if d' from the 2-back task was included as an additional covariate (Hypothesis H5d), $\beta$=–0.12, SE = 0.05, $t$=–2.57, p=0.010 (see *Appendix 1—table 7*). Finally, we predicted that our results would persist independent of whether or not the scores from the cognitive ability test were included as an additional covariate (Hypothesis H5e), which was indeed the case, $\beta$=–0.14, SE = 0.05, $t$=–2.85, p=0.005 (see *Appendix 1—table 8*). It should be noted that all our regression models included both CIT and AD as predictors to separate out the potentially competing influences of these predictors, as well as age, gender, and educational attainment as demographic covariates.

We furthermore preregistered to conduct the same tests for the AD factor but without any directional hypotheses. AD was not significantly linked to any changes in reminder bias, $\beta$=0.07, SE = 0.05, $t$=1.46, p=0.15 (see *Appendix 1—table 4*), absolute number of reminders, $\beta$=0.06, SE = 0.05, $t$=1.33, p=0.18 (see *Appendix 1—table 5*), or AIP, $\beta$=–0.08, SE = 0.05, $t$=–1.76, p=0.08, (see *Appendix 1—table 6*) controlling for age, gender, and educational attainment. This null effect did not change when working memory, $\beta$=0.06, SE = 0.05, $t$=1.23, p=0.22 (see *Appendix 1—table 7*), or scores from the cognitive ability test were included as additional covariates, $\beta$=0.07, SE = 0.05, $t$=1.41, p=0.16 (see *Appendix 1—table 8*).

Taken together, these results suggest that compulsive individuals are less biased towards offloading, in contrast to our hypothesised direction of the effect, but consistent with the observation of increased confidence in their ability on this task.

## No evidence for impaired confidence-offloading link

We predicted to find support for the *Metacognitive Control Mechanism*, meaning that CIT would act as a moderator on the link between confidence and offloading (Hypothesis H7a). In other words, we expected to find that the correlation between the metacognitive and the reminder bias to be weakened in highly compulsive individuals. However, the interaction between metacognitive bias and compulsivity in a model predicting the reminder bias was not significant, $\beta$=–0.01, SE = 0.04, $t$=–0.18, p=0.86, controlling for age, gender, and educational attainment (see *Appendix 1—table 9*). This means that in our task, confidence and offloading were linked just as much as in their low compulsive counterparts. These results remained the same even if working memory performance (d' from the 2-back task) was included as an additional covariate (Hypothesis H7b), $\beta$=–0.01, SE = 0.04, $t$=–0.26,

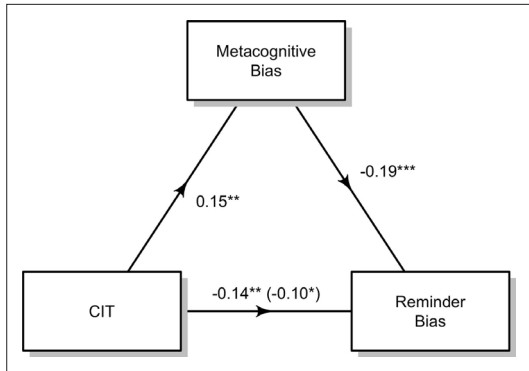

**Figure 5.** Diagram of the mediation analysis testing for the influence of the 'compulsive behaviour and intrusive thought' (CIT) factor on reminder bias, both directly and indirectly through the metacognitive bias. Standardised regression coefficients are given for each path. The value in parentheses indicates the influence of CIT on reminder bias controlling for the influence of the metacognitive bias. Asterisks indicate significance: '***': <0.001; '**': <0.01; '*': <0.05.

p=0.79 (see *Appendix 1—table 10*), or if scores from the cognitive ability test were included as an additional covariate (Hypothesis H7c), $\beta$=–0.01, SE = 0.04, $t$=–0.18, p=0.86 (see *Appendix 1— table 11*).

Contrary to our initial hypotheses, we found that increased CIT was associated with decreased rather than increased bias towards offloading. Seeing as CIT was also associated with increased confidence, and high confidence predicts low bias towards offloading, we tested whether the relationship between CIT and offloading was mediated via confidence: whilst parts of the reduction of reminders could be traced back to overconfidence, $\beta$=–0.19, SE = 0.04, $t$=–4.66, p<0.001, there was still a significant proportion of variance that was linked to compulsivity independently of this effect, $\beta$=–0.10, SE = 0.05, $t$=–2.14, p=0.032. *Figure 5* summarises this incomplete mediation effect. In previous sections, we have already reported the total effect of CIT on the reminder bias, $\beta$=–0.14, SE = 0.05, $t$=–2.91, p=0.004. Equally, we have already reported the effect of CIT on the mediator (the metacognitive bias), $\beta$=0.15, SE = 0.05, $t$=3.11, p=0.002.

We validated this outcome through an exploratory causal mediation analysis. The indirect influence of the CIT factor on reminder bias going through the metacognitive bias was calculated to be (0.15) * (–0.19)=–0.0285. To determine the significance of this influence, we implemented bootstrapping procedures. We computed unstandardised indirect effects for each of the 1000 bootstrapped samples, followed by the calculation of the 95% confidence interval, identifying the indirect effect at the 2.5th and 97.5th percentiles. The bootstrapped unstandardised indirect effect (the average causal mediation effect) computed to be –0.0256, with the 95% confidence interval ranging from –0.05 to –0.01. This indicated that the effect was statistically significant at p=0.002.

Finally, we preregistered to run the same analysis for the AD factor without hypothesising about any specific direction for any potential effects. However, we did not find evidence for a moderation effect (an interaction between AD scores and metacognitive bias when predicting the reminder bias), $\beta$=–0.04, SE = 0.04, $t$=–0.94, p=0.35, controlling for age, gender, and educational attainment (see *Appendix 1—table 12*).

In summary, whilst we found no support for the *Metacognitive Control Mechanism* (as would be reflected in a disrupted link between confidence and offloading), we did find support for both the *Metacognitive Monitoring Mechanism* (reduced pro-reminder bias as a downstream consequence of overconfidence) and the *Direct Mechanism* (independent contribution of CIT on offloading). Appendix 1 furthermore lists several additional analyses, both planned and exploratory.

## Discussion

In the current study, we explored the behavioural and cognitive correlates of two transdiagnostic traits: 'CIT' and 'AD'. We focused on changes in cognitive offloading and metacognition related to transdiagnostic compulsivity. Our results replicated that more compulsive individuals were relatively overconfident, while those who were more AD were relatively underconfident. Contrary to expectations, we observed a decreased bias towards reminders among more compulsive participants. This reduction in bias was only partially accounted for by their relative overconfidence. This partial mediation can be interpreted through both a *Metacognitive Monitoring Mechanism* (differences in the formation of the confidence signal rather than its behavioural application) and a *Direct Mechanism* (no metacognitive involvement). We found no support for a *Metacognitive Control Mechanism*, which would centre on how confidence is used to adapt behaviour (Nelson & Narens, 1990; *Boldt and Gilbert, 2022*).

## Perfectionism and the need to control as potential explanations

Contrary to our hypothesis, our study revealed an inverse relationship between transdiagnostic compulsivity and offloading: the reminder bias was reduced in more compulsive individuals. One possible interpretation is perfectionism: Some compulsive individuals may avoid using reminders altogether due to rigid, perfectionistic beliefs about needing to remember everything without relying on external aids, and using reminders could trigger their anxiety or feed into their obsessions about being forgetful or unreliable. This interpretation aligns with findings, suggesting that perfectionism serves as a transdiagnostic maintaining and risk factor for various mental health conditions, including compulsive disorders like eating disorders and OCD (*Egan et al., 2011*).

## No effect of anxiety on offloading

Interestingly, we found no significant influence of the AD transdiagnostic phenotype on offloading. This aligns with a recent study by *Kirk et al., 2021*, which also found no effect of anxiety on offloading. However, their study, which used the 'trait' component of the STAI to measure anxiety (*Spielberger et al., 1983*), found no relative underconfidence among anxious participants either. Our transdiagnostic approach likely revealed this confidence effect by separating the counteracting influences of AD and CIT factors. This distinction underscores the value of a transdiagnostic approach.

Our findings align with those reported in a recent study by *Mohr et al., 2024*. The authors observed that while high-AD participants were underconfident in a perceptual task, this underconfidence did not lead to increased information-seeking behaviour. Future research should explore whether this is due to their pessimism regarding the effectiveness of confidence-modulated strategies (i.e. setting reminders or seeking information) or whether it stems from apathy. Another possibility is that the relevant downstream effects of anxiety were not measured in our study and instead may lie in reminder-checking behaviours.

## No evidence for an impaired confidence-action link in compulsivity

Contrary to *Seow and Gillan, 2020*, and *Vaghi et al., 2017*, our study did not find the impaired confidence-action link (*Metacognitive Control Mechanism*) reported for transdiagnostic compulsivity and OCD patients. This may be because of differences between tasks – prior work used a reinforcement learning task with a clear learning element from trial to trial. Alternatively, it is possible our study was underpowered, as our sample size was designed to detect overconfidence in compulsivity, not the more nuanced but still psychometrically robust confidence-action link (*Loosen et al., 2022*), which would have required a far larger sample size. Recent studies also failed to find decreased action-confidence coupling with relatively small groups of OCD patients and controls (*Hoven et al., 2023b*; *Marzuki et al., 2022*). Indeed, both our paradigm and the earlier predictive-inference task tested for an interaction effect, which is more challenging to power adequately. Future research should consider using more direct measures that ideally aim to manipulate confidence directly.

## Implications

Participants in our current study were recruited from the general population through Prolific, meaning that the variance likely represents primarily subclinical sources. Consequently, caution should be exercised when extrapolating these results to clinical populations. For example, a recent study indicated that metacognitive impairments in OCD originate from different mechanisms than those observed in transdiagnostic compulsivity (*Hoven et al., 2023c*). Given its metacognitive impairments and the prevalent symptom of checking, OCD still remains a particularly relevant patient group for studying reminder setting, and future studies need to explore this area further. Due to their underconfidence, OCD patients might engage in more frequent reminder setting. This behaviour could serve as a compensatory mechanism, especially since OCD patients often face challenges with working memory (*Harkin and Kessler, 2011*) and prospective memory (*Harris et al., 2010*; *Racsmany et al., 2011*). However, it could also worsen their checking symptoms as more reminders mean more opportunities to check.

On the other hand, it is possible that the observed underconfidence in OCD populations may actually reflect the impact of an uncontrolled anxiety factor, effectively neutralising the influence of compulsivity on confidence. This confounding issue could explain the inconsistent findings regarding confidence bias in both compulsivity and OCD. If this was the case, then future research should

investigate which influences on confidence – the reductions caused by the AD factor or the increases caused by the CIT factor – are the driving force behind any changes in reminder setting in OCD.

A pivotal question remains: will the overall reduction in reminder setting, referred to as a 'direct effect' in this study, also be observed in OCD patients and other compulsive disorders? Such findings could support the hypothesis that an inherent aspect of compulsivity leads to the decreased use of external aids, potentially due to perfectionism or a need for control.

### Limitations

Our results are based on a well-validated paradigm which our lab has previously used in other, published studies (as reviewed in *Gilbert et al., 2023*). However, reliance on a single behavioural task also means that our results might not generalise onto cognitive offloading more broadly or even reminder setting in other contexts. As a first step, future work should aim to replicate our findings in the context of other experimental designs.

Another limitation is that in the present study, we focused solely on measuring two transdiagnostic factors: CIT and AD. We omitted the third factor, 'social withdrawal'. By doing so, we were able to reduce the number of items from 6 clinical questionnaires to 49 (*Wise and Dolan, 2020*), thereby shortening the required time for completion – an essential consideration for online research (*Sauter et al., 2020*). Nevertheless, this focused approach could introduce variability in capturing these transdiagnostic phenotypes. A recent preprint from *Hopkins et al., 2022* supports this approach. They used machine learning to select 71 items capable of reliably measuring all three factors, suggesting that future transdiagnostic studies might similarly adopt more concise item sets.

### Conclusion

With the present study, we investigated the downstream cognitive and behavioural effects of two transdiagnostic traits, CIT and AD. In particular, we were interested in the effect these factors have on metacognition and cognitive offloading, operationalised as prospective confidence and reminder setting, respectively. We replicated the finding that more compulsive individuals tend to be relatively overconfident, whereas AD individuals tend to be relatively underconfident. Contrary to our hypotheses, however, we found that compulsivity was linked reduced offloading, and that this effect was only in part explained by overconfidence.

Fulfilling delayed intentions (i.e. prospective memory) is a vital process for daily living and behavioural independence. However, this process is also highly fallible (e.g. *Crawford et al., 2003*). External memory aids are highly effective and commonplace tools that compensate for these memory failures (e.g. *Jones et al., 2021*; *Scullin et al., 2022*). Our findings suggest that compulsive individuals are at particular risk of inadequate external memory support and would potentially benefit from interventions that target cognitive offloading strategies.

## Materials and methods

**Key resources table**

| Reagent type (species) or resource | Designation | Source or reference | Identifiers | Additional information |
|---|---|---|---|---|
| Software, algorithm | R | *R Development Core Team, 2024* | 4.4.2; RRID:SCR_001905 | |
| Software, algorithm | RStudio | *RStudio Team, 2020* | 2024.09.1+394; RRID:SCR_000432 | |
| Software, algorithm | diagram | *Soetaert, 2020* | 1.6.5; RRID:SCR_026982 | R package |
| Software, algorithm | effectsize | *Ben-Shachar et al., 2020* | 0.8.9; RRID:SCR_026983 | R package |
| Software, algorithm | lmerTest | *Kuznetsova et al., 2017* | 3.1-3; RRID:SCR_015656 | R package |
| Software, algorithm | lme4 | *Bates et al., 2015* | 1.1-35.5; RRID:SCR_015654 | R package |
| Software, algorithm | mediation | *Tingley et al., 2014* | 4.5.0; RRID:SCR_026984 | R package |
| Software, algorithm | plyr | *Wickham, 2011* | 1.8.9; RRID:SCR_026985 | R package |
| Software, algorithm | pwr | *Champely, 2020* | 1.3-0; RRID:SCR_025480 | R package |
| Software, algorithm | quickpsy | *Linares and López-Moliner, 2016* | 0.1.5.1; RRID:SCR_026986 | R package |

## Task and procedure

For the present, preregistered study, we used a novel variant of an online cognitive-offloading task ('optimal reminders task'; cf. *Gilbert et al., 2020*). This task allowed us to measure how people set reminders in relation to their confidence. All procedures, hypotheses, and planned analyses were preregistered at https://osf.io/kztf8 prior to the commencement of data collection.

On every trial, participants were instructed to move several numbered, yellow circles to the bottom of a square in consecutive order (see *Figure 6A*). Whenever a circle was removed, a new one appeared up to a total of 15 circles. The source of difficulty of this task stems from the 'special' circles, which constitute the delayed intentions people have to fulfil. These circles flashed in a colour (blue, orange, or magenta) when they first appeared on screen before fading to yellow. Participants' task was to drag these circles to their colour-corresponding side once the time had come to remove the respective special circle (top, left, or right). There were six special circles per trial. On some trials, participants had to rely on their own memory to complete the task and remember the target locations of the special circles. On other trials, they set spatial reminders, indicating the locations to which the special circles must be moved to. More specifically, they were taught to move the special circle next to the border through which it would have to be moved out of the square later.

Every trial began with a decision: participants could choose to do the task without reminders and earn 10 points for every special circle they remembered to move to the correct border, or they could choose to use reminders but earn less for each special circle (*Figure 6B*). Critically, this lesser amount was varied between 2 and 9 points, allowing us to calculate the participants' indifference point when trading off the benefit of reminders with their reduced reward. This AIP could then be contrasted against their OIP, calculated from participants' accuracy with or without reminders, see below for further details. Since our task included only 4 trials each with or without reminders, we counterbalanced the assignment of odd or even target values to these conditions.

Together, there were three key conditions in our task presented intermixed throughout the experiment: the Forced Internal condition (FI; 4 trials) in which participants had to remember the circles unaided, the Forced External condition (FE; 4 trials) in which they had to use reminders, and the Choice Only controlling for age, gender, and educational attainment (CO; 8 trials) in which they were free to choose whichever strategy they preferred but the trial ended after only six circles and without any special circles. To give participants the impression of maximum agency over the task, we only told them that their choice would be overwritten whenever there was a mismatch with the pseudo-randomly assigned condition (25.3% of all trials; SD = 5.8; see *Figure 6C*). This way, participants were unable to tell which condition they were currently in and whether it would be a partial trial. Participants used reminders on average on 49.9% of trials (SD = 16.5).

Participants were asked to rate their confidence once during the experiment, being asked to indicate the 'percentage of the special circles [they] can correctly drag to the instructed side of the square' (*Figure 6D*). Importantly, this confidence judgement was given after the first practice trials and before the offloading strategy was introduced to ensure participants answered this question with regard to their own perceived memory capabilities. Average confidence was 55.6% (proportion of trials on which participants predicted to remember to move the special circles; SD = 24.2).

In addition to the reminder task, we included items from six individual differences questionnaires shortened to include only the items required to reliably measure the CIT and AD factors (*Wise and Dolan, 2020*). These questionnaires were presented in random order: 4 items from the Apathy Evaluation Scale (AES; *Marin et al., 1991*), 8 items from the Zung Depression Scale (SDS; *Zung, 1965*), 4 items from the Eating Attitudes Test (EAT-26; *Garner et al., 1982*), 12 items from the Barratt Impulsiveness Scale (BIS-11; *Patton et al., 1995*), 11 items from the Obsessive Compulsive Inventory – Revised (OCIR; *Foa et al., 2002*), and 11 items from the 'trait' part of the State-Trait Anxiety Inventory (STAI; *Spielberger et al., 1983*). A list of all included items can be found in Appendix 1.

We also included a catch item in the BIS-11 ('I competed in the 1917 Summer Olympics Games.') to ensure participants were paying attention to the task, as well as three covariates aimed at measuring cognitive ability (a 5-item version of the International Cognitive Ability Resource; ICAR5; *Kirkegaard and Bjerrekær, 2016*; *Condon and Revelle, 2014*); educational attainment mapped onto a 1–9 scale and based on the ISCED 2011 categories (see Appendix 1); and working memory, assessed using 100 consecutive letters from the 2-back task (e.g. *Kirchner, 1958*). The logic behind including the latter covariate was that whilst our key dependent variables already corrected for working memory

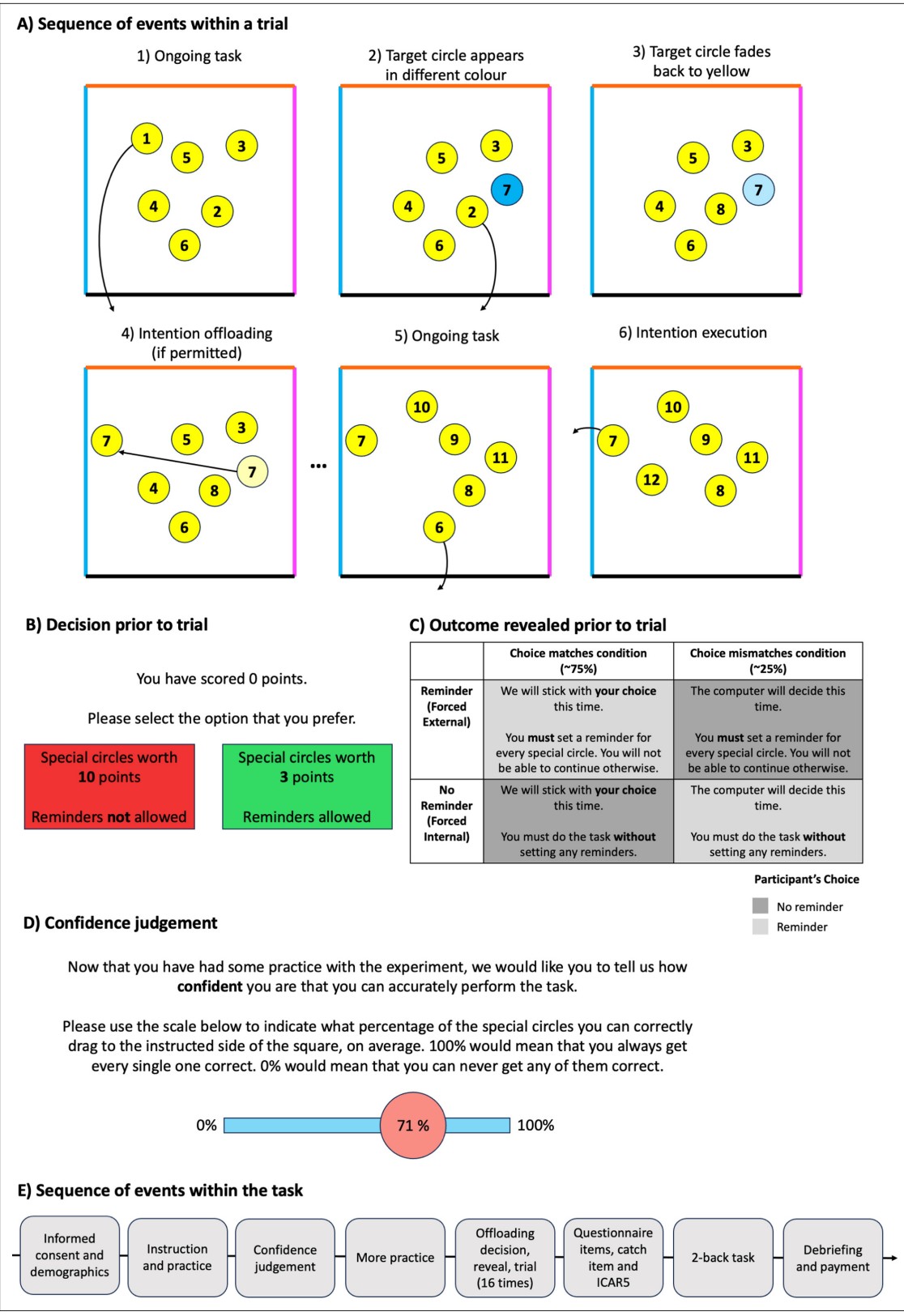

**Figure 6.** Overview of the intention offloading paradigm. (**A**) Example sequence of events within a single trial. Trajectories of movement made by a fictive participant are shown as black arrows. The blue coloured circle corresponds to the left boundary of the square and indicates that this circle must be moved to this side rather than the bottom. (**B**) Example of an offloading decision which participants were required to make before each trial. (**C**) After each decision, they were informed whether or not they would perform the upcoming trial with reminders. The cell's shading indicates the participant's

*Figure 6 continued on next page*

*Figure 6 continued*

original choice. (**D**) Confidence was rated once before the introduction of the offloading strategy on a scale ranging from 0% to 100%. (**E**) Sequence of events within the task. All aspects of the task were performed online in the web browser.

(more specifically: unaided prospective memory performance), this could tap into additional working memory components not measured already and potentially impacted in compulsivity based on the finding that they have often been found to be impaired in OCD (*Harkin and Kessler, 2011*). Together, these elements resulted in a total duration of approximately 35 min. The sequence of events within the task is shown in *Figure 6E*.

## Participants

Ethical approval for this study was received from the local Ethics Committee at University College London (UCL) under the reference number 1584/003. Informed consent was obtained from all participants prior to the study. Participants were invited on prolific.co to participate for £3.90. Based on points won during the main task, the upper 50% of participants were furthermore rewarded with a bonus payment of £1. We restricted our search to the Prolific standard sample, allowed participants from all countries, with a minimum of 18 years. All participants had to be fluent in English and were required to have an approval rate of over 90% based on Prolific's criteria. Moreover, we required participants to not have participated in one of the four pilots prior to this study.

All analyses were conducted with R in RStudio (*R Development Core Team, 2024*; *RStudio Team, 2020*) together with the following R packages: effectsize (*Ben-Shachar et al., 2020*), lmerTest (*Kuznetsova et al., 2017*), lme4 (*Bates et al., 2015*), mediation (*Tingley et al., 2014*), plyr (*Wickham, 2011*), pwr (*Champely, 2020*), and quickpsy (*Linares and López-Moliner, 2016*). We calculated our sample size based on the link between confidence and transdiagnostic compulsivity as reported in two recent studies (*Rouault et al., 2018*; *Seow and Gillan, 2020*). To be able to detect a link between these variables of $\beta$=0.23, p<0.001, as in *Rouault et al., 2018*, we required *N*=288 participants (two-sided testing, power = 0.8, CL = 0.95). To be able to detect a link of $\beta$=6.74, p<0.001, as in *Seow and Gillan, 2020*, we required *N*=291 participants (two-sided testing, power = 0.8, CL = 0.95). In both cases, the power calculation was based on a partial regression approach, excluding the effect in question from the model and comparing the explained variance compared to the full model. Since we are furthermore aiming to test a moderation effect of compulsivity on the link between the metacognitive bias and the reminder bias, we decided to collect a larger sample of *N*=600 after exclusions.

We preregistered six exclusion criteria, based on which we excluded and replaced 69 participants: Nine participants were excluded due to a higher hit rate on forced internal than forced external trials, 22 participants were excluded due to less than 70% accuracy on FE trials, and 3 participants due to less than 10% accuracy on FI trials. We furthermore preregistered to exclude participants with a negative correlation between value and reminder choice (1=reminder, 0=no reminder), as this would indicate participants did not understand the instructions: in order to maximise points in our task, participants should preferentially choose reminders when this strategy brings a higher number of points. Based on this, we excluded 40 participants. No participants were excluded based on scoring lower or higher three times the median absolute deviation calculated separately based on both the reminder bias and the metacognitive bias. Finally, we excluded 9 participants because they failed to answer *with* 'Do not agree at all' to the catch item. *Figure 1* visualises the exclusions shown in red. In total, we excluded 10.3% of all participants. There were an additional 26 participants excluded for technical reasons, raising the exclusion rate to 13.7%.

## Key dependent variables

Our task allowed us to calculate several dependant variables relevant in the context of our study question. The first is the *OIP*, the optimal indifference point. The OIP describes the number of points at which an unbiased, reward-maximising participant is indifferent between the two strategies (reminders or no reminders) and is calculated as:

$$OIP = \left( 10 * ACC_{FI} \right) / ACC_{FE}$$

where $ACC_{FE}$ is the accuracy measured during trials in which the participants had to solve the task using reminders (FE condition), and $ACC_{FI}$ is the accuracy measured during trials in which participants had to solve the task without reminders (FI condition). In contrast, the *AIP* is the the AIP is the actual indifference point, which is the point cut-off at which participants actually were indifferent and is operationalised as the threshold parameter from fitting a psychometric function to the choice data (target values predicting the decision whether or not to use reminders). Fitting was done using the *quickpsy* package in *R,* and more detail is given in Appendix 1. It should be noted that the OIP has a slightly finer resolution due to the number of special circles per trial.

Setting the OIP and the AIP in relation, we can calculate the *reminder bias*, reflecting participants' tendency to use reminders corrected for their actual performance and calculated as the difference between both indifference points:

$$bias_{rem} = OIP - AIP$$

Positive values reflect that people set more reminders relative to the optimal strategy. The fourth measure is the *metacognitive bias*, reflecting participants' over- or underconfidence relative to their performance and was calculated as:

$$bias_{meta} = confidence - ACC_{FI}$$

Negative values can be interpreted as underconfidence.

Crucially, our study relies on the key assumption that the metacognitive bias can predict the reminder bias, but $ACC_{FI}$ contributes to both biases. To avoid circularity, we therefore split the accuracy data to avoid potentially inflating the correlation. More specifically, we included only the even trials to calculate the $ACC_{FI}$ for the OIP, whereas we included only the odd trials to calculate the $ACC_{FI}$ for the metacognitive bias. All available trials from the FE condition were used to calculate the OIP.

It should be noted that we had incorrectly stated in the preregistration that accuracy from forced external trials would contribute to the calculation of the metacognitive bias. However, the metacognitive bias is a judgement given about the unaided memory performance, in fact confidence is measured before participants were even introduced to the offloading strategy (see above). We therefore used only the internal trials in calculating the metacognitive bias.

Finally, the transdiagnostic scores for the 'CIT' factor and the 'AD' factor were calculated from participants' ratings to the individual differences questionnaires by multiplying them with the item weights from *Wise and Dolan, 2020*, prior to summing them. The items composing the CIT and AD scores, respectively, were non-overlapping with 24 items forming the AD score and 25 items forming the CIT score.

## Preregistered hypotheses and statistical analyses

We preregistered eight hypotheses (see *Table 1*), half of which were sanity checks (H1-H4) aimed to establish whether our task would generally lead to the same patterns as previous studies using a similar task (as reviewed in *Gilbert et al., 2023*). H1 was a replication of the central finding of the link between confidence and offloading. More specifically, we entered the unconfounded metacognitive bias and reminder bias into a Pearson correlation analysis. We expected to find a negative relationship between the two measures, which we planned to test for significance using a one-sided test. We furthermore expected to find that people would use more reminders than optimal. This pro-reminder bias would be reflected in a positive reminder bias (H2). We planned to test this using a one-sided paired t-test. Relatedly, we expected to find people to be generally underconfident (i.e. expecting to remember fewer special circles than they actually did when doing the task without reminders). Such underconfidence would be reflected in a negative metacognitive bias (H3), which we again planned to test using a one-sided paired t-test. Furthermore, we expected that those who required more reminders would also be the ones to use them more, as reflected in a positive correlation between the AIP and OIP, again as a one-sided test (H4). We decided to use Spearman's rho due to the data most likely being distributed around the extremes of the scale. For H2-H4 (as well as H5, H6, and H8, see below), we used the biases and indifference points calculated from all available trials as there was no circularity issue.

Hypotheses H5-H8 were the key hypotheses of our study. Here, we address them out of order in the interest of an improved logical flow. Hypothesis H6 predicted that more compulsive individuals

**Table 1.** List of preregistered hypotheses together with the empirical support our study found.

White background indicates sanity check hypotheses, and grey background indicates key hypotheses. OIP = optimal indifference point. AIP = actual indifference point. CIT = compulsive behaviour and intrusive thought.

| Number | Hypothesis | Support? |
|---|---|---|
| H1 | The reminder bias and metacognitive bias are negatively correlated. | Yes |
| H2 | Participants use reminders excessively. | Yes |
| H3 | Participants are underconfident in their own memory. | Yes |
| H4 | OIP and AIP are positively correlated. | Yes |
| H5a | Positive link between CIT and reminder bias. | No (significant negative effect) |
| H5b | Positive link between CIT and absolute number of reminders chosen. | No (negative effect but significance not reached) |
| H5c | Positive link between CIT and AIP. | No (significant negative effect) |
| H5d | Positive link between CIT and reminder bias even if working memory is included as a covariate. | No (significant negative effect) |
| H5e | Positive link between CIT and reminder bias even if cognitive ability is included as a covariate. | No (significant negative effect) |
| H6a | A significant link exists between CIT and metacognitive bias (preregistered as a two-sided test, so either more or less confident). | Yes (positive) |
| H6b | A significant link exists between CIT and raw confidence. | Yes (positive) |
| H6c | A significant link exists between CIT and metacognitive bias even if cognitive ability is included as a covariate. | Yes (positive) |
| H7a | CIT acts as a moderator on the link between confidence and offloading. In other words, we expect to find that the correlation between the metacognitive and the reminder bias to be weakened in highly compulsive individuals. | No |
| H7b | CIT acts as a moderator on the link between confidence and offloading even if working memory is included as a covariate. | No |
| H7c | CIT acts as a moderator on the link between confidence and offloading even if cognitive ability is included as a covariate. | No |
| H8a | A significant negative link exists between AD and metacognitive bias (i.e. more anxious-depressed individuals tend to be underconfident). | Yes |
| H8b | A significant negative link exists between AD and raw confidence. | Yes |
| H8c | A significant negative link exists between AD and metacognitive bias even if cognitive ability is included as a covariate. | Yes |

would show an effect in confidence bias, reflected in a reliable predictor of the CITs scores on the metacognitive bias from the following regression model:

$$bias_{meta} \sim CIT + AD + age + gender + education + \varepsilon$$

Though we did not preregister a direction for this effect, in the light of recent findings, it has now become clear that compulsivity would most likely be linked to overconfidence (*Rouault et al., 2018*; *Seow and Gillan, 2020*; *Benwell et al., 2022*; *Fox et al., 2023*; *Fox et al., 2024*; *Hoven et al., 2023a*). The same model was used to test hypothesis H8, predicting that more AD individuals tend to be underconfident. This would be reflected in AD scores being negatively linked to the metacognitive bias. The model above represents the main models designed to test hypotheses H6a and H8a. We furthermore also tested these hypotheses but predicted raw confidence (percentage of circles participants predicted they would remember; H6b and H8b, respectively), as well as extending the main model with the scores from the cognitive ability test (ICAR5) as an additional covariate (H6c and H8c, respectively). For this, as well as all following regression models, we *z*-transformed all non-binary variables prior to fitting the models.

With H5, we predicted that more compulsive individuals would show a bias towards more offloading, reflected in a positive regression coefficient when using the CIT score as a predictor of the reminder bias. This hypothesis was not a replication; consequently, we decided to carry out the test two-sided.

Throughout this section, whenever not explicit specified, we plan to carry out a test two-sided. Due to the diametrically opposing effects of CIT and AD, both transdiagnostic scores need to be entered into the model, alongside our demographic covariates age, gender, and educational attainment:

$$bias_{rem} \sim CIT + AD + age + gender + education + \varepsilon$$

We fitted several different versions of this model: the main model predicted the reminder bias (H5a), but we also fit one with the absolute number of reminders chosen (H5b) or the AIP (H5c). To understand whether any differences in offloading behaviour could stem from differences in working memory capacity not already captured by our correction for unaided task performance, we furthermore extended the main model by also including the d' from a 2-back task as a covariate (H5d). Finally, we fit an extended version of the main model with scores from the cognitive ability test (ICAR5) as an additional covariate to capture cognitive ability (H5e). We ran the same analysis but for the AD factor. We included this test as a preregistered analysis but did not specify any directional hypotheses.

Our final hypothesis, H7, aimed to differentiate between the *Metacognitive Monitoring Mechanism*, the *Metacognitive Control Mechanism,* and the *Direct Mechanism*. We tested how compulsivity would affect the relationship between confidence and offloading. More specifically, we predicted that CIT scores would act as a moderator variable between the metacognitive and the reminder bias, and that highly compulsive individuals would have a weaker link. We tested this by fitting the following regression model to the data:

$$bias_{rem} \sim bias_{meta} * CIT + AD + age + gender + education + \varepsilon$$

To avoid circularity, we used the unconfounded metacognitive bias and reminder bias for this analysis. The moderation of CIT is reflected in its interaction term with the $bias_{meta}$ predictor. A significant interaction term can be interpreted as support for the *Metacognitive Control Mechanism*. In addition to this main model (H7a), we furthermore also tested whether this effect would persist if working memory (2-back d'; H7b) or educational attainment (H7c) were included as additional covariates. We ran the same analysis but for the AD factor. We included this test as a preregistered analysis but did not specify any directional hypotheses.

It should be noted that whilst not explicitly preregistered, our planned models also allow testing for a mediation effect (metacognitive bias acting as a mediator on the effect of the CIT score on the reminder bias). This is done by comparing the effect of CIT on the reminder bias when the effect of the metacognitive bias is accounted for (Hypothesis 7) to when it is not (Hypothesis 5). In addition, we included a causal mediation analysis (not preregistered) using the *mediation* package in R. This analysis involved testing of the indirect effect using bootstrapping. More specifically, we computed unstandardised indirect effects for each of our 1000 bootstrapped samples and based on those the 95% confidence interval. To keep the information entering into the mediation analysis constant, we re-fitted the models from our sections on H5 and H6/H8 but with the unconfounded metacognitive bias and reminder bias, respectively. Furthermore, we had to treat the covariate 'gender' as a continuous variable as the *mediation* package would otherwise not have been able to fit the data. We expect that this difference is unlikely to cause any issues with the interpretation of our effects. A significant mediation effect can be interpreted as support for the *Metacognitive Monitoring Mechanism*. A significant direct effect can be interpreted as support for the *Direct Mechanism*.

## Acknowledgements

The Wellcome Trust, 206480/Z/17/Z, Annika Boldt. Research Ireland's Frontiers, 19/FFP/6418, Claire Gillan. European Research Council (ERC), ERC-H2020-HABIT, Claire Gillan.

## Additional information

### Funding

| Funder | Grant reference number | Author |
| --- | --- | --- |
| Wellcome Trust | 10.35802/206480 | Annika Boldt |

| Funder | Grant reference number | Author |
|---|---|---|
| Ireland's Research Frontiers | 19/FFP/6418 | Claire M Gillan |
| European Research Council | ERC-H2020-HABIT | Claire M Gillan |

The funders had no role in study design, data collection and interpretation, or the decision to submit the work for publication. For the purpose of Open Access, the authors have applied a CC BY public copyright license to any Author Accepted Manuscript version arising from this submission.

### Author contributions

Annika Boldt, Conceptualization, Resources, Data curation, Software, Formal analysis, Funding acquisition, Validation, Investigation, Visualization, Methodology, Writing – original draft, Project administration, Writing – review and editing; Celine Ann Fox, Claire M Gillan, Conceptualization, Resources, Methodology, Writing – review and editing; Sam Gilbert, Conceptualization, Resources, Software, Supervision, Funding acquisition, Methodology, Writing – review and editing

### Author ORCIDs

Annika Boldt https://orcid.org/0000-0002-6913-5099
Celine Ann Fox https://orcid.org/0000-0003-1740-3765
Claire M Gillan https://orcid.org/0000-0001-9065-403X
Sam Gilbert https://orcid.org/0000-0002-3839-7045

### Ethics

Both informed consent and consent to publish were obtained. Ethical approval for this study was received from the local Ethics Committee at University College London (UCL) under the reference number 1584/003.

Reviewer #1 (Public review): https://doi.org/10.7554/eLife.98114.4.sa1
Author response https://doi.org/10.7554/eLife.98114.4.sa2

## Additional files

### Supplementary files

Source data 1. Contains all data needed to reproduce the analyses.

Source data 2. Contains item weights from *Wise and Dolan, 2020* available for download at https://osf.io/q3a6v.

Source code 1. Contains an RMarkdown document which includes code to run all analyses and reproduce all figures.

Source code 2. Contains the output from the RMarkdown script.

MDAR checklist

### Data availability

All data and analysis scripts are available for download at https://osf.io/b9rxz/.

The following dataset was generated:

| Author(s) | Year | Dataset title | Dataset URL | Database and Identifier |
|---|---|---|---|---|
| Boldt A, Fox CA, Gillan C, Gilbert S | 2025 | Compulsivity, confidence and reminder setting | https://osf.io/b9rxz/ | Open Science Framework, b9rxz |

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

## Appendix 1

**Appendix 1—table 1.** Predicting metacognitive bias.
All continuous variables are *z*-transformed. SE = standard error; m=male; f=female; o=other.

|  | β | SE | t | p |
|---|---|---|---|---|
| Intercept | 0.07 | 0.05 | 1.30 | 0.193 |
| AD | −0.23 | 0.05 | −4.99 | <0.001 |
| CIT | 0.15 | 0.05 | 3.11 | 0.002 |
| Age | −0.02 | 0.04 | −0.55 | 0.586 |
| gender1 (m vs. f) | −0.17 | 0.08 | −2.06 | 0.040 |
| gender2 (m vs. o) | −0.29 | 0.38 | −0.78 | 0.438 |
| education | −0.0001 | 0.04 | −0.005 | 0.996 |

**Appendix 1—table 2.** Predicting confidence.
All continuous variables are *z*-transformed. SE = standard error; m=male; f=female; o=other.

|  | β | SE | t | p |
|---|---|---|---|---|
| Intercept | 0.09 | 0.05 | 1.81 | 0.070 |
| AD | −0.29 | 0.04 | −6.43 | <0.001 |
| CIT | 0.12 | 0.05 | 2.76 | 0.006 |
| Age | −0.14 | 0.04 | −3.44 | <0.001 |
| gender1 (m vs. f) | −0.24 | 0.08 | −2.93 | 0.004 |
| gender2 (m vs. o) | −0.23 | 0.37 | −0.63 | 0.528 |
| education | 0.04 | 0.04 | 1.01 | 0.311 |

**Appendix 1—table 3.** Predicting metacognitive bias with ICAR5 scores as an additional covariate.
All continuous variables are *z*-transformed. SE = standard error; m=male; f=female; o=other.

|  | β | SE | t | p |
|---|---|---|---|---|
| Intercept | 0.08 | 0.05 | 1.68 | 0.094 |
| AD | −0.20 | 0.05 | −4.46 | <0.001 |
| CIT | 0.12 | 0.05 | 2.57 | 0.011 |
| Age | −0.03 | 0.04 | −0.66 | 0.507 |
| gender1 (m vs. f) | −0.22 | 0.08 | −2.61 | 0.009 |
| gender2 (m vs. o) | −0.45 | 0.37 | −1.21 | 0.226 |
| education | 0.04 | 0.04 | 0.91 | 0.364 |
| ICAR5 | −0.20 | 0.04 | −4.84 | <0.001 |

**Appendix 1—table 4.** Predicting reminder bias.
All continuous variables are *z*-transformed. SE = standard error; m=male; f=female; o=other.

|  | β | SE | t | p |
|---|---|---|---|---|
| Intercept | −0.01 | 0.05 | −0.24 | 0.813 |
| AD | 0.07 | 0.05 | 1.46 | 0.146 |
| CIT | −0.14 | 0.05 | −2.91 | 0.004 |
| Age | 0.07 | 0.04 | 1.69 | 0.092 |

*Appendix 1—table 4 Continued on next page*

*Appendix 1—table 4 Continued*

| | β | SE | t | p |
|---|---|---|---|---|
| gender1 (m vs. f) | 0.005 | 0.08 | 0.06 | 0.955 |
| gender2 (m vs. o) | 0.88 | 0.38 | 2.32 | 0.021 |
| education | −0.06 | 0.04 | −1.42 | 0.157 |

**Appendix 1—table 5.** Predicting absolute number of reminders.
All continuous variables are *z*-transformed. SE = standard error; m=male; f=female; o=other.

| | β | SE | t | p |
|---|---|---|---|---|
| Intercept | −0.03 | 0.05 | −0.68 | 0.496 |
| AD | 0.06 | 0.05 | 1.33 | 0.183 |
| CIT | −0.09 | 0.05 | −1.94 | 0.053 |
| Age | 0.18 | 0.04 | 4.38 | <0.001 |
| gender1 (m vs. f) | 0.07 | 0.08 | 0.86 | 0.393 |
| gender2 (m vs. o) | 0.73 | 0.38 | 1.93 | 0.054 |
| education | −0.10 | 0.04 | −2.58 | 0.010 |

**Appendix 1—table 6.** Predicting actual indifference point (AIP).
All continuous variables are *z*-transformed. SE = standard error; m=male; f=female; o=other.

| | β | SE | t | p |
|---|---|---|---|---|
| Intercept | 0.02 | 0.05 | 0.45 | 0.657 |
| AD | −0.08 | 0.05 | −1.76 | 0.079 |
| CIT | 0.10 | 0.05 | 2.25 | 0.025 |
| Age | −0.17 | 0.04 | −3.95 | <0.001 |
| gender1 (m vs. f) | −0.04 | 0.08 | −0.45 | 0.652 |
| gender2 (m vs. o) | −0.75 | 0.38 | −1.99 | 0.047 |
| education | 0.09 | 0.04 | 2.24 | 0.025 |

**Appendix 1—table 7.** Predicting reminder bias with 2-back d' as an additional covariate.
All continuous variables are *z*-transformed. SE = standard error; m=male; f=female; o=other.

| | β | SE | t | p |
|---|---|---|---|---|
| Intercept | −0.01 | 0.05 | −0.23 | 0.821 |
| AD | 0.06 | 0.05 | 1.23 | 0.219 |
| CIT | −0.12 | 0.05 | −2.57 | 0.010 |
| Age | 0.07 | 0.04 | 1.78 | 0.076 |
| gender1 (m vs. f) | 0.004 | 0.08 | 0.05 | 0.961 |
| gender2 (m vs. o) | 0.86 | 0.38 | 2.25 | 0.025 |
| education | −0.06 | 0.04 | −1.56 | 0.120 |
| 2-back d' | 0.10 | 0.04 | 2.41 | 0.016 |

**Appendix 1—table 8.** Predicting reminder bias with ICAR5 scores as an additional covariate.
All continuous variables are *z*-transformed. SE = standard error; m=male; f=female; o=other.

| | β | SE | t | p |
|---|---|---|---|---|
| Intercept | −0.01 | 0.05 | −0.26 | 0.796 |

*Appendix 1—table 8 Continued*

|  | β | SE | t | p |
|---|---|---|---|---|
| AD | 0.07 | 0.05 | 1.41 | 0.160 |
| CIT | −0.14 | 0.05 | −2.85 | 0.005 |
| Age | 0.07 | 0.04 | 1.70 | 0.091 |
| gender1 (m vs. f) | 0.01 | 0.08 | 0.09 | 0.927 |
| gender2 (m vs. o) | 0.90 | 0.38 | 2.33 | 0.020 |
| education | −0.06 | 0.04 | −1.45 | 0.147 |
| 2-back d' | 0.01 | 0.04 | 0.32 | 0.751 |

**Appendix 1—table 9.** Predicting reminder bias with metacognitive bias as an additional covariate (i.e. testing for a moderation effect).

All continuous variables are *z*-transformed. SE = standard error; m=male; f=female; o=other; MetaBias = metacognitive bias.

|  | β | SE | t | p |
|---|---|---|---|---|
| Intercept | −0.01 | 0.05 | −0.25 | 0.802 |
| Metacognitive bias | −0.19 | 0.04 | −4.66 | <0.001 |
| AD | 0.03 | 0.05 | 0.67 | 0.506 |
| CIT | −0.10 | 0.05 | −2.14 | 0.032 |
| Age | 0.13 | 0.04 | 3.22 | 0.001 |
| gender1 (m vs. f) | 0.003 | 0.08 | 0.04 | 0.969 |
| gender2 (m vs. o) | 0.99 | 0.37 | 2.65 | 0.008 |
| education | −0.08 | 0.04 | −2.03 | 0.043 |
| CIT X MetaBias | −0.01 | 0.04 | −0.18 | 0.857 |

**Appendix 1—table 10.** Predicting reminder bias with metacognitive bias and 2-back d' as additional covariates.

All continuous variables are *z*-transformed. SE = standard error; m=male; f=female; o=other; MetaBias = metacognitive bias.

|  | β | SE | t | p |
|---|---|---|---|---|
| Intercept | −0.01 | 0.05 | −0.26 | 0.797 |
| Metacognitive Bias | −0.17 | 0.04 | −4.27 | <0.001 |
| AD | 0.03 | 0.05 | 0.55 | 0.584 |
| CIT | −0.09 | 0.05 | −1.90 | 0.058 |
| Age | 0.14 | 0.04 | 3.28 | 0.001 |
| gender1 (m vs. f) | 0.004 | 0.08 | 0.06 | 0.953 |
| gender2 (m vs. o) | 0.97 | 0.37 | 2.60 | 0.010 |
| education | −0.09 | 0.04 | −2.13 | 0.034 |
| 2-back d' | 0.08 | 0.04 | 1.95 | 0.052 |
| CIT X MetaBias | −0.01 | 0.04 | −0.26 | 0.793 |

**Appendix 1—table 11.** Predicting reminder bias with metacognitive bias and ICAR5 scores as additional covariates.

All continuous variables are *z*-transformed. SE = standard error; m=male; f=female; o=other; MetaBias = metacognitive bias.

|  | β | SE | t | p |
|---|---|---|---|---|
| Intercept | −0.01 | 0.05 | −0.27 | 0.789 |
| Metacognitive bias | −0.19 | 0.04 | −4.57 | <0.001 |
| AD | 0.03 | 0.05 | 0.64 | 0.521 |
| CIT | −0.10 | 0.05 | −2.11 | 0.035 |
| Age | 0.13 | 0.04 | 3.22 | 0.001 |
| gender1 (m vs. f) | 0.005 | 0.08 | 0.06 | 0.949 |
| gender2 (m vs. o) | 1.00 | 0.37 | 2.66 | 0.008 |
| education | −0.08 | 0.04 | −2.03 | 0.043 |
| ICAR5 | 0.009 | 0.04 | 0.22 | 0.829 |
| CIT X MetaBias | −0.01 | 0.04 | −0.18 | 0.859 |

**Appendix 1—table 12.** Predicting reminder bias with metacognitive bias and ICAR5 scores as additional covariates.
All continuous variables are *z*-transformed. SE = standard error; m=male; f=female; o=other; MetaBias = metacognitive bias.

|  | β | SE | t | p |
|---|---|---|---|---|
| Intercept | −0.02 | 0.05 | −0.37 | 0.712 |
| Metacognitive bias | −0.19 | 0.04 | −4.65 | <0.001 |
| AD | 0.03 | 0.05 | 0.67 | 0.501 |
| CIT | −0.10 | 0.05 | −2.15 | 0.032 |
| Age | 0.13 | 0.04 | 3.15 | 0.002 |
| gender1 (m vs. f) | 0.004 | 0.08 | 0.04 | 0.966 |
| gender2 (m vs. o) | 0.99 | 0.37 | 2.65 | 0.008 |
| education | −0.08 | 0.04 | −1.91 | 0.057 |
| AD X MetaBias | −0.04 | 0.04 | −0.94 | 0.349 |

## Questionnaire items
### Apathy Evaluation Scale
Response scores:

> Not at all characteristic (1)
> Slightly characteristic (2)
> Somewhat characteristic (3)
> Very characteristic (4)

> ***AES_2.*** I get things done during the day. (Reverse)
> ***AES_7.*** I approach life with intensity. (Reverse)
> ***AES_17.*** I have initiative. (Reverse)
> ***AES_18.*** I have motivation. (Reverse)

### Barrett's Impulsivity Scale
Response scores:

> Rarely/never (1)
> Occasionally (2)
> Often (3)
> Almost always/Always (4)

Items:

> **BIS_1.** I plan tasks carefully. (Reverse)
> **BIS_6.** I have 'racing' thoughts.
> **BIS_9.** I concentrate easily. (Reverse)
> **BIS_13.** I plan for job security. (Reverse)
> **BIS_14.** I say things without thinking.
> **BIS_15.** I like to think about complex problems. (Reverse)
> **BIS_17.** I act 'on impulse'.
> **BIS_20.** I am a steady thinker. (Reverse)
> **BIS_22.** I buy things on impulse.
> **BIS_25.** I spend or charge more than I earn.
> **BIS_26.** I often have extraneous thoughts when thinking.
> **BIS_check.** I competed in the 1917 Summer Olympics Games.

## Eating Attitudes Test

Response scores:

> Always (3)
> Usually (2)
> Often (1)
> Sometimes (0)
> Rarely (0)
> Never (0)

*Items:*

> **EAT_1.** I am terrified about being overweight.
> **EAT_11.** I am preoccupied with a desire to be thinner.
> **EAT_12.** I think about burning up calories when I exercise.
> **EAT_14.** I am preoccupied with the thought of having fat on my body.

## Obsessive Compulsive Inventory

Response scores:
Not at all (0)
A little (1)
Moderately (2)
A lot (3)
Extremely (4)
Items:

> **OCI_1.** I have saved up so many things that they get in the way.
> **OCI_2.** I check things more often than necessary.
> **OCI_4.** I feel compelled to count while I am doing things.
> **OCI_6.** I find it difficult to control my own thoughts.
> **OCI_7.** I collect things I don't need.
> **OCI_9.** I get upset if others change the way I have arranged things.
> **OCI_11.** I sometimes have to wash or clean myself simply because I feel contaminated.
> **OCI_12.** I am upset by unpleasant thoughts that come into my mind against my will.
> **OCI_13.** I avoid throwing things away because I am afraid I might need them later.
> **OCI_16.** I feel that there are good and bad numbers.
> **OCI_18.** I frequently get nasty thoughts and have difficulty in getting rid of them.

## Self-rating Depression Scale (SDS)

Response scores:

> A little of the time (1)

Some of the time (2)
Good part of the time (3)
Most of the time (4)

Items:

**SDS_11.** My mind is as clear as it used to be. (Reverse)
**SDS_12.** I find it easy to do the things I used to. (Reverse)
**SDS_13.** I am restless and can't keep still.
**SDS_14.** I feel hopeful about the future. (Reverse)
**SDS_16.** I find it easy to make decisions. (Reverse)
**SDS_17.** I feel that I am useful and needed. (Reverse)
**SDS_18.** My life is pretty full. (Reverse)
**SDS_20.** I still enjoy the things I used to do. (Reverse)

## State Trait Anxiety Inventory

Response scores:

Almost never (1)
Sometimes (2)
Often (3)
Almost always (4)
Items:

**STAI_1.** I feel pleasant. (Reverse)
**STAI_3.** I feel satisfied with myself. (Reverse)
**STAI_5.** I feel like a failure.
**STAI_8.** I feel that difficulties are piling up so that I cannot overcome them.
**STAI_9.** I worry too much over something that really doesn't matter.
**STAI_10.** I am happy. (Reverse)
**STAI_12.** I lack self-confidence.
**STAI_13.** I feel secure. (Reverse)
**STAI_16.** I am content. (Reverse)
**STAI_19.** I am a steady person. (Reverse)
**STAI_20.** I get in a state of tension or turmoil as I think over my recent concerns and interests.

## Educational attainment questions

Educational attainment was based on the ISCED 2011 categories and included the following options mapped onto 1–9 in response to the question 'What is the highest level of education you have completed to this date?':

1. 'Early childhood education or no formal education (e.g. early childhood education and development, play school, reception, pre-primary, pre-school, educación inicial)'
2. 'Primary education (e.g. primary education, elementary education, basic education; typically ends around age 10–12 years)'
3. 'Lower secondary education (e.g. lower grades of secondary school, junior secondary school, middle school, junior high school)'
4. 'Upper secondary education (e.g. upper grades of secondary school, senior secondary school, senior high school; typically ends around age 17–18 years)'
5. 'Post-secondary non-tertiary education (e.g. technician diploma, primary professional education, préparation aux carrières administratives; usually designed for direct labour market entry)'
6. 'Short-cycle tertiary education (e.g. junior college, higher technical education, community college education, technician or advanced/higher vocational training, associate degree, bac+2; practically based, occupationally specific and prepare for the labour market but can also be a pathway to other tertiary education programmes)'
7. 'Bachelor's or equivalent level'
8. 'Master's or equivalent level'

9. 'Doctoral or equivalent level'

## Psychometric curve fitting

We used the *quickpsy* package in R to fit psychometric curves to each participant's choice data to derive their *AIP*, which was operationalised as the threshold parameter when predicting reminder choices from target values. We set the initial parameter ranges from 2 to 9 for the threshold parameter and from 1 to 500 for the slope parameter, based on the task's properties and pilot data. Such a restriction of the threshold parameter was intended to increase the comparability between AIP and OIP, and hence improved the calculation of the *reminder bias*. Apart from those parameter ranges, we used only default settings of the *quickpsy*() function.

Each participant has only 16 trials (2 for each target value) contribute to the curve fitting. To understand the robustness of the AIP based on such limited data, we conducted a parameter recovery analysis. We simulated 16 trials based on each psychometric function and re-ran the curve fitting based on those simulated choices. There was close correspondence between the actual and recovered threshold parameters (or AIPs) with a correlation of *r*=0.94, p<0.001 (see also *Appendix 1—figure 1*). In contrast, the slope parameter – which was not central to any of our analyses – exhibited greater variability during the initial fitting. This increased uncertainty likely contributed to slightly poorer recovery in the simulation (*r*=0.23, p<0.001).

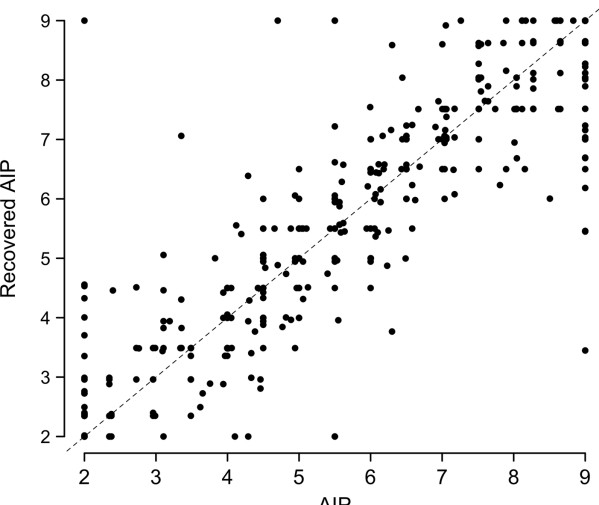

**Appendix 1—figure 1.** The actual indifference point (AIP) is shown on the x-axis against its recovered estimates on the y-axis. Each marker represents one participant's estimates.

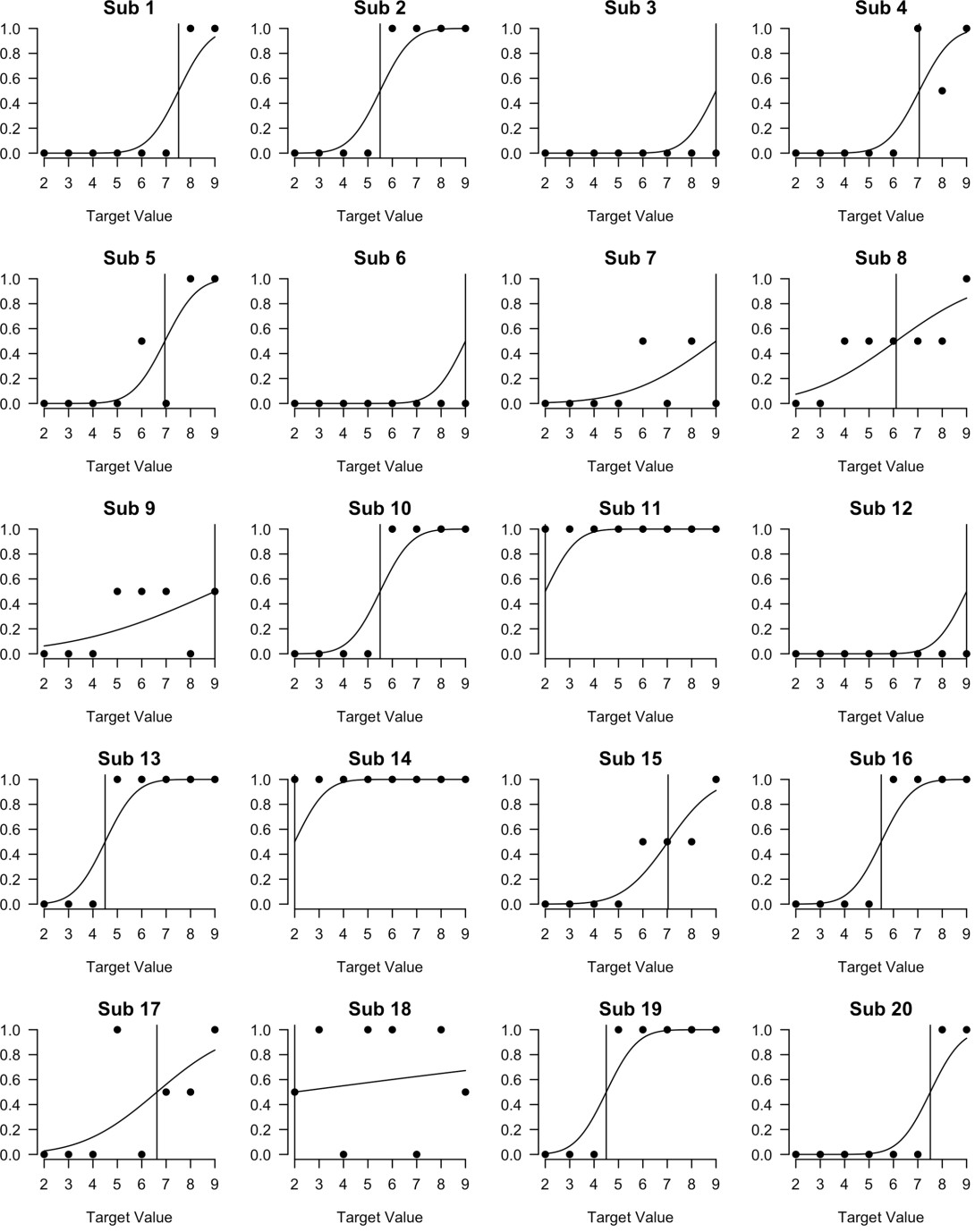

**Appendix 1—figure 2.** Psychometric functions linking target values to offloading choices. The average choice data is shown as dots. Panels show the individual curves for participants 1–20.

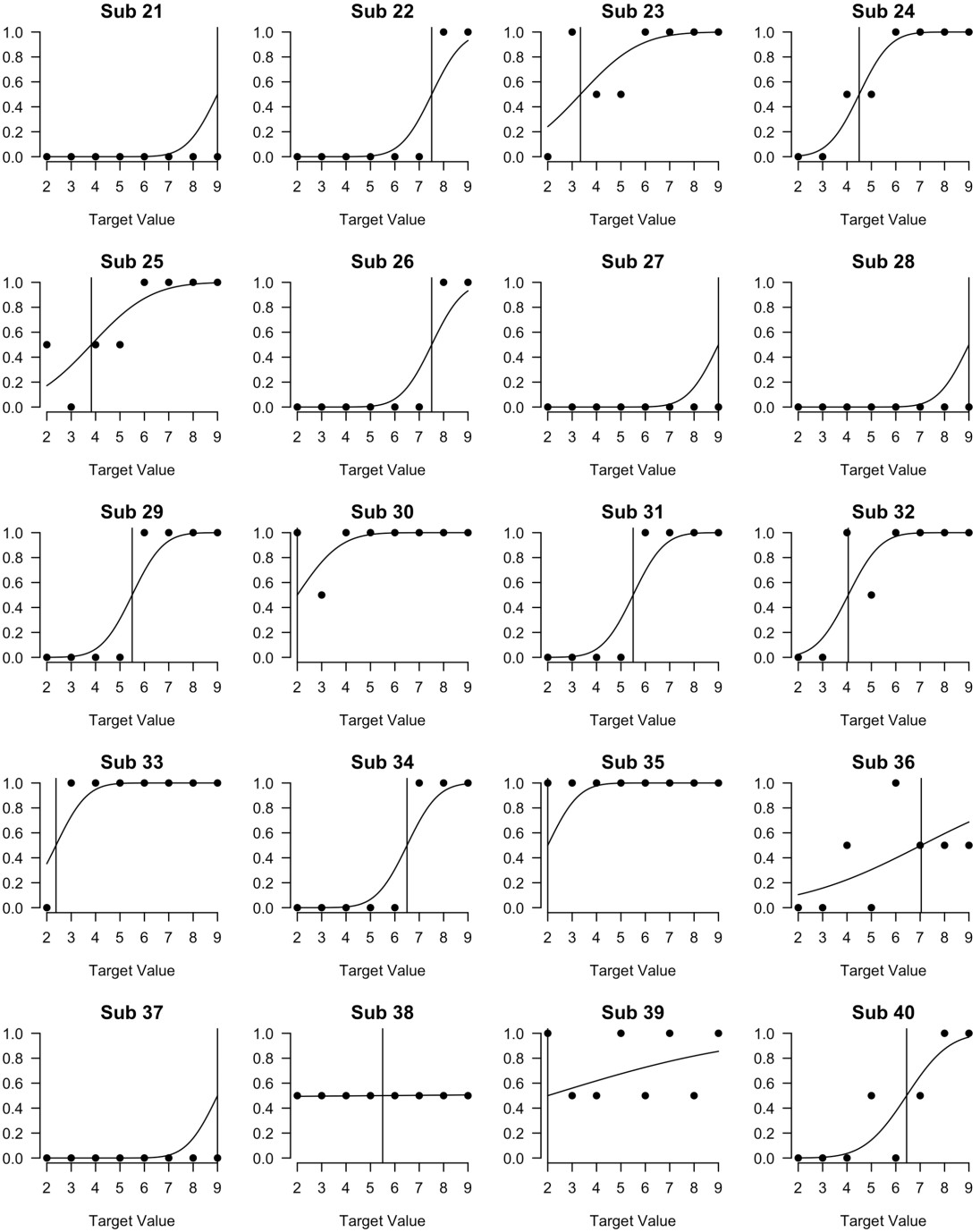

**Appendix 1—figure 3.** Psychometric functions linking target values to offloading choices. The average choice data is shown as dots. Panels show the individual curves for participants 21–40.

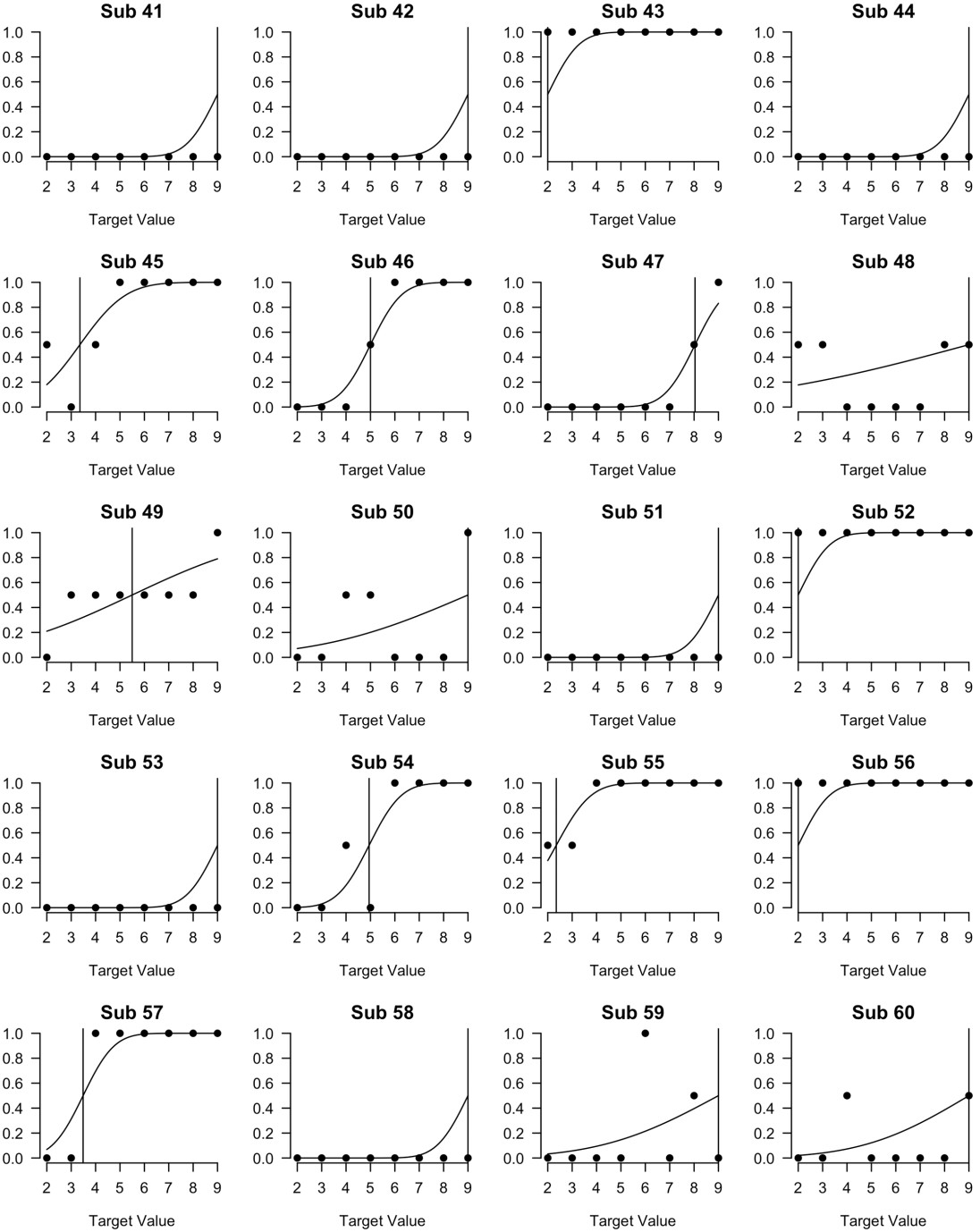

**Appendix 1—figure 4.** Psychometric functions linking target values to offloading choices. The average choice data is shown as dots. Panels show the individual curves for participants 41–60.

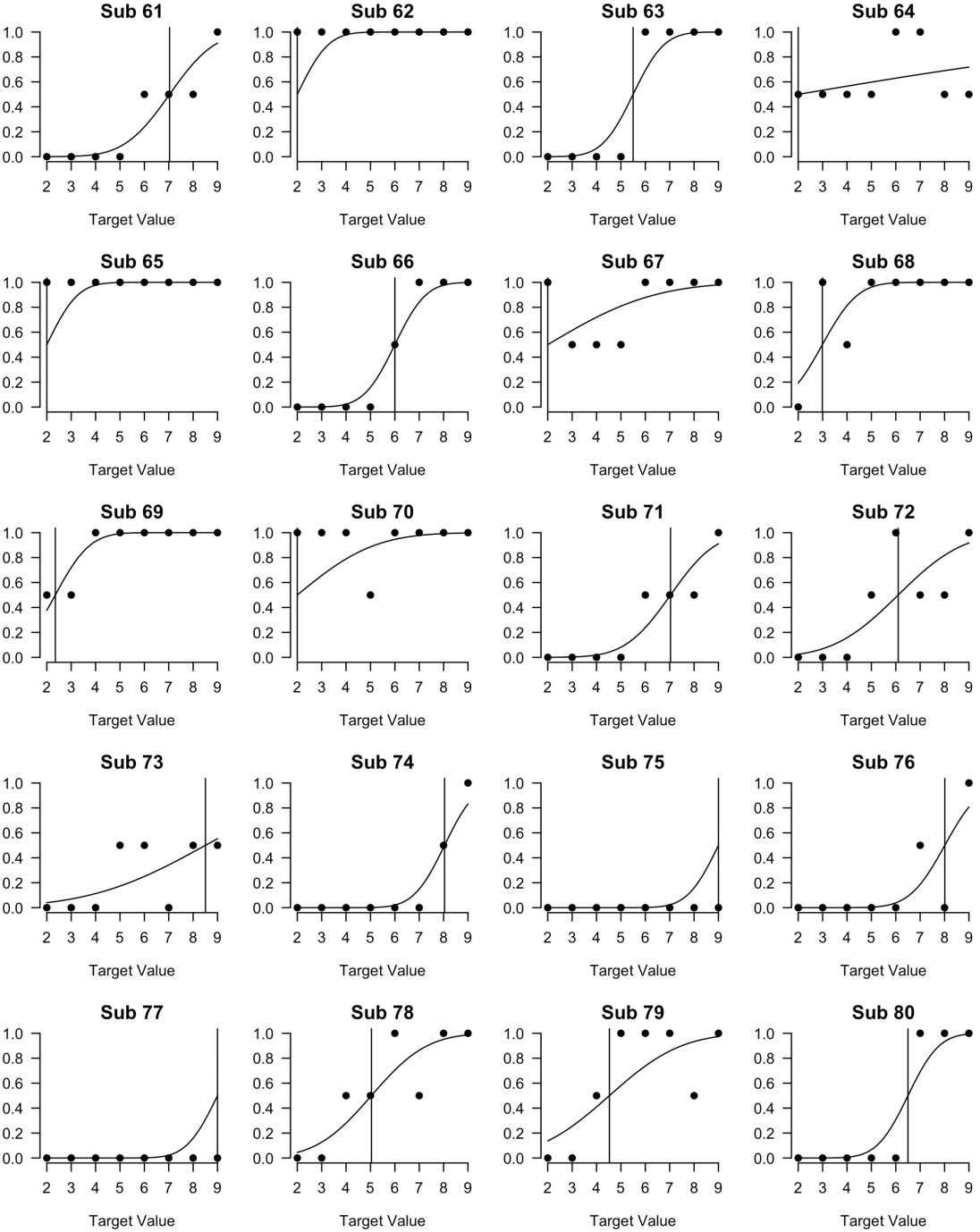

**Appendix 1—figure 5.** Psychometric functions linking target values to offloading choices. The average choice data is shown as dots. Panels show the individual curves for participants 61–80.

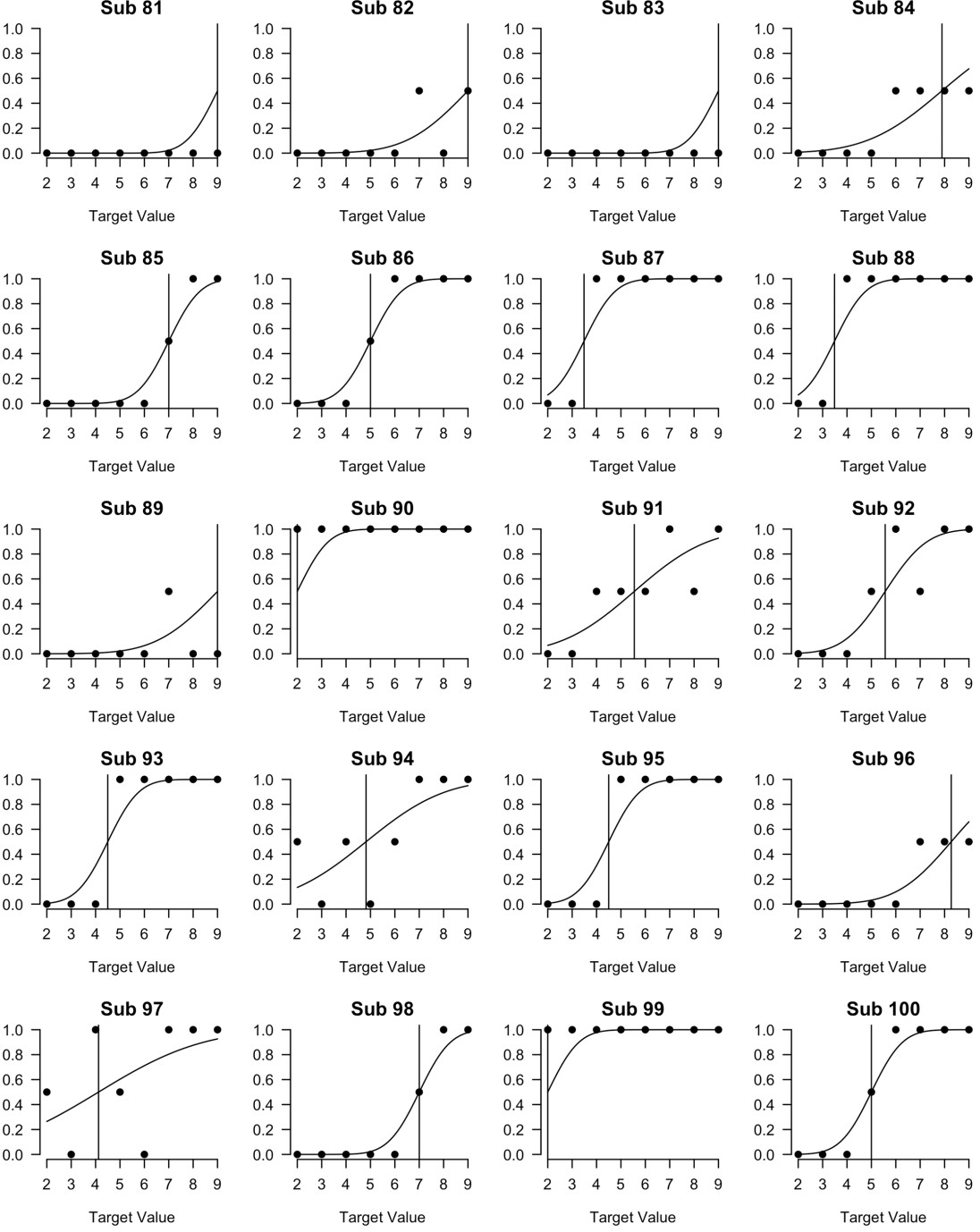

**Appendix 1—figure 6.** Psychometric functions linking target values to offloading choices. The average choice data is shown as dots. Panels show the individual curves for participants 81–100.

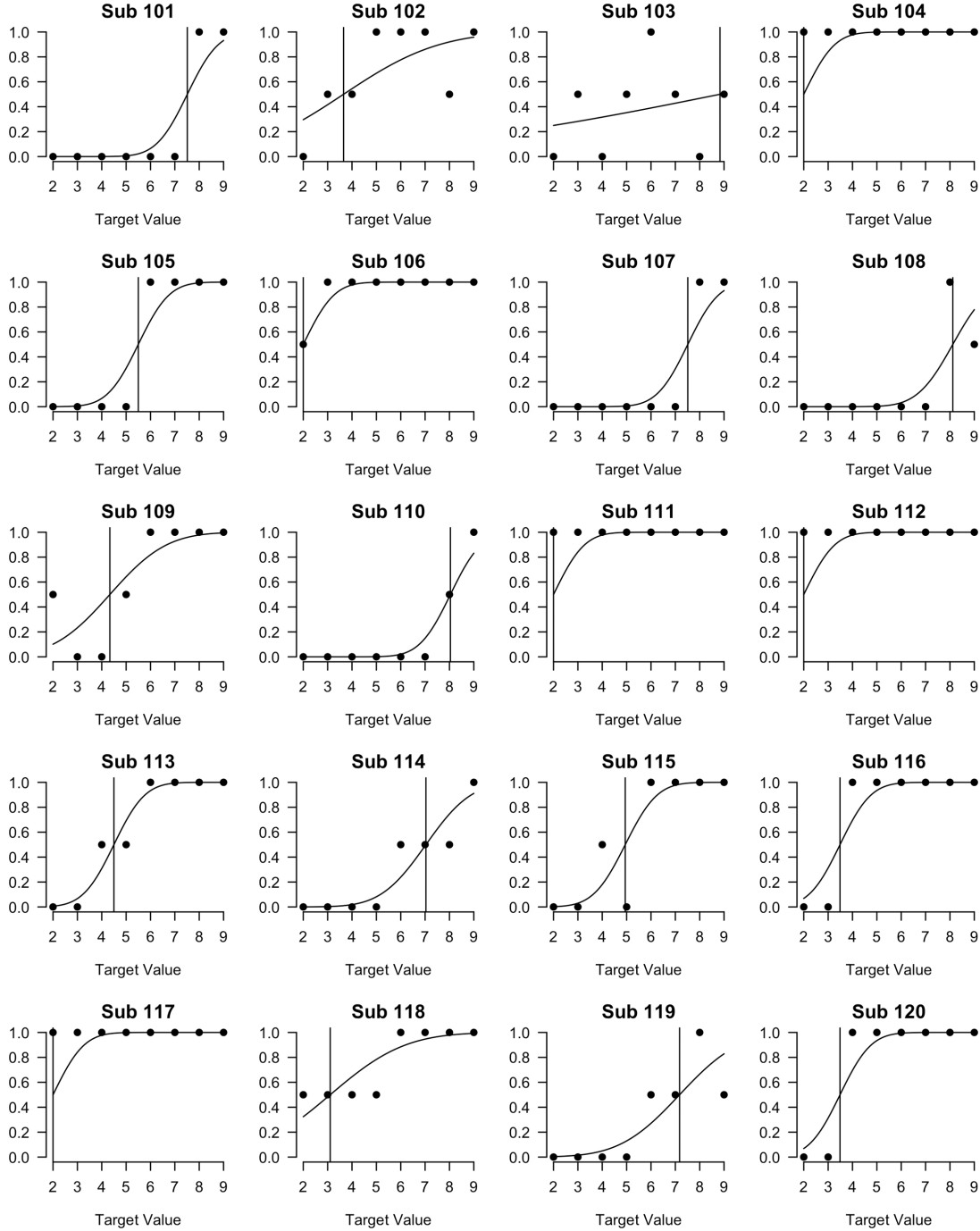

**Appendix 1—figure 7.** Psychometric functions linking target values to offloading choices. The average choice data is shown as dots. Panels show the individual curves for participants 101–120.

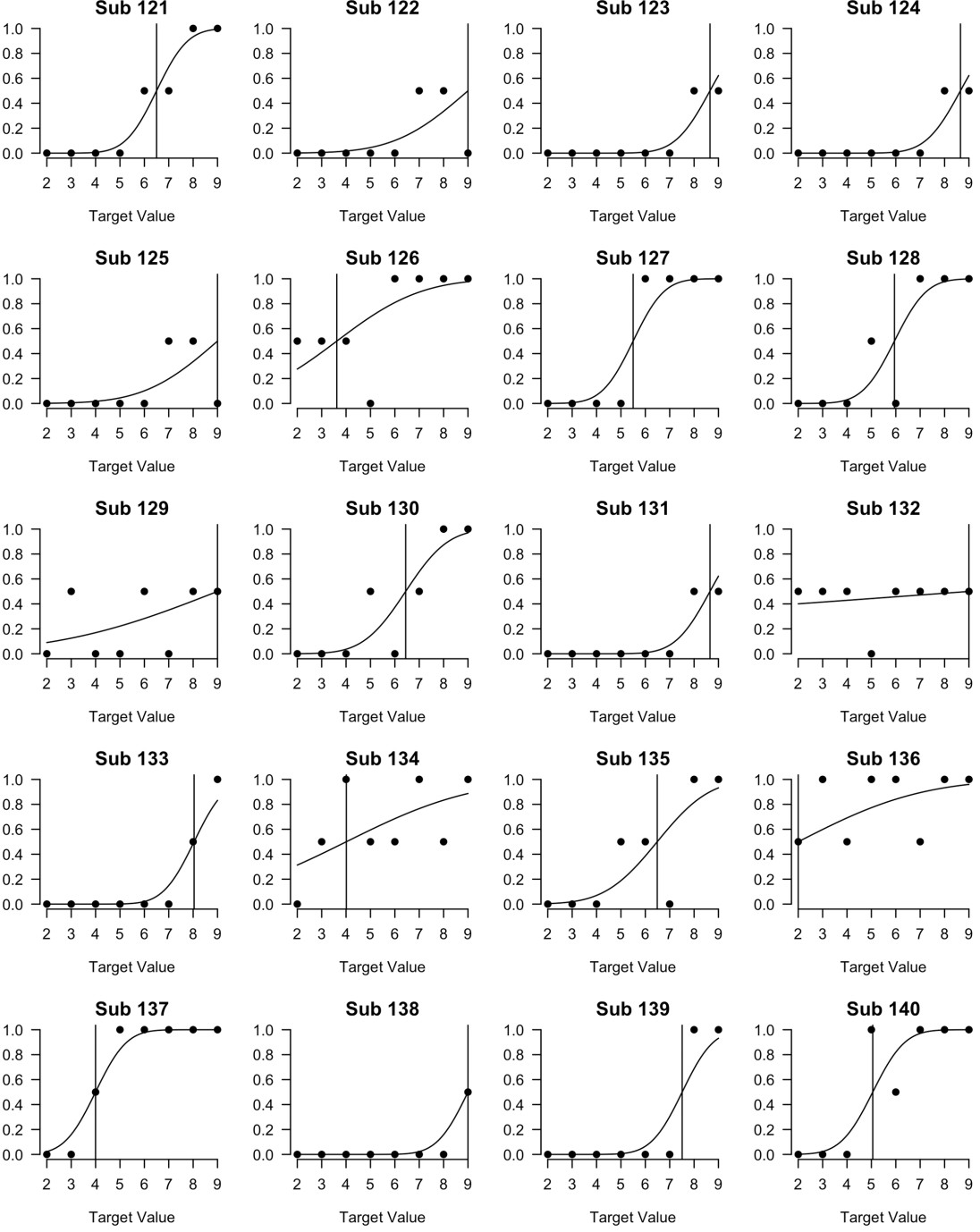

**Appendix 1—figure 8.** Psychometric functions linking target values to offloading choices. The average choice data is shown as dots. Panels show the individual curves for participants 121–140.

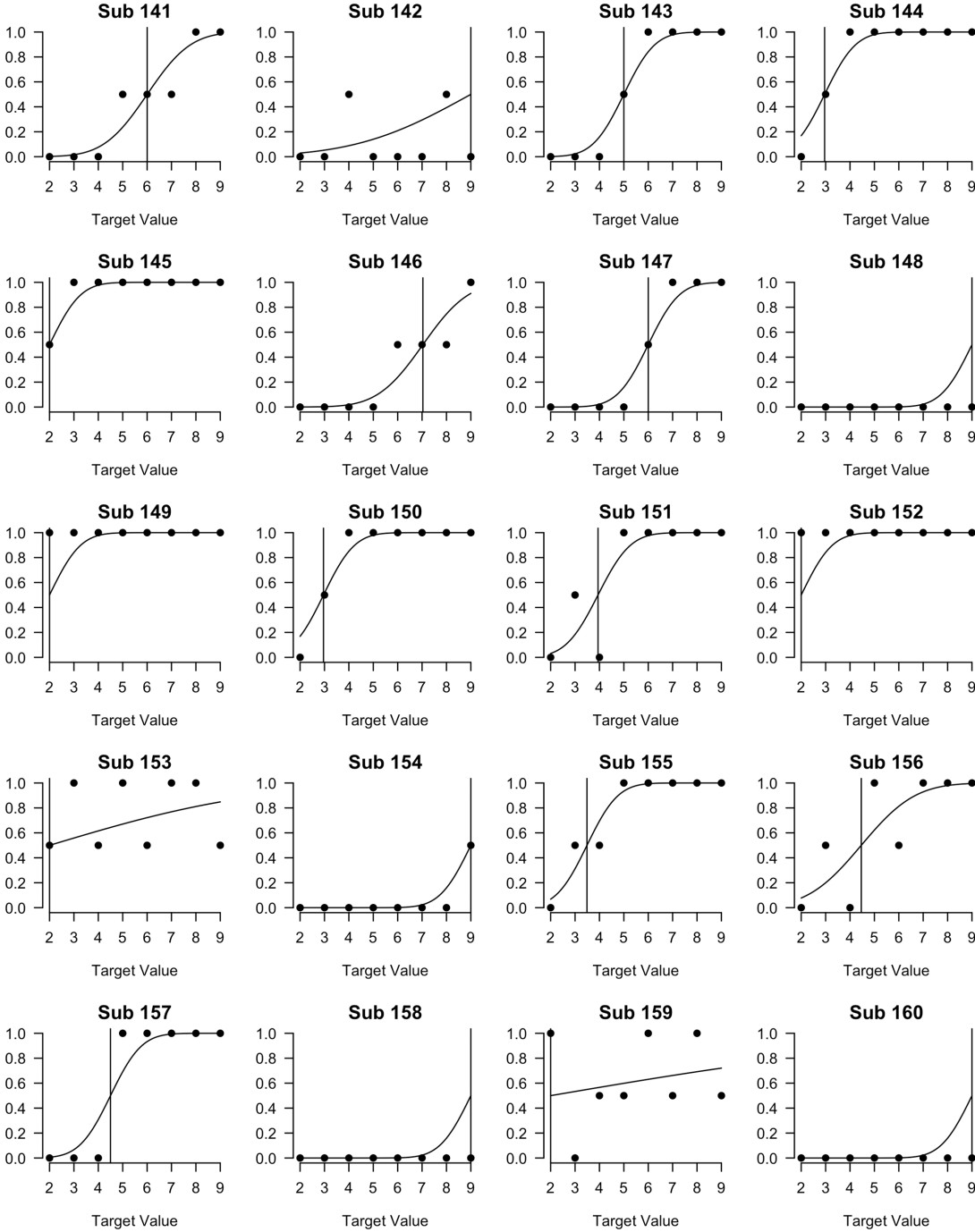

**Appendix 1—figure 9.** Psychometric functions linking target values to offloading choices. The average choice data is shown as dots. Panels show the individual curves for participants 141–160.

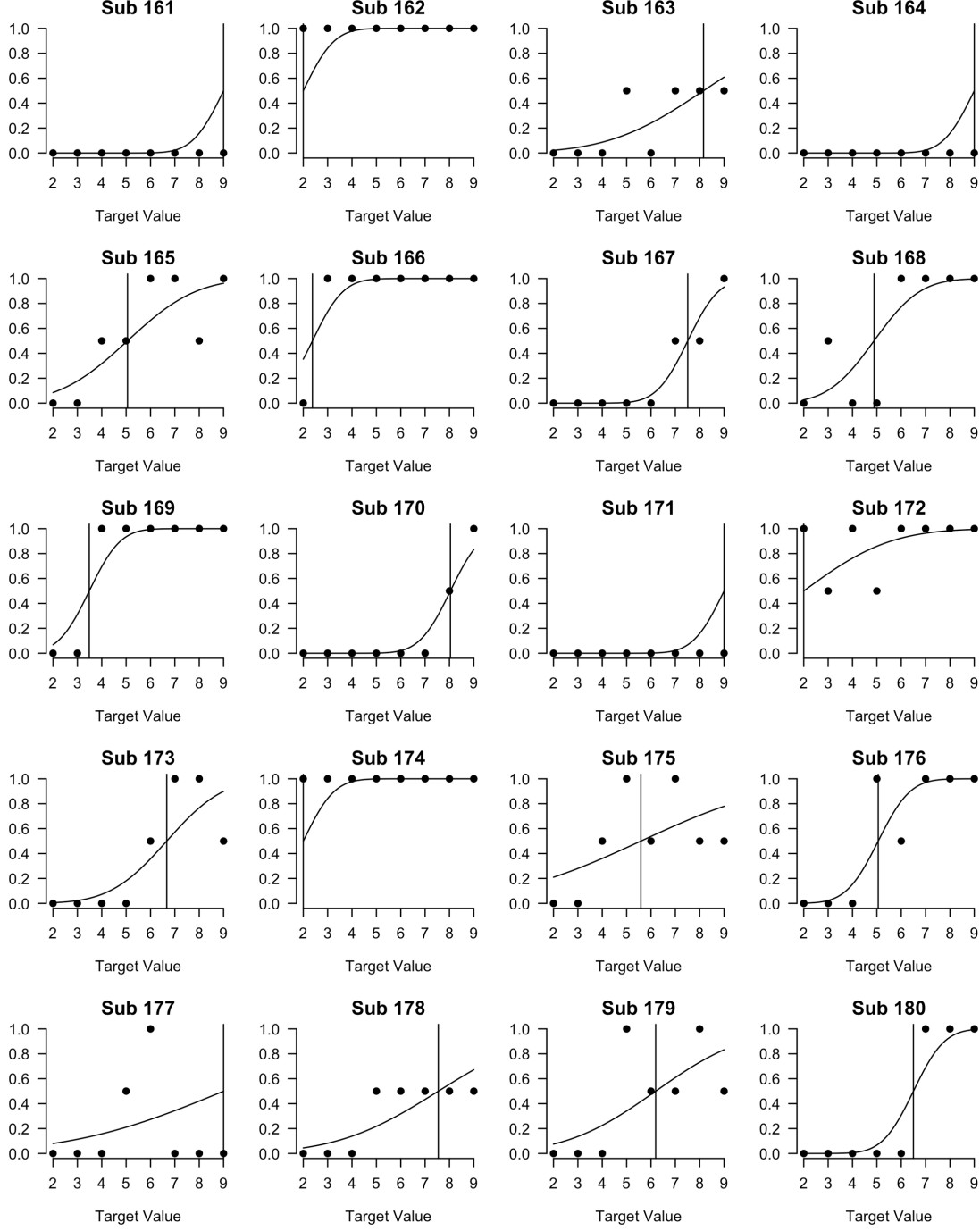

**Appendix 1—figure 10.** Psychometric functions linking target values to offloading choices. The average choice data is shown as dots. Panels show the individual curves for participants 161–180.

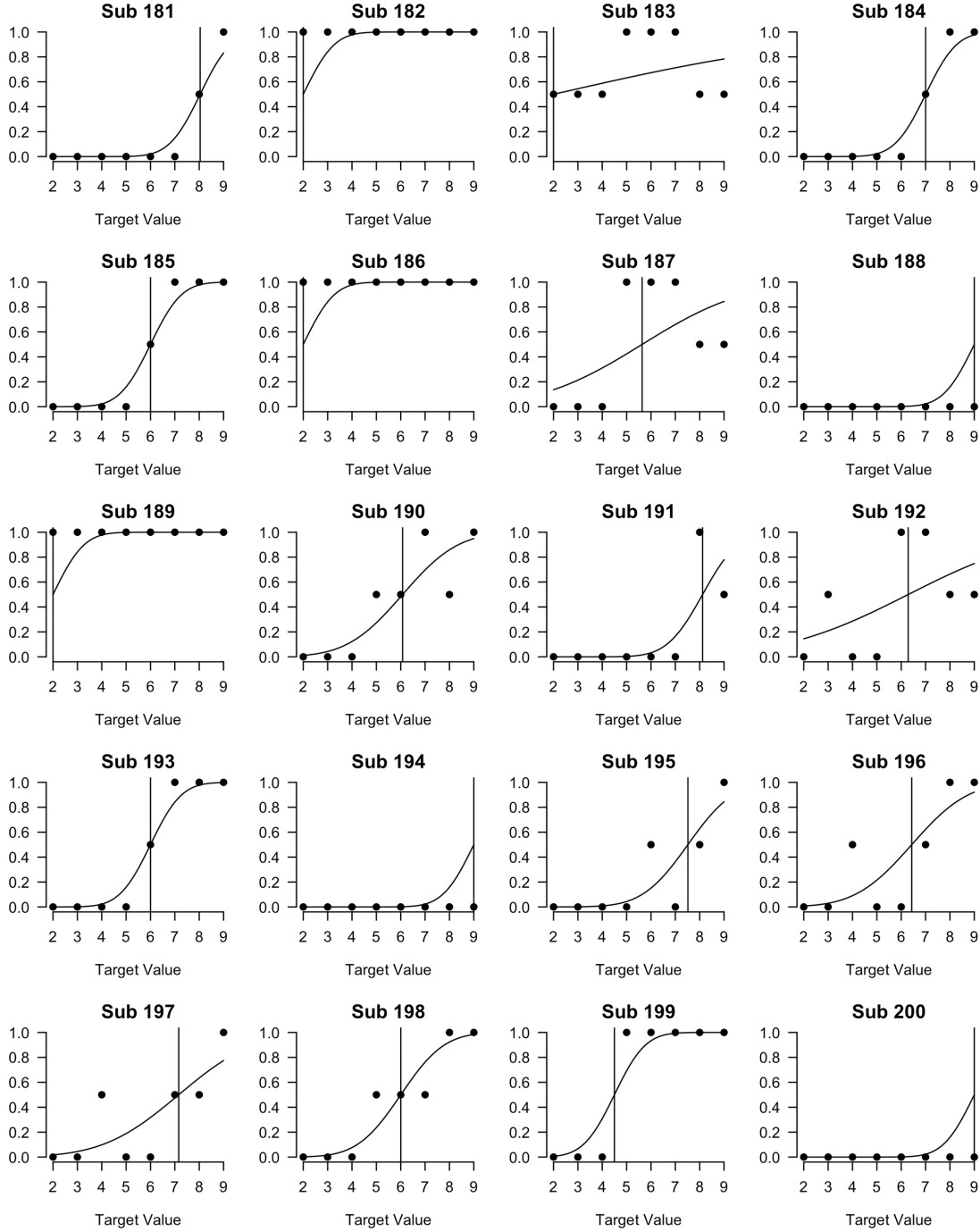

**Appendix 1—figure 11.** Psychometric functions linking target values to offloading choices. The average choice data is shown as dots. Panels show the individual curves for participants 181–200.

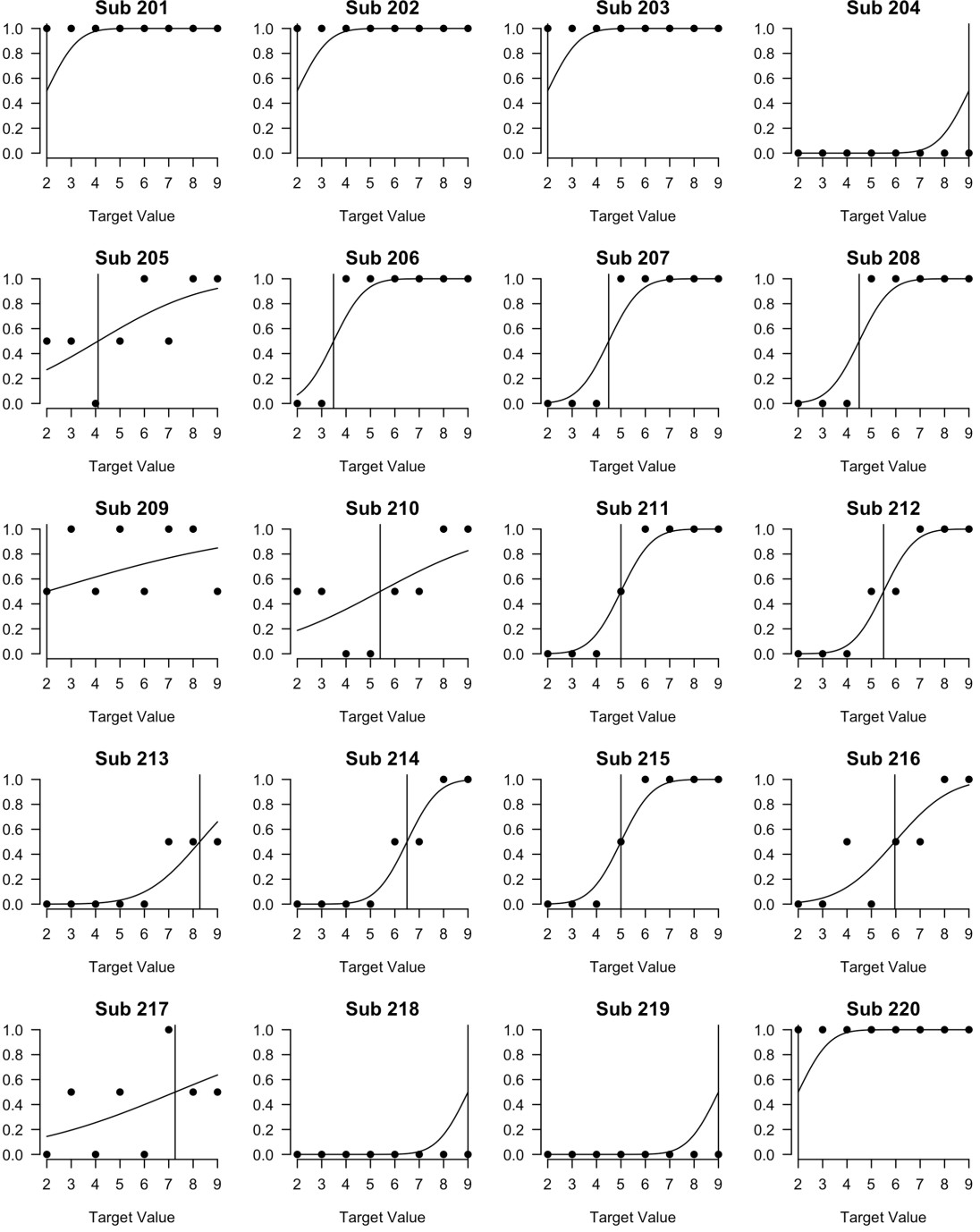

**Appendix 1—figure 12.** Psychometric functions linking target values to offloading choices. The average choice data is shown as dots. Panels show the individual curves for participants 201–220.

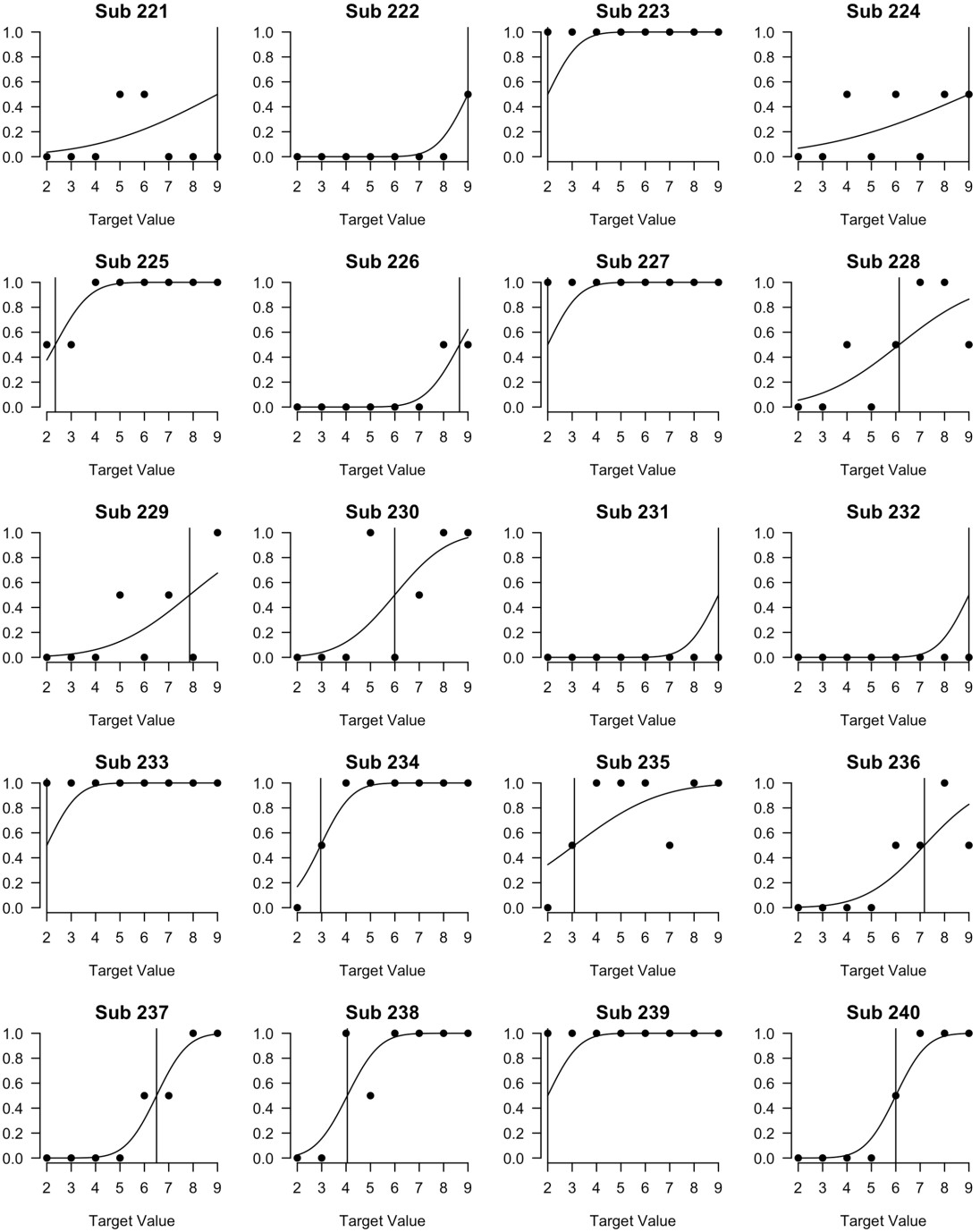

**Appendix 1—figure 13.** Psychometric functions linking target values to offloading choices. The average choice data is shown as dots. Panels show the individual curves for participants 221–240.

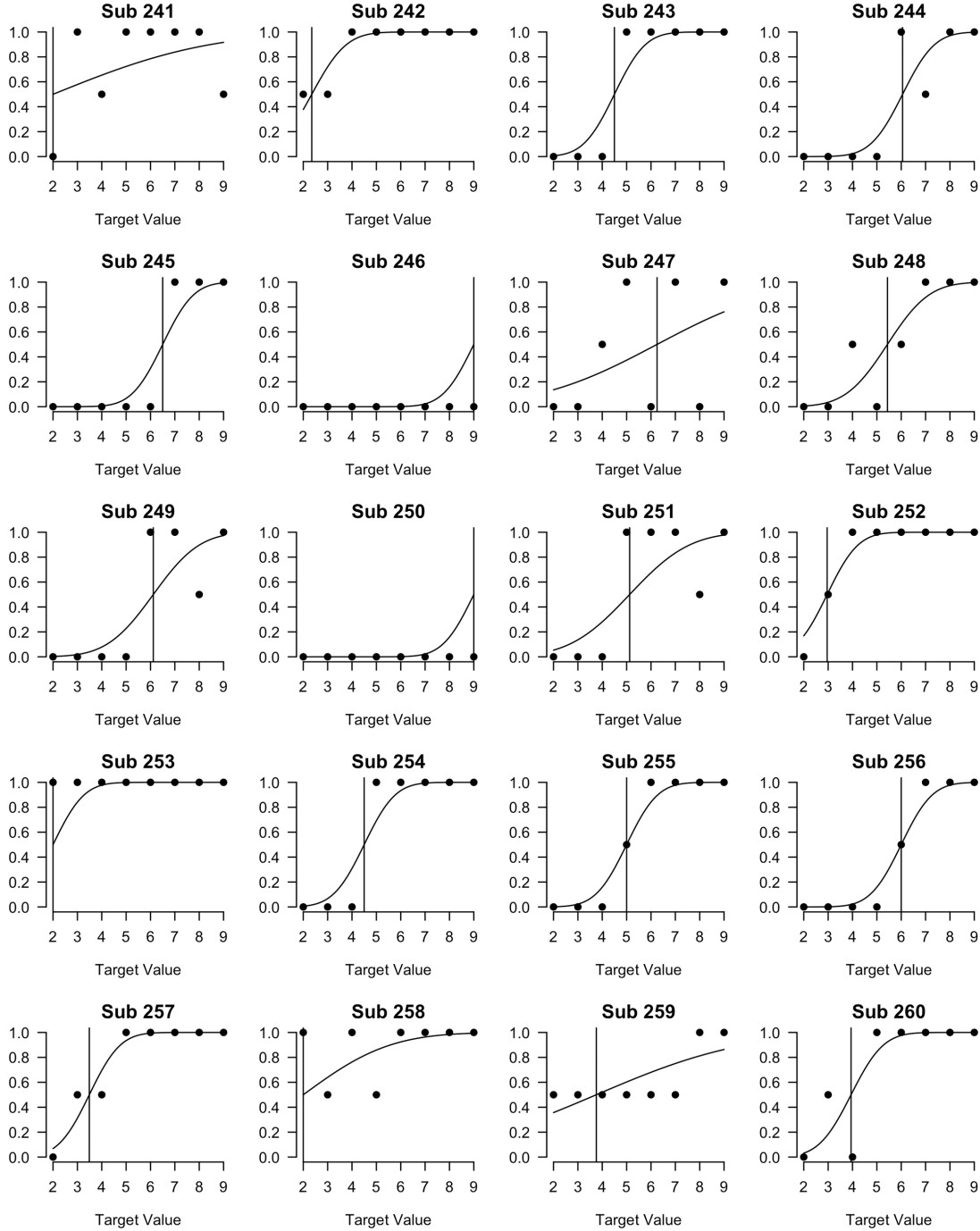

**Appendix 1—figure 14.** Psychometric functions linking target values to offloading choices. The average choice data is shown as dots. Panels show the individual curves for participants 241–260.

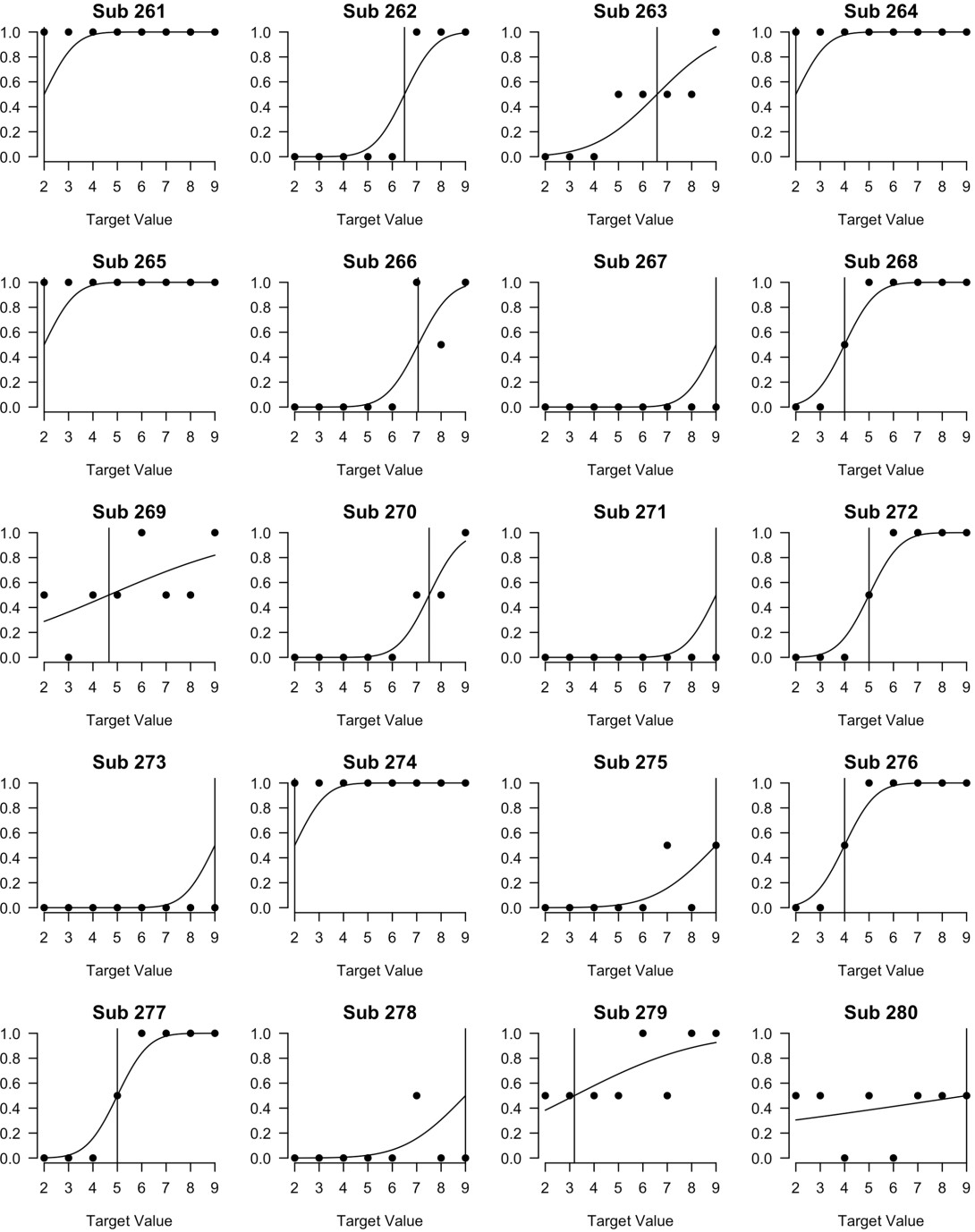

**Appendix 1—figure 15.** Psychometric functions linking target values to offloading choices. The average choice data is shown as dots. Panels show the individual curves for participants 261–280.

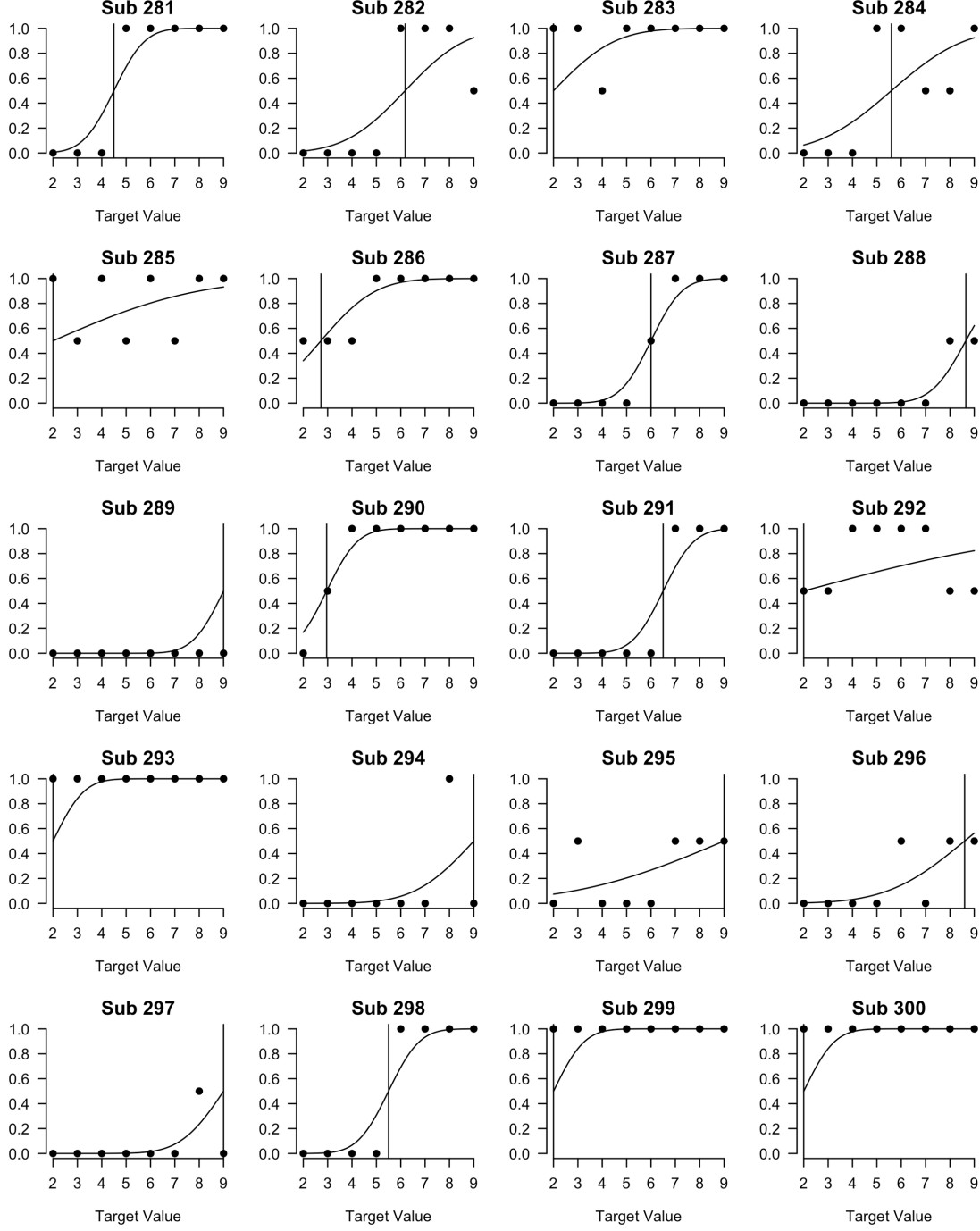

**Appendix 1—figure 16.** Psychometric functions linking target values to offloading choices. The average choice data is shown as dots. Panels show the individual curves for participants 281–300.

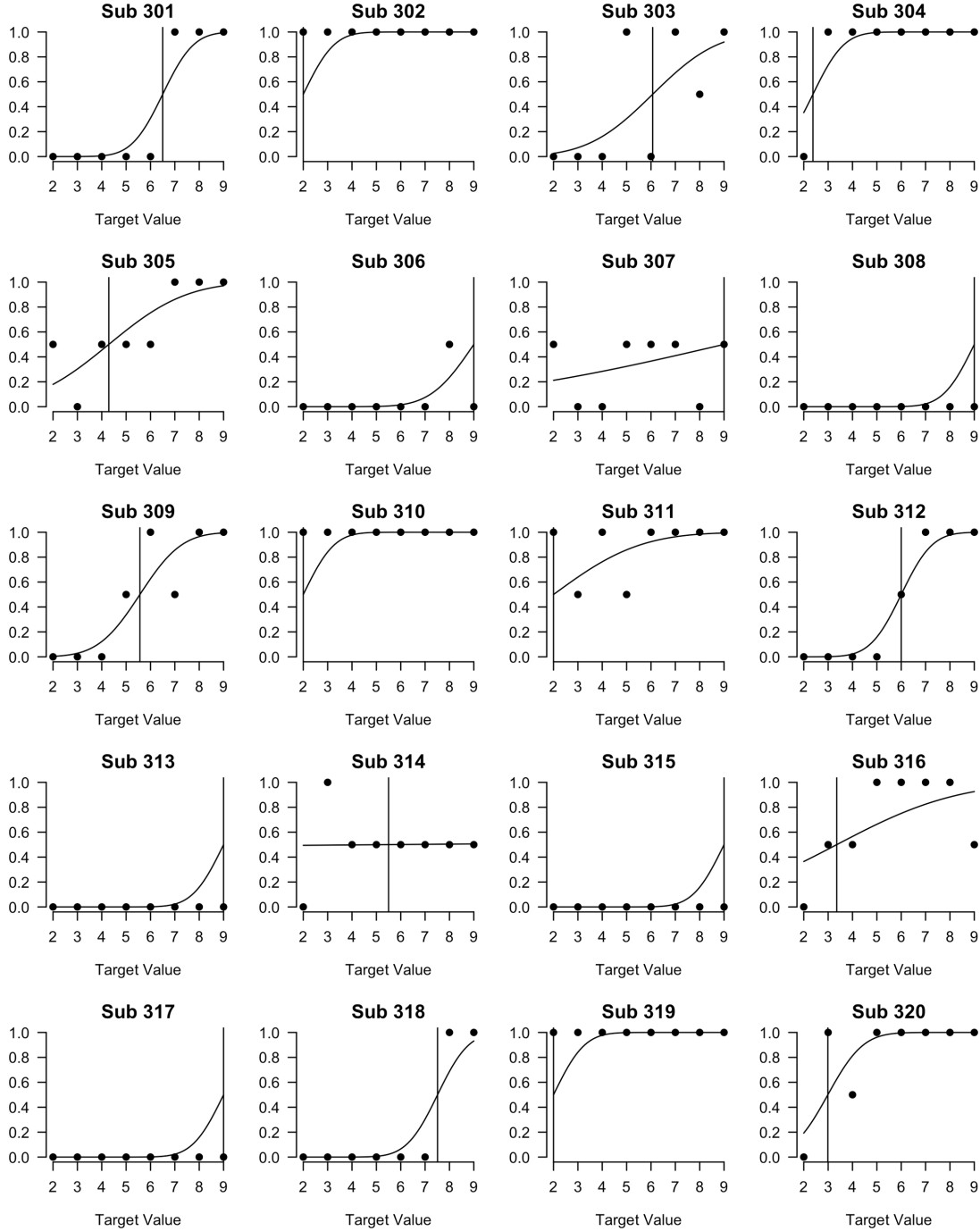

**Appendix 1—figure 17.** Psychometric functions linking target values to offloading choices. The average choice data is shown as dots. Panels show the individual curves for participants 301–320.

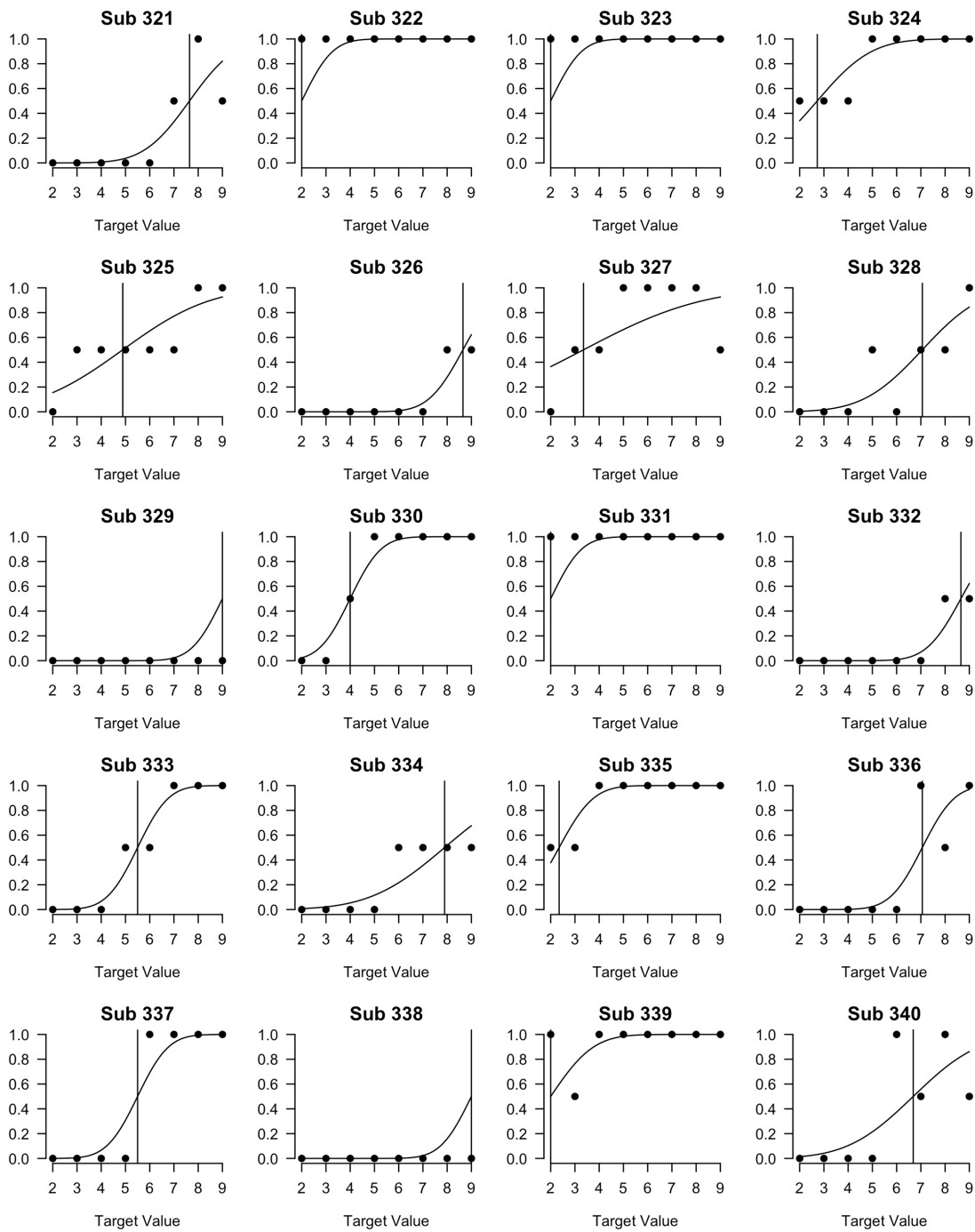

**Appendix 1—figure 18.** Psychometric functions linking target values to offloading choices. The average choice data is shown as dots. Panels show the individual curves for participants 321–340.

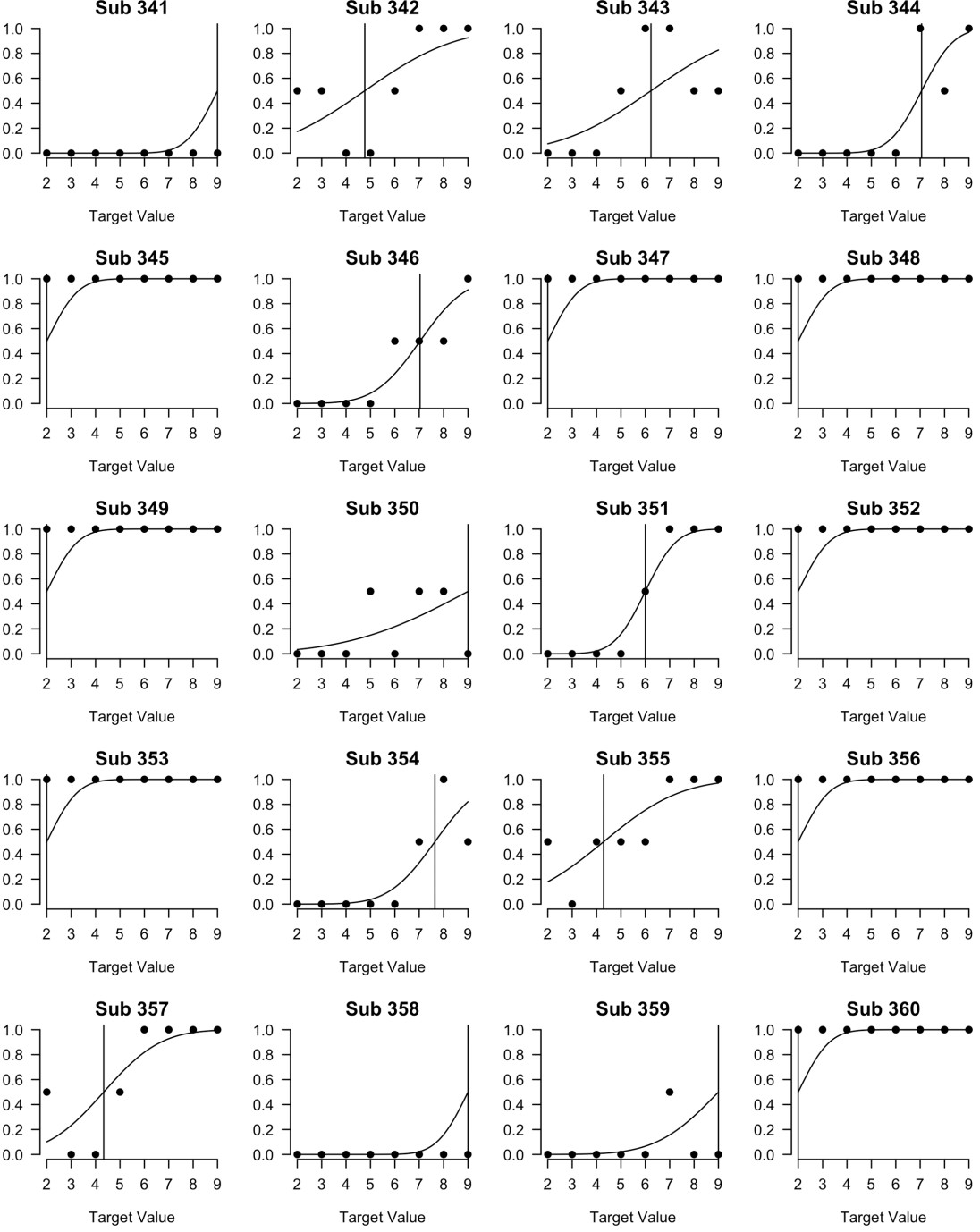

**Appendix 1—figure 19.** Psychometric functions linking target values to offloading choices. The average choice data is shown as dots. Panels show the individual curves for participants 341–360.

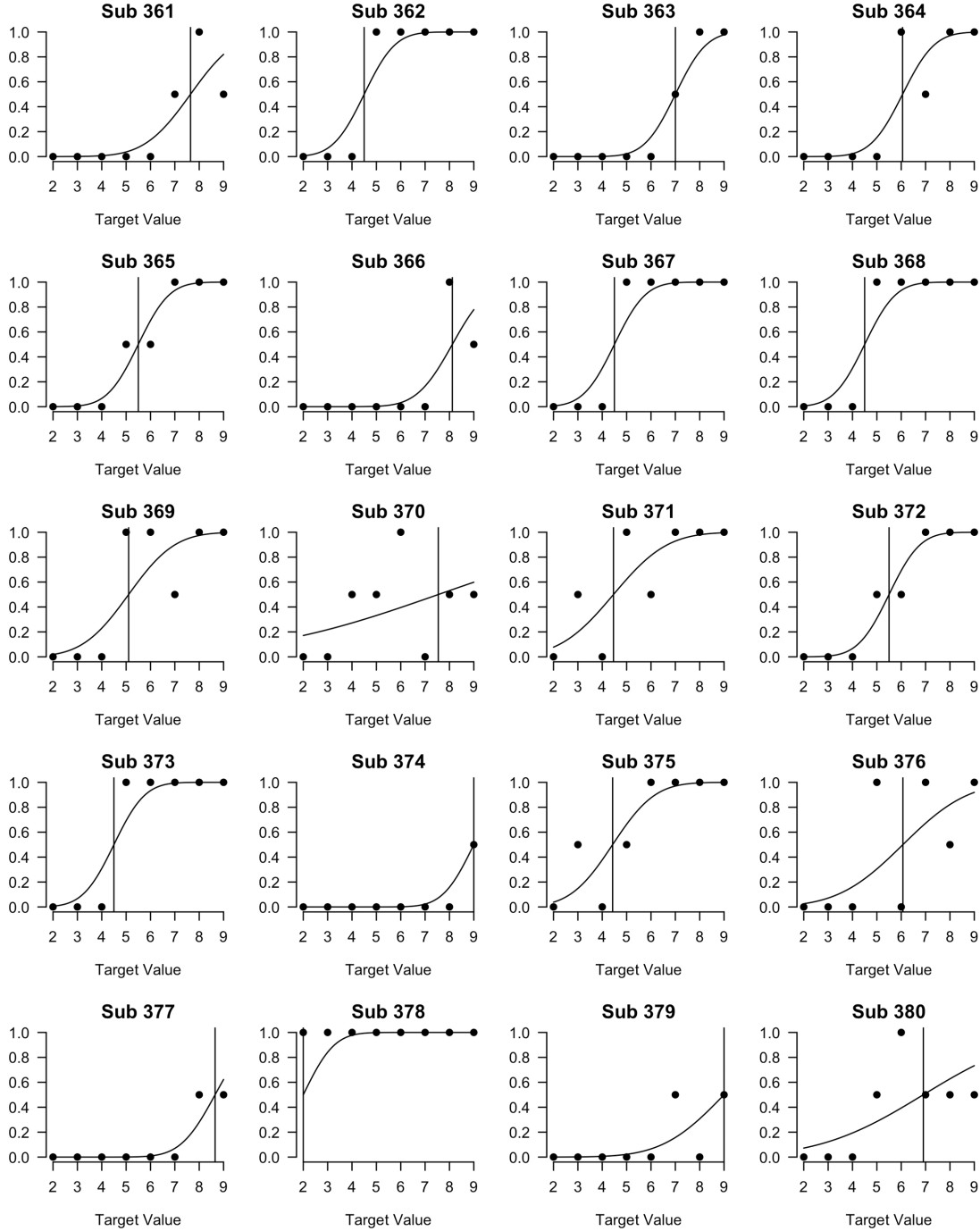

**Appendix 1—figure 20.** Psychometric functions linking target values to offloading choices. The average choice data is shown as dots. Panels show the individual curves for participants 361–380.

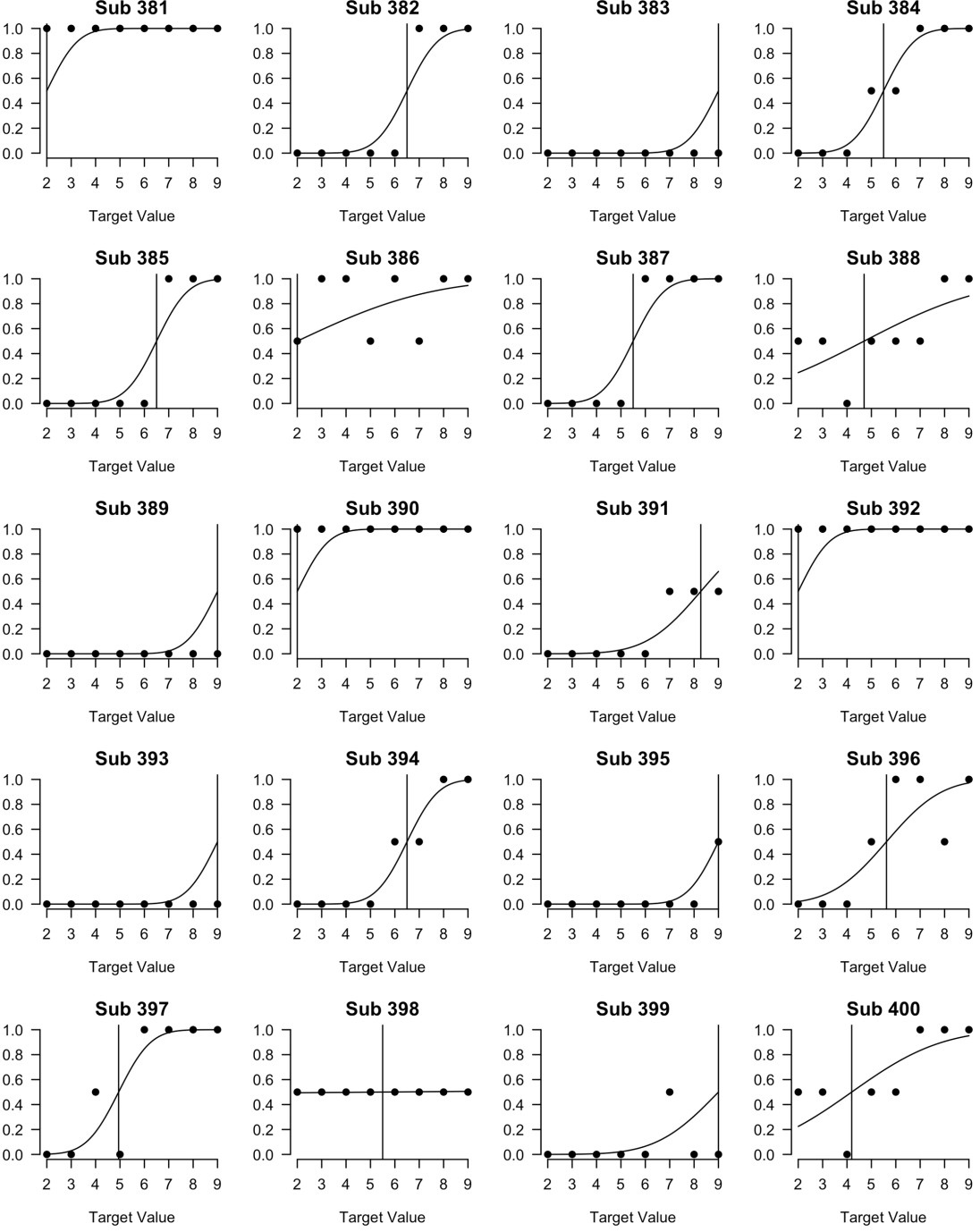

**Appendix 1—figure 21.** Psychometric functions linking target values to offloading choices. The average choice data is shown as dots. Panels show the individual curves for participants 381–400.

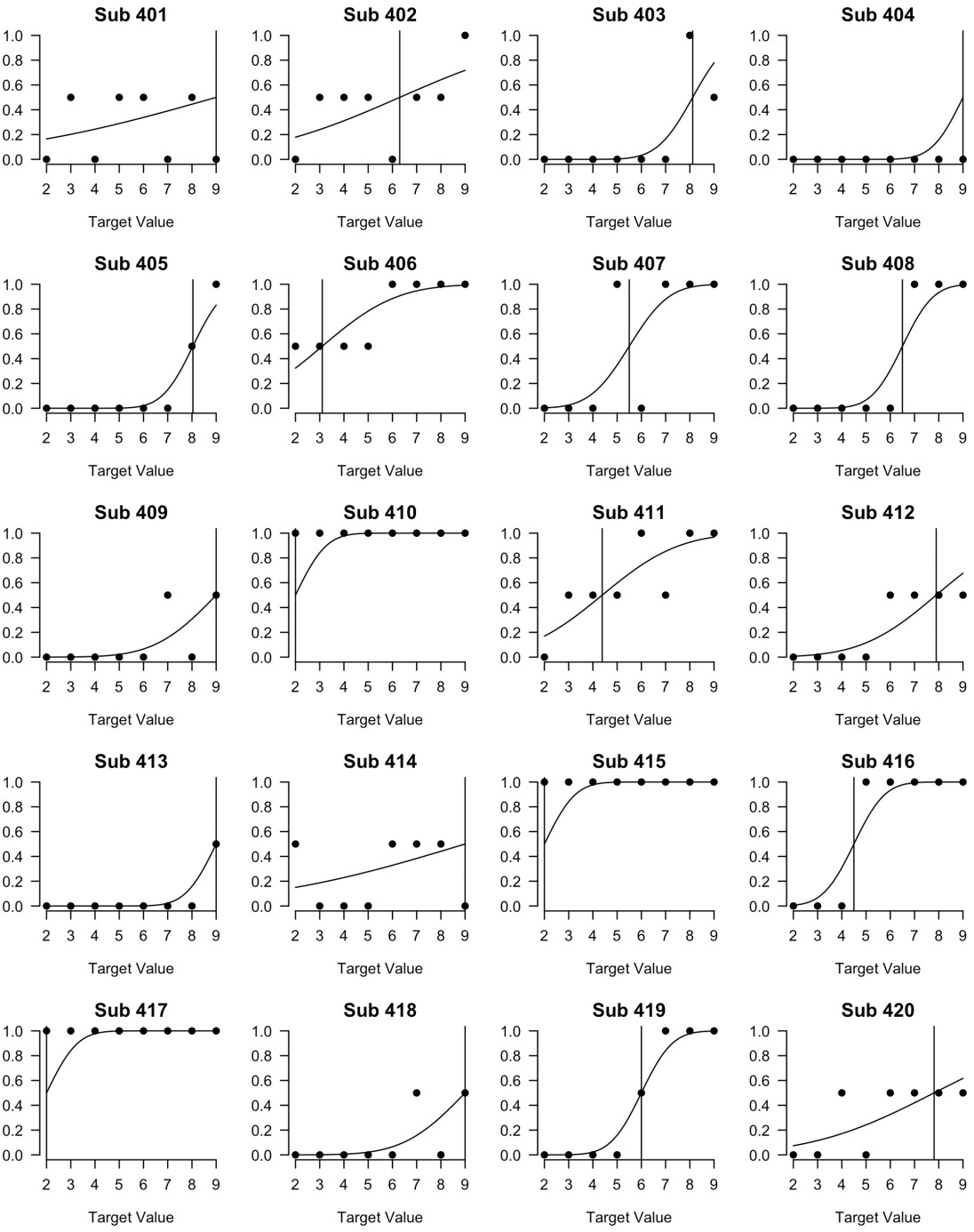

**Appendix 1—figure 22.** Psychometric functions linking target values to offloading choices. The average choice data is shown as dots. Panels show the individual curves for participants 401–420.

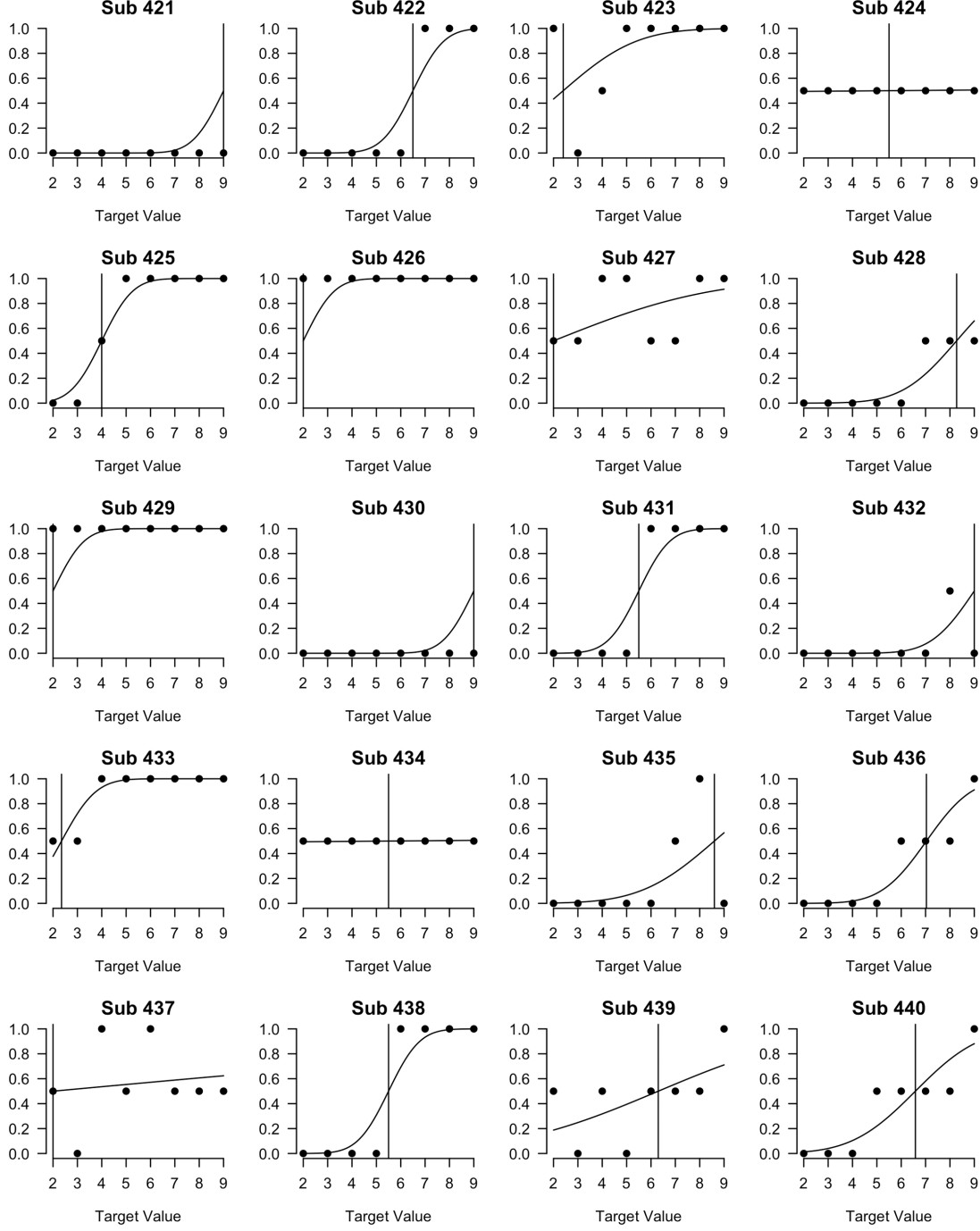

**Appendix 1—figure 23.** Psychometric functions linking target values to offloading choices. The average choice data is shown as dots. Panels show the individual curves for participants 421–440.

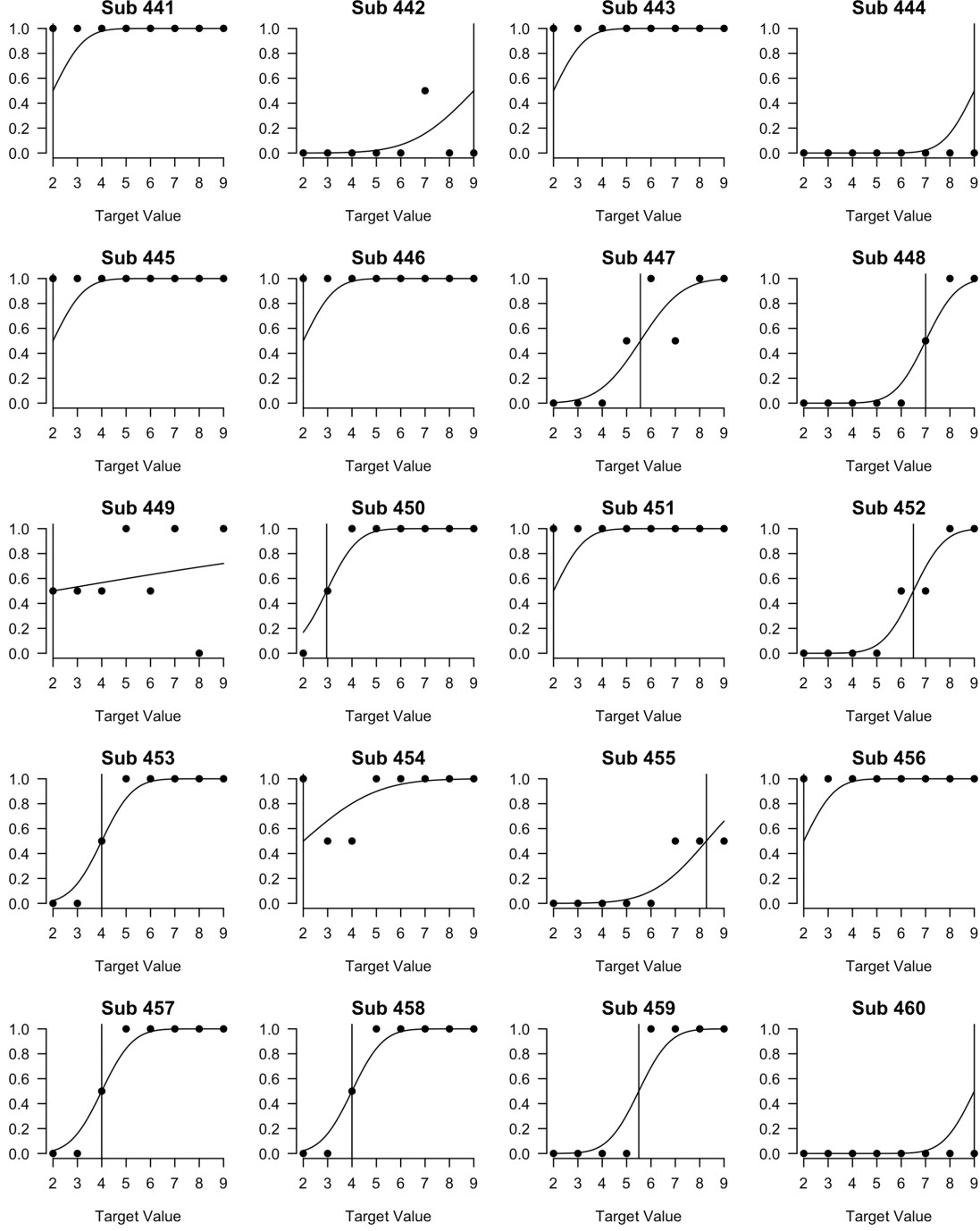

**Appendix 1—figure 24.** Psychometric functions linking target values to offloading choices. The average choice data is shown as dots. Panels show the individual curves for participants 441–460.

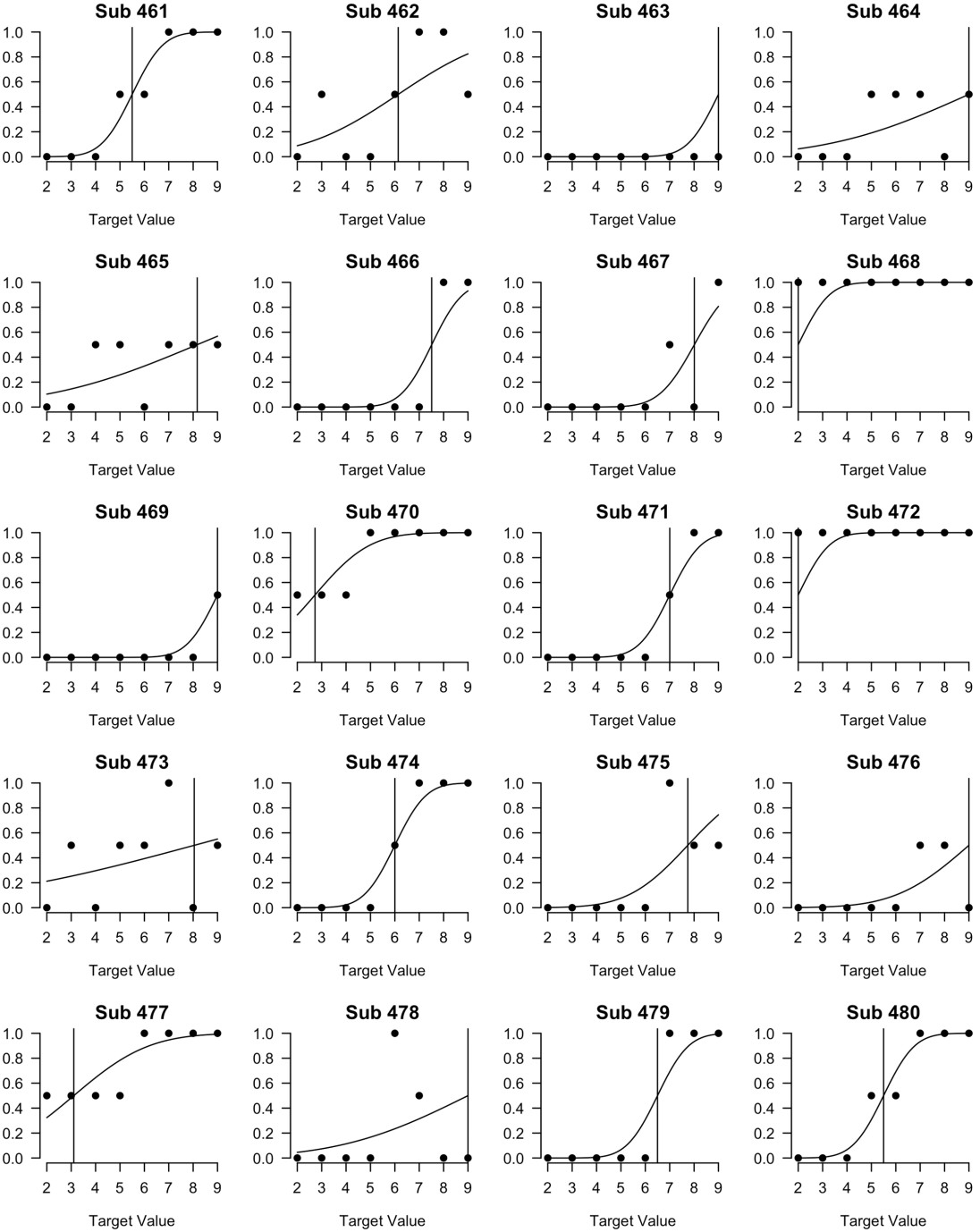

**Appendix 1—figure 25.** Psychometric functions linking target values to offloading choices. The average choice data is shown as dots. Panels show the individual curves for participants 461–480.

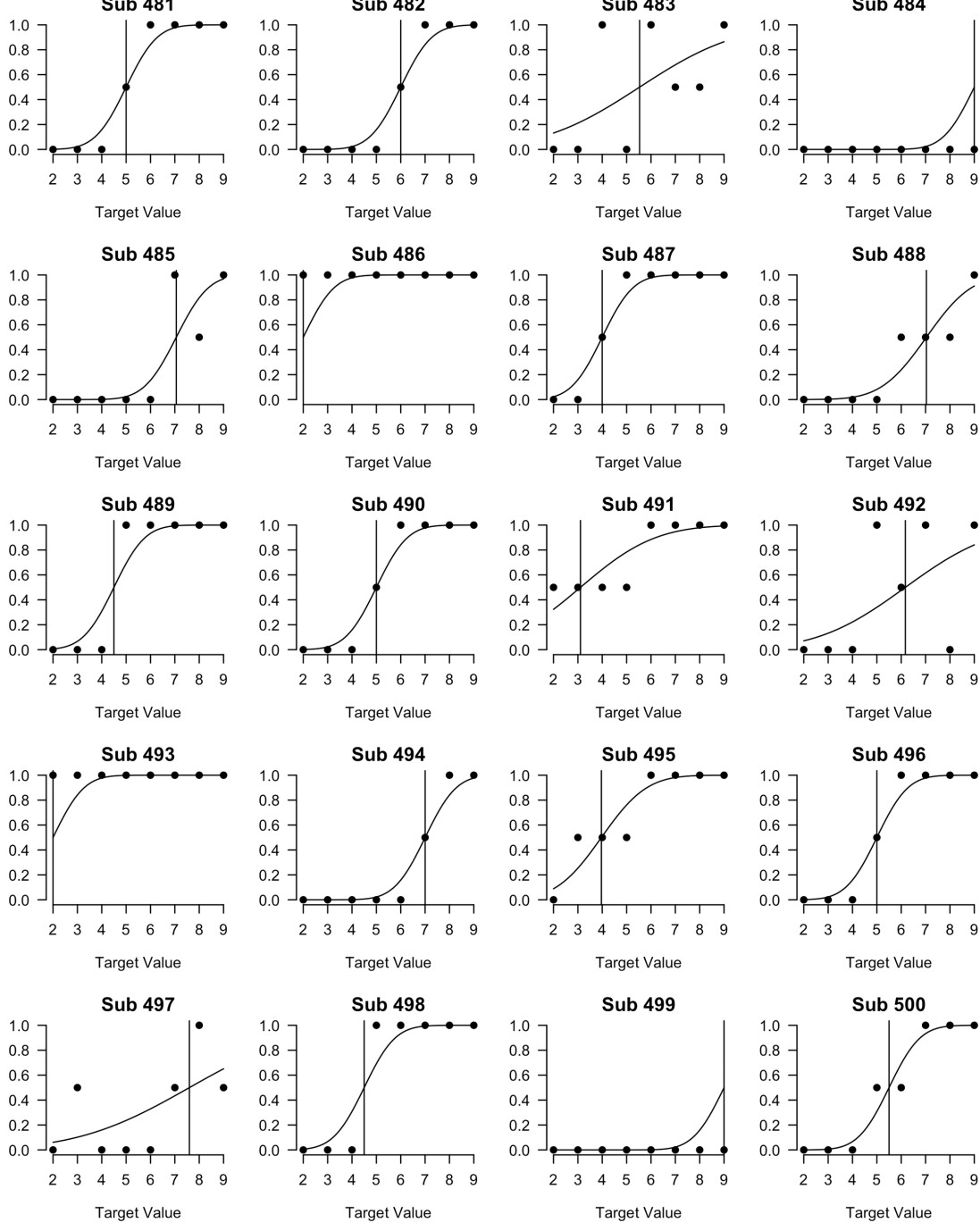

**Appendix 1—figure 26.** Psychometric functions linking target values to offloading choices. The average choice data is shown as dots. Panels show the individual curves for participants 481–500.

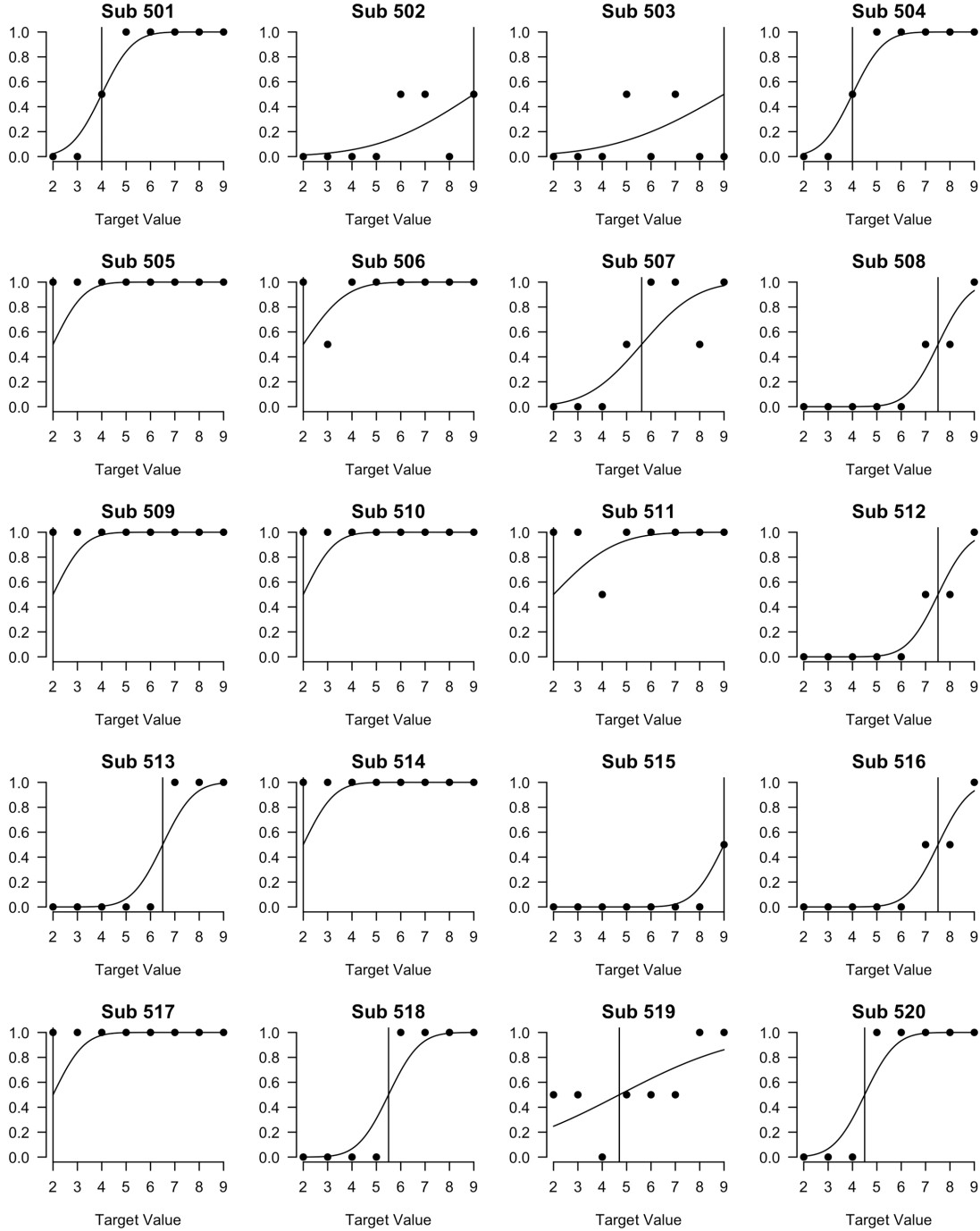

**Appendix 1—figure 27.** Psychometric functions linking target values to offloading choices. The average choice data is shown as dots. Panels show the individual curves for participants 501–520.

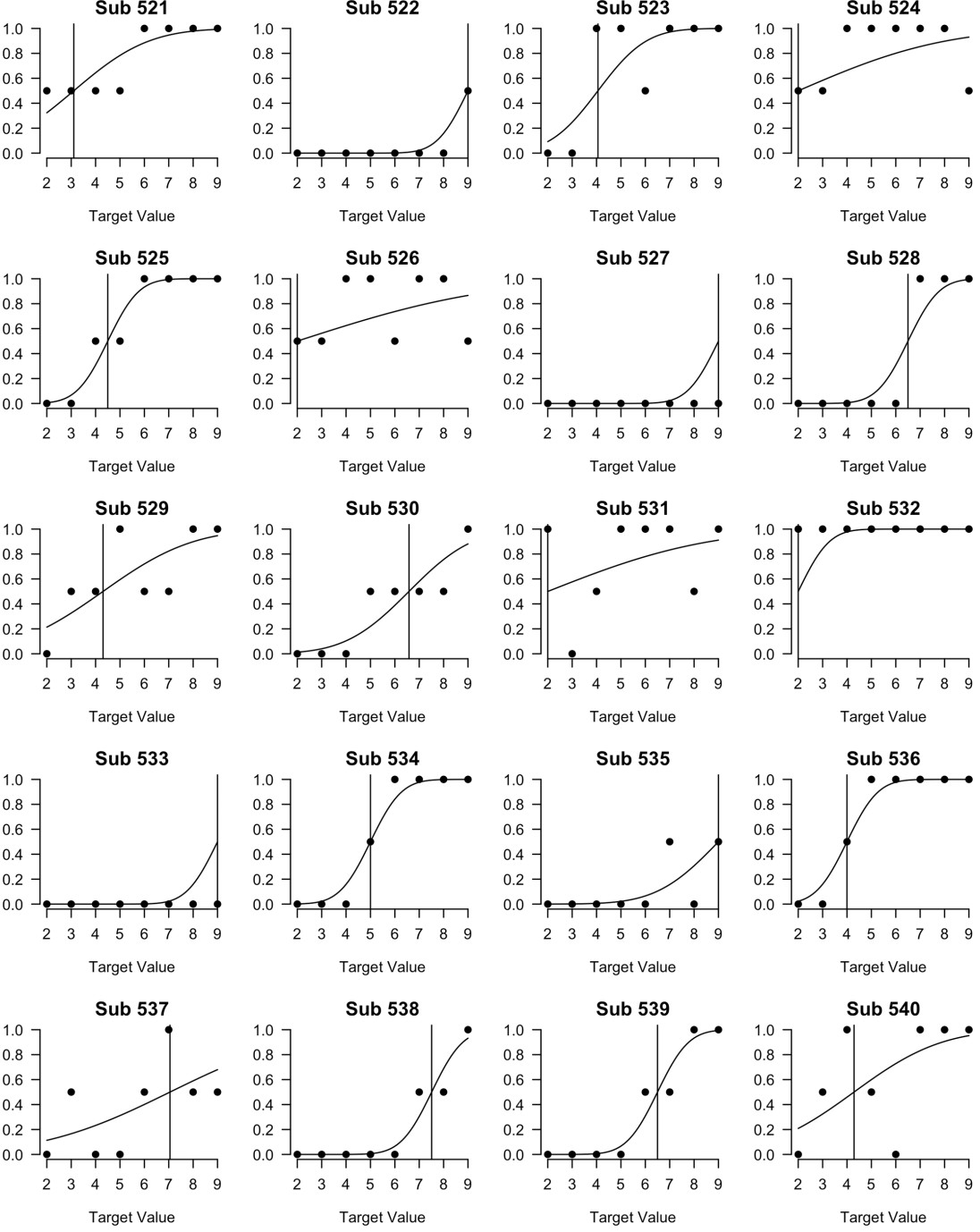

**Appendix 1—figure 28.** Psychometric functions linking target values to offloading choices. The average choice data is shown as dots. Panels show the individual curves for participants 521–540.

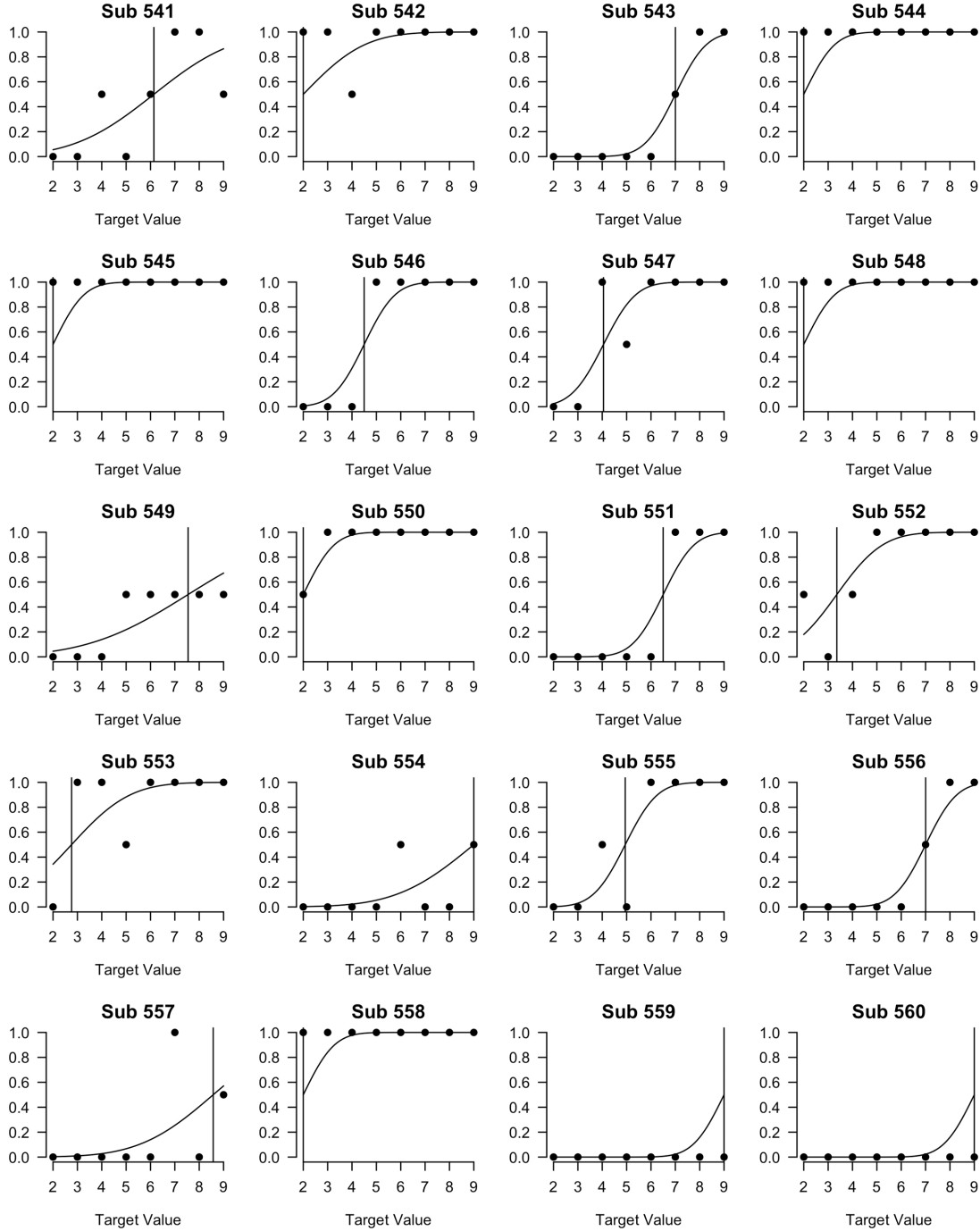

**Appendix 1—figure 29.** Psychometric functions linking target values to offloading choices. The average choice data is shown as dots. Panels show the individual curves for participants 541–560.

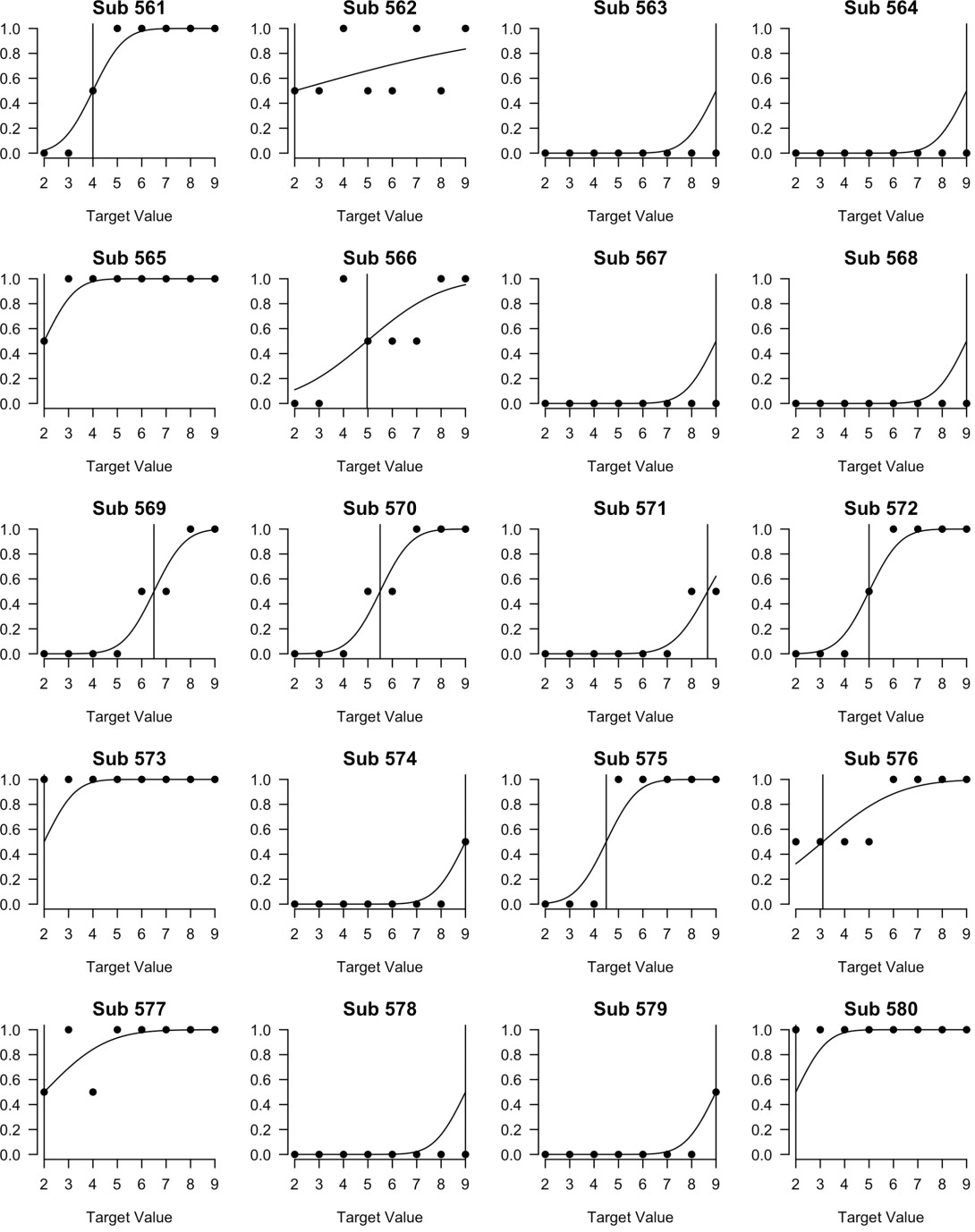

**Appendix 1—figure 30.** Psychometric functions linking target values to offloading choices. The average choice data is shown as dots. Panels show the individual curves for participants 561–580.

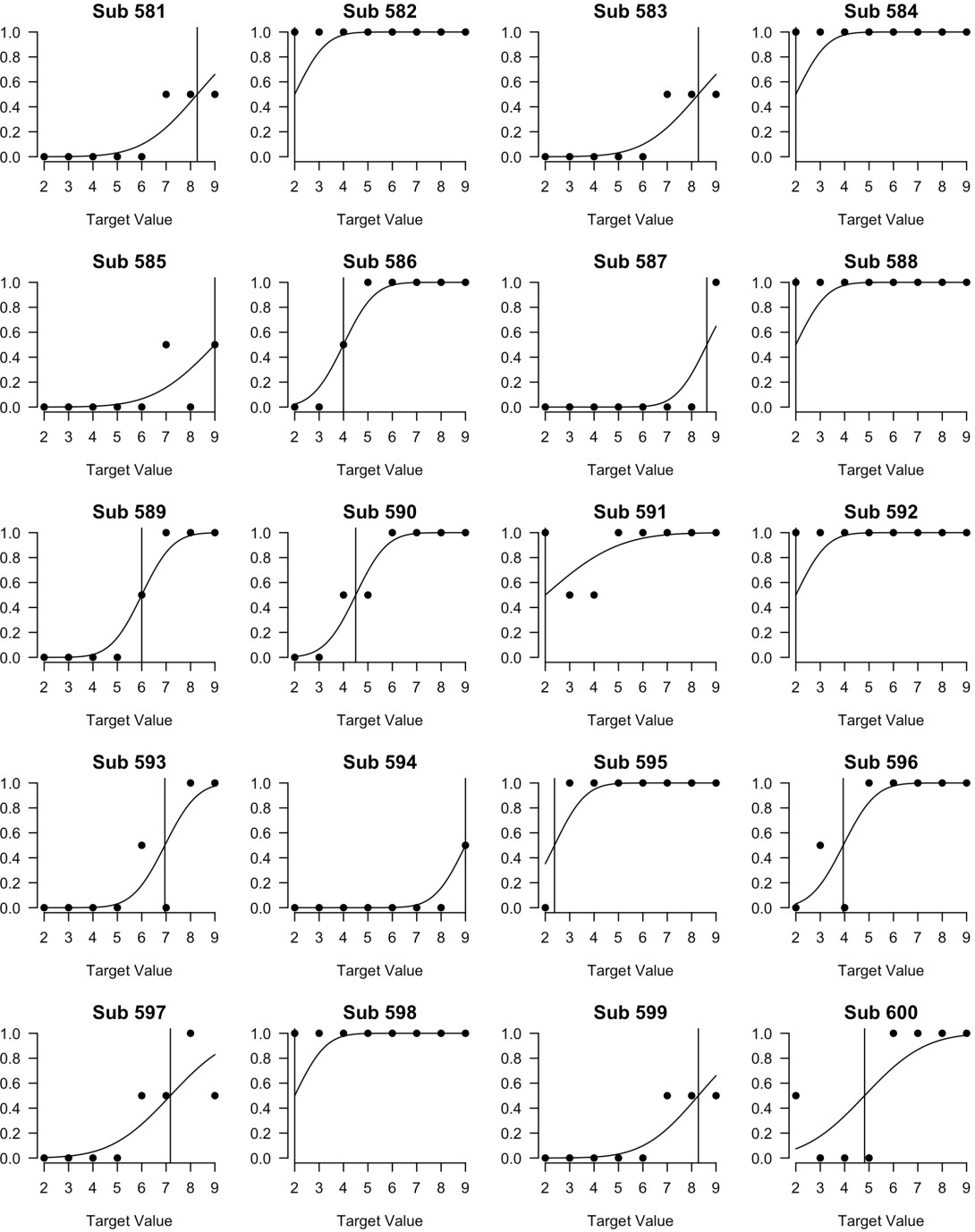

**Appendix 1—figure 31.** Psychometric functions linking target values to offloading choices. The average choice data is shown as dots. Panels show the individual curves for participants 581–600.

## Planned additional analyses

We preregistered several analyses that whilst planned were not thought out in detail and thus of a more exploratory nature.

First, we investigated the relationship between the transdiagnostic phenotypes and unaided task performance (accuracy on trials in which participants were not allowed to use a reminder). We predicted task performance (accuracy on FI trials) from the two transdiagnostic factors and our demographic covariates (age, gender, and education).

$$ACC_{FI} \sim CIT + AD + age + gender + education + \varepsilon$$

Neither AD, $\beta$=–0.02, SE = 0.05, $t$=–0.39, p=0.69; nor CIT, $\beta$=–0.06, SE = 0.05, $t$=–1.33, p=0.18, were significant predictors, suggesting there was no evidence to support meaningful performance differences for these two transdiagnostic phenotypes. The full results can be found in *Appendix 1— table 13* as well as in *Appendix 1—figure 32*.

**Appendix 1—table 13.** Predicting internal accuracy.
All continuous variables are *z*-transformed. SE = standard error; m=male; f=female; o=other.

|  | $\beta$ | SE | t | p |
|---|---|---|---|---|
| Intercept | 0.01 | 0.05 | 0.27 | 0.784 |
| AD | –0.02 | 0.05 | –0.39 | 0.693 |
| CIT | –0.06 | 0.05 | –1.33 | 0.183 |
| Age | –0.15 | 0.04 | –3.50 | <0.001 |
| gender1 (m vs. f) | –0.04 | 0.08 | –0.52 | 0.606 |
| gender2 (m vs. o) | 0.15 | 0.38 | 0.40 | 0.687 |
| education | 0.05 | 0.04 | 1.29 | 0.199 |

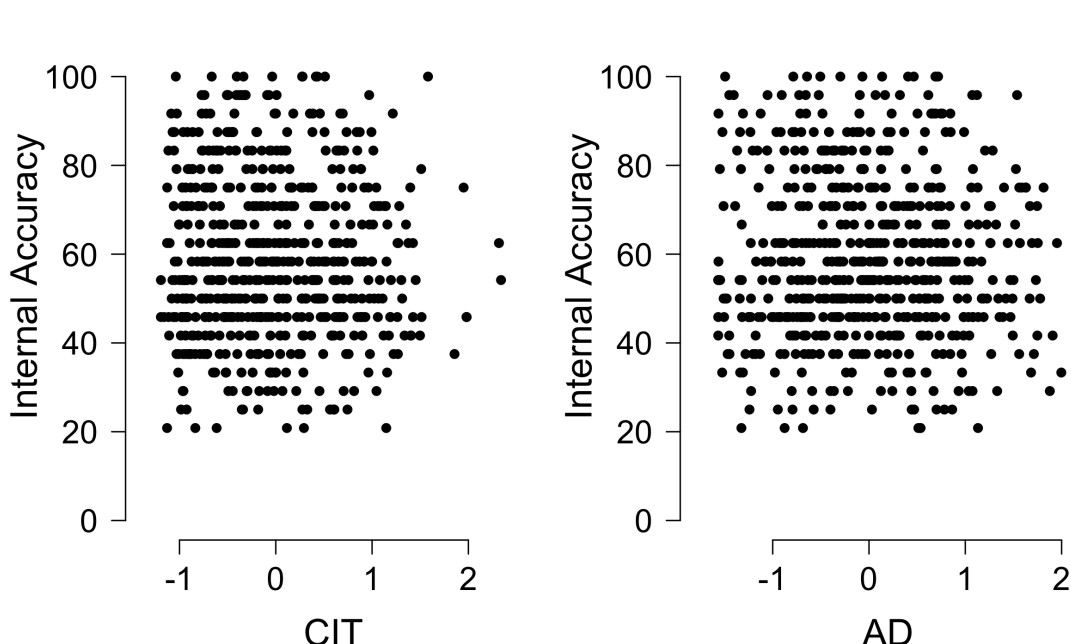

**Appendix 1—figure 32.** The distribution of the unaided task performance (internal accuracy) as a function of 'compulsive behaviour and intrusive thought' (CIT; left panel) and 'anxious depression' (AD; right panel).

Second, we planned to investigate whether participants would show a response 'stickiness' in their reminder use as reported by *Scarampi and Gilbert, 2020*, and whether such response perseverance would correlate with their CIT (e.g. *Shahar et al., 2021*) and AD scores. We conducted this analysis based on a subset of participants: only participants whose first trial was not a partial trial were included (choice condition, as no strategy was performed that could later be repeated). Furthermore, only participants who indicated with their strategy choice that they would not have chosen the strategy they were randomly assigned were included. This resulted in a subset of *N*=157 participants. We then calculated the proportion of the remaining trials in which participants chose to repeat this strategy and compared the resulting proportion to 50%. Participants repeated this strategy on only 31.6% of the remaining trials, significantly lower than 50%, *t*(156) = –8.43, p<0.001, *d*=0.67. We need to keep in mind that the strategy in question was the one they did not choose on the first trial. Most people

have an overall preference for or against reminders in this task independent of the manipulation of reward. A person who shows numbers lower than 50% might thus have repeatedly chosen the same strategy they had overwritten on that first trial, reflecting stable biases for their preferred response strategies. We furthermore tested whether this response perseverance was predicted by CIT or AD (and demographic covariates) fitting the following regression model to the data:

$$perseverance \sim CIT + AD + age + gender + education + \varepsilon$$

We indeed found that the effect was modulated by CIT: Compulsive individuals showed more response perseverance, $\beta$=0.20, SE = 0.09, $t$=2.17, p=0.03, whereas there was no significant effect on AD, $\beta$=–0.08, SE = 0.09, $t$=–0.85, p=0.40. As a conclusion, we can say that the more compulsive individuals show a tendency to repeat that first forced/overwritten trial later in the remaining experiment. However, caution should be advised to not over-interpret this effect as only a small subset of our sample was included in the analysis.

## Exploratory analyses

During our analysis, several additional questions arose, which we aimed to address with exploratory analyses.

First, we aimed to understand whether highly compulsive individuals would approach our task differently, potentially even struggling with it. For instance, some OCD patients prefer ordered sequences, and it might therefore have been aversive for our compulsive individuals to move the target circles out of the numbered order to set reminders. Additionally, they might have been put off by the scattered nature of the visual display and might have spent time rearranging circles, e.g., in a grid-like fashion. We therefore tested whether the transdiagnostic phenotypes were reliable predictors of response times (RTs), depending on condition with the following linear mixed model with random intercepts for participants:

$$RT \sim condition * (CIT + AD) + age + gender + education + \varepsilon$$

where *condition* denoted whether or not participants did the task with (FE) or without (FI) reminders. CIT did not significantly predict RT, $\beta$=–0.01, SE = 0.02, $t$(1153)=–0.42, p=0.67. AD, on the other hand, was a significant predictor of RT, $\beta$=–0.04, SE = 0.02, $t$(1228)=–2.05, p=0.04. There was furthermore a crucially significant interaction between CIT and *condition*, $\beta$=0.08, SE = 0.02, $t$(9580)=3.36, p<0.001. In other words, when reminders were possible, compulsive individuals were slower (0.04); but when reminders were not possible, compulsive individuals were faster (–0.04). There was no such interaction effect for the AD factor, $\beta$=–0.02, SE = 0.02, $t$(9577)=–0.82, p=0.41.

We followed up this analysis by fitting two additional linear mixed models with random intercepts for participants to gain insight into what actions highly compulsive individuals might have been performing during the reminder trials that might have led to the increase in RT. The first model predicted the trial-wise number of times that participants rearranged a circle they had previously already moved:

$$circles_{movedagain} \sim condition * CIT + AD + age + gender + education + \varepsilon$$

Compulsive individuals showed a tendency to this more often; however, this effect did not reach our level of significance, $\beta$=0.03, SE = 0.02, $t$(1245)=1.80, p=0.07.

The second model focused on a smaller subset of trials ($m$=4.2 trials per participant; min = 4, max = 12) in which more than one circle was moved and expressed the extent to which these circles were moved in their numbered order:

$$circles_{movedinorder} \sim condition * CIT + AD + age + gender + education + \varepsilon$$

Numerically and contrary to expectation, highly compulsive individuals showed a reduced tendency to move circles in their numbered order. However, this effect did not reach our level of significance, $\beta$=–0.06, SE = 0.04, $t$(929.5)=–1.61, p=0.107. Taken together, high CIT individuals took significantly longer on reminder trials, but we cannot say with certainty why, and this will therefore need to be the focus of future studies.

Second, we asked whether the transdiagnostic phenotypes affected the compensatory nature of reminder use (cf. Hypothesis 4), meaning people who need reminders more tend to be the ones who use them more. The motivation for this analysis was the compulsive individual's tendency towards a reduced reminder bias. To this end, we fit a regression model to predict the AIP from the OIP, the transdiagnostic phenotypes, and the demographic covariates:

$$AIP \sim OIP * (CIT + AD) + age + gender + education + \varepsilon$$

The key effects of interest were the interaction terms between the transdiagnostic phenotypes and the OIP.

Indeed, this seems to be the case reflected numerically in the interaction between CIT and OIP when predicting AIP, in other words, there was a stronger link between AIP and OIP in low compulsive individuals, compared to high compulsive individuals. However, this effect did not reach our required level of significance, $\beta$=–0.08, SE = 0.04, $t$=–1.75, p=0.08. There was no such interaction effect for the AD factor, $\beta$=0.01, SE = 0.04, $t$=0.34, p=0.73. In a follow-up analysis, we then investigated the influence of compulsivity on accuracy in the FE condition by fitting the following regression model:

$$ACC_{FE} \sim CIT + AD + age + gender + education + \varepsilon$$

Compulsive individuals were found to be less accurate on reminder trials, $\beta$=–0.12, SE = 0.5, $t$=–2.53, p=0.01, pointing towards a picture of not only impaired reminder setting but also impaired reminder use. We note that whilst this is an interesting finding, it certainly needs follow-up in future studies to understand the mechanisms at play.

