## [Editor Report · eLife Assessment]

This **important** work addresses the relationship between the transdiagnostic compulsivity dimension and confidence as well as confidence-related behaviours like reminder setting. The relationship between confidence and compulsive disorders has recently received a lot of attention and has been considered to be a key cognitive change. The authors paired an elegant experimental design and pre-registration to give **convincing** evidence of the relationship between compulsivity, reminder setting, and confidence. In the revised version they thoroughly addressed the reviewer's comments, in particular adding new analyses clarifying how their findings relate to prediction error based learning as well as presenting additional recovery analyses and psychometric curves further strengthening the manuscript.

---

## [Referee Report · Reviewer #1 (Public review)]

Summary:

Boldt et al test several possible relationships between trandiagnostically-defined compulsivity and cognitive offloading in a large online sample. To do so, they develop a new and useful cognitive task to jointly estimate biases in confidence and reminder-setting. In doing so, they find that over-confidence is related to less utilization of reminder-setting, which partially mediates the negative relationship between compulsivity and lower reminder-setting. The paper thus establishes that, contrary to the over-use of checking behaviors in patients with OCD, greater levels of transdiagnostically-defined compulsivity predicts less deployment of cognitive offloading. The authors offer speculative reasons as to why (perhaps it's perfectionism in less clinically-severe presentations that lowers the cost of expending memory resources), and sets an agenda to understand the divergence in cognitive between clinical and nonclinical samples. Because only a partial mediation had robust evidence, multiple effects may be at play, whereby compulsivity impacts cognitive offloading via overconfidence and also by other causal pathways.

Strengths:

The study develops an easy-to-implement task to jointly measure confidence and replicates several major findings on confidence and cognitive offloading. The study uses a useful measure of cognitive offloading - the tendency to set reminders to augment accuracy in the presence of experimentally manipulated costs. Moreover, the utilizes multiple measures of presumed biases -- overall tendency to set reminders, the empirically estimated indifference point at which people engage reminders, and a bias measure that compares optimal indifference points to engage reminders relative to the empirically observed indifference points. That the study observes convergenence along all these measures strengthens the inferences made relating compulsivity to the under-use of reminder-setting. Lastly, the study does find evidence for one of several a priori hypotheses and sets a compelling agenda to try to explain why such a finding diverges from an ostensible opposing finding in clinical OCD samples and the over-use of cognitive offloading.

Weaknesses:

Although I think this design and study are very helpful for the field, I felt that a feature of the design might reduce the tasks's sensitivity to measuring dispositional tendencies to engage cognitive offloading. In particular, the design introduces prediction errors, that could induce learning and interfere with natural tendencies to deploy reminder-setting behavior. These PEs comprise whether a given selected strategy will be or not be allowed to be engaged. We know individuals with compulsivity can learn even when instructed not to learn (e.g., Sharp, Dolan and Eldar, 2021, Psychological Medicine), and that more generally, they have trouble with structure knowledge (eg Seow et al; Fradkin et al), and thus might be sensitive to these PEs. Thus, a dispositional tendency to set reminders might be differentially impacted for those with compulsivity after an NPE, where they want to set a reminder, but aren't allowed to. After such an NPE, they may avoid moreso the tendency to set reminders. Those with compulsivity likely have superstitious beliefs about how checking behaviors lead to a resolution of catastrophes, that might in part originate from inferring structure in the presence of noise or from purely irrelevant sources of information for a given decision problem.

It would be good to know if such learning effects exist, if they're modulated by PE (you can imagine PEs are higher if you are more incentivized - e.g., 9 points as opposed to only 3 points - to use reminders, and you are told you cannot use them), and if this learning effect confounds the relationship between compulsivity and reminder-setting.

A more subtle point, I think this study can be more said to be an exploration than a deductive of test of a particular model -> hypothesis -> experiment. Typically, when we test a hypothesis, we contrast it with competing models. Here, the tests were two-sided because multiple models, with mutually exclusive predictions (over-use or under-use of reminders) were tested. Moreover, it's unclear exactly how to make sense of what is called the direct mechanism, which is supported by the partial (as opposed to complete) mediation.

Comments on revisions:

I have the following final comments for your manuscript revisions:

To improve the clarity of the work, I suggest a final note to the authors to say more explicitly that objective accuracy has a finer resolution *due to the number of "special circles" per trial* in their task. This task detail got lost in my read of the manuscript, and confused me with respect to the resolution of each accuracy measure. Similarly for clarification, they could point out that their exclusion criteria removes subjects that have lower OIP than their AIP analysis allows (which is good for comparison between OIP and AIP). Thus, it removes the possibility that very poor performing subjects (OIP) are forced to have a higher than actual AIP due to the range.

---

## [Author Response]

The following is the authors’ response to the current reviews.

**Reviewer #1:**
(1) To improve the clarity of the work, I suggest a final note to the authors to say more explicitly that objective accuracy has a finer resolution *due to the number of "special circles" per trial* in their task. This task detail got lost in my read of the manuscript, and confused me with respect to the resolution of each accuracy measure.

We agree with the reviewer that this would be a useful clarification and have therefore added the following statement to the Methods section on p. 20:

“It should be noted that the OIP has a slightly finer resolution due to the number of special circles per trial.”

(2) Similarly for clarification, they could point out that their exclusion criteria removes subjects that have lower OIP than their AIP analysis allows (which is good for comparison between OIP and AIP). Thus, it removes the possibility that very poor performing subjects (OIP) are forced to have a higher than actual AIP due to the range.

We agree this would be a useful statement to add and have included the following sentence in the Supplement on p. 8:

“Such a restriction of the threshold parameter was intended to increase the comparability between AIP and OIP, and hence improved the calculation of the reminder bias.”

The following is the authors’ response to the previous reviews.

**Reviewer #1:**
(1) Upon reading their response to the question I had regarding AIP and OIP, a few more questions came up regarding OIP, AIP, how they're calculations differ, and how the latter was computed in R. I hope these help readers to clarify how to interpret these key measures, and the hypotheses that rely upon them.Regarding fitting, and in relation to power, is16 queries adequate to estimate an AIP using the R's quickpsy? That is, assuming some noise in the choice process, how recoverable is a true indifference points from 16 trials? If there's a parameter recovery analysis (ie generating choice via the fitting parameters, which will have built-in stochasticity, and seeing how well you recover the parameter) of interest would be helpful. It may help to characterize why the present study might differ from prior studies (maybe a power issue here).

The reviewer is absolutely correct that we should have provided more detail when describing our fitting procedure for the psychometric curves. We have now addressed this by adding the following statements to the Methods section and Supplement:

Page 20 in the main manuscript: “Fitting was done using the *quickpsy* package in *R* and more detail is given in the Supplement.”

Pages 8 and 9 in the Supplement:

“Psychometric curve fitting

We used the *quickpsy* package in R to fit psychometric curves to each participant’s choice data to derive their actual indifference point (*AIP*), which was operationalised as the threshold parameter when predicting reminder choices from target values. We restricted the possible parameter ranges from 2 to 9 for the threshold parameter and from 1 to 500 for the slope parameter, based on the task’s properties and pilot data. Apart from those parameter ranges, we used only default settings of the *quickpsy()* function.

Each participant has only 16 trials (2 for each target value) contribute to the curve fitting. To understand the robustness of the AIP based on such limited data, we conducted a parameter recovery analysis. We simulated 16 trials based on each psychometric function and re-ran the curve fitting based on those simulated choices. There was close correspondence between the actual and recovered threshold parameters (or AIPs) with a correlation of *r* = 0.97, *p* < 0.001 (see also Figure S1). In contrast, the slope parameter—which was not central to any of our analyses—exhibited greater variability during the initial fitting. This increased uncertainty likely contributed to its poor recovery in the simulation, as evidenced by a near-zero correlation (*r* = −0.01, *p* = 0.82).”

(2) Along these lines, it would be helpful for the reader to actually see the individual psychometric curve, now how quickpsy was used (did you fit left and right asymptotes), etc, to understand how that fitting procedure works and how the assumptions of the fitting procedure compare to what can be gleaned through seeing the choice curves plotted.

As stated above, we used default settings of the *quickpsy()* function and hence assumed symmetric asymptotes at 0 and 1. However, the reviewer mentions “left and right asymptotes”, so maybe this question is about restricting the possible parameter range for the threshold, which we restricted to values from 2 to 9, as described above.

Regarding the individual curves, we have now include the following statement on page 9 in the Supplement: “Figures S2 to S31 show the individual psychometric curves that were estimated for each participant.” Please refer to the Supplement for the added figures.

(3) A more full explanation of quickpsy, its parameters, and how choice curves look might also generate interesting further questions to think about with respect to biases and compulsivity. Two individuals might have similar indifference points, but an asymptote might reflect a bias to always have some percent chance of for example to take the reminders even at the lowest offer available for them.

We agree that this is an interesting focus which we will keep in mind for future studies.

(4) Regarding comparing OIP to AIP:For OIP, as far as I can understand, the resolution of it is decreased compared to AIP. Accuracies for OIP can only be 0/4,1/4,2/4,3/4, or 4/4. Yet, the resolution for AIP is the full range of offers (2 to 9) with respect to the parameter of interest (the indifference point). Could this bias the estimation of OIP for instance, someone who scored 25% might actually be much closer to either 50 or 0, but we can't tell due to resolution?

As mentioned in response to comment (1), we restricted the parameter range for the thresholds to 2 to 9 to increase comparability. The reviewer is right to point out that the OIP still has lower resolution than the AIP, which is one of the downsides of having a shortened paradigm (cf. the longer version in Gilbert et al., 2019), which is optimised for online testing, especially if used in combination with additional questionnaires. We have no reason to believe though that this could have led to any bias, especially none that would contribute to the individual differences which are the main focus of our study.

Gilbert, S. J., Bird, A., Carpenter, J. M., Fleming, S. M., Sachdeva, C., & Tsai, P.-C. (2020). Optimal use of reminders: Metacognition, effort, and cognitive offloading. *Journal of Experimental Psychology: General*, *149*(3), 501–517. https://doi.org/10.1037/xge0000652

(5) Additionally, it seems like the upper and lower bounds of OIP (0 and 10) differ from AIP (2 and 9). Could this also introduce bias (for example, if someone terrible performance, the mean would artificially be higher under AIP than OIP because the smallest indifference point is 2 under AIP, but could be 0 under OIP).

See our response to comment (1), we fixed the range to 2 to 9 (which was the range of target values used in our study).

(6) Finally seeing how CIT actually corresponds to accuracy overall (not a relative measure like AIP compared to OIP) I think would also be helpful as this is related to most points noted above.

We included the suggested test as an exploratory analysis on pages 42-43 in the Supplement: “Third, we were interested in how the transdiagnostic phenotypes would correspond to performance. We therefore fitted a model which predicted internal accuracy (that is, unaided task performance on trials where no reminders could be used) from AD, CIT, and the other covariates (age, education and gender). We found that neither AD, β = -0.02, SE = 0.05, *t* = 0.44, *p* = 0.658, nor CIT, β = -0.03, SE = 0.05, *t* = -0.66, *p* = 0.510, predicted internal accuracy.

The full results can be found in Table S13 as well as in Figure S32.”